# mcRigor: a statistical method to enhance the rigor of metacell partitioning in single-cell data analysis

**Pan Liu[1] & Jingyi Jessica Li** [ORCID] [1,2,3] ✉

In single-cell data analysis, addressing sparsity often involves aggregating the profiles of homogeneous single cells into metacells. However, existing metacell partitioning methods lack checks on the homogeneity assumption and may aggregate heterogeneous single cells, potentially biasing downstream analysis and leading to spurious discoveries. To fill this gap, we introduce mcRigor, a statistical method to detect dubious metacells, which are composed of heterogeneous single cells, and optimize the hyperparameter(s) of a metacell partitioning method. The core of mcRigor is a feature-correlation-based statistic that measures the heterogeneity of a metacell, with its null distribution derived from a double permutation scheme. As an optimizer for existing metacell partitioning methods, mcRigor has been shown to improve the reliability of discoveries in single-cell RNA-seq and multiome (RNA + ATAC) data analyses, such as uncovering differential gene co-expression modules, enhancer-gene associations, and gene temporal expression. Moreover, mcRigor enables benchmarking and selection of the most suitable metacell partitioning method with optimized hyperparameter(s) tailored to a specific dataset, ensuring reliable downstream analysis. Our results indicate that among existing metacell partitioning methods, MetaCell and SEACells consistently outperform MetaCell2 and SuperCell, albeit with the trade-off of longer runtimes.

Single-cell sequencing technologies have catalyzed a paradigm shift in genomics by uncovering cellular heterogeneity with unprecedented resolution across multiple modalities, including transcriptomics via single-cell RNA sequencing (scRNA-seq)[1–3], epigenomics through single-cell assay for transposase-accessible chromatin using sequencing (scATAC-seq)[4,5], and multiome assays that simultaneously measure RNA-seq and ATAC-seq[6,7]. The majority of these technologies are high-throughput and droplet-based, capable of profiling millions of cells, but they are often compromised by high sparsity in the sequencing read counts due to low per-cell sequencing depth and imperfections in the reverse transcription and amplification steps[8].

The high sparsity presents a substantial challenge for data analysis[9], with common strategies to mitigate it including imputation and metacell partitioning. Imputation addresses sparsity by predicting missing feature measurements, where features represent genes or chromatin regions, using cells and/or features with similar measurement profiles. Numerous imputation methods have been developed for single-cell data, including scImpute[10], SAVER[11], MAGIC[12], and DCA[13], as well as deep generative models[14,15]. Imputation methods have the advantage of retaining the full set of single cells; however, they can sometimes induce false positives in the downstream differential gene expression (DGE) detection[16] and may suffer from oversmoothing[11,17],

[1]Department of Statistics and Data Science, University of California, Los Angeles, CA, USA. [2]Present address: Biostatistics Program, Public Health Sciences Division, Fred Hutchinson Cancer Center, Seattle, WA, USA. [3]Present address: Department of Biostatistics, University of Washington, Seattle, WA, USA. ✉e-mail: lijy03@fredhutch.org

which creates artificial similarities among cells. As an alternative to imputation, the metacell approach groups cells representing the same cell state into a metacell and uses the metacell's measurement profile, typically obtained by averaging the single-cell measurement profiles, for subsequent analysis[17]. The metacell approach is expected to reduce noise and thereby accentuate biological signals that are often obscured in sparse datasets.

The metacell concept differs from the pseudobulk approach, though both involve cell aggregation. Specifically, a pseudobulk is created by merging all cells within a predefined cell population—typically a cell type—into a single profile, while a metacell aggregates a much smaller, homogeneous group of cells, allowing for multiple metacells within a single cell type. The pseudobulk approach reduces data sparsity and enables the use of computational methods designed for bulk data[18,19]. However, by merging all cells of a cell type into one pseudobulk, this approach removes all within-cell-type variation. In contrast, metacells aim to preserve this variation, maintaining the resolution advantage of single-cell data that allows cell-type-specific analysis. For instance, metacell partitioning has been demonstrated to be beneficial for gene co-expression analysis[20]. While a pseudobulk sample does not permit cell-type-specific gene co-expression analysis (because each gene has only one aggregated expression level per cell type, making it impossible to calculate correlations between two genes)[21,22], performing co-expression analysis using single cells within a cell type is often hindered by the low sensitivity and high technical noise of scRNA-seq data[23]. The metacell approach offers a valuable middle ground by enhancing co-expression signals within specific cell types.

Despite the increasing use of the metacell concept in high-profile single-cell studies—such as investigating cell differentiation states[24], characterizing tissue compartments and diverse cell populations[25–27], patient stratification for individualized immunotherapy design[28], and temporal analysis of cell transcriptomes[29,30]—there is still no rigorous definition of metacell or a universally accepted strategy for constructing metacells. This lack of consensus can result in inconsistencies across studies utilizing the metacell concept, undermining the reliability of analysis outcomes.

In addition to the various approaches used for partitioning single cells into metacells in in-house data analyses, several general methods have been developed for this purpose. The most popular ones include MetaCell[17], MetaCell2[31], SuperCell[32], and SEACells[33]. MetaCell employs a k-nearest neighbor (kNN) graph of cells, uses graph resampling and clustering to update the graph, and finally identifies metacells as small clusters. It also includes additional steps to detect and exclude outlier cells that are not incorporated into any metacells. Developed by the same authors as MetaCell, MetaCell2 is designed for faster performance through divide-and-conquer. SuperCell applies a walktrap clustering method to a PCA-derived kNN graph of cells. SEACells uses a kernel to define a cell-cell similarity matrix, treating these similarities as cell embeddings for archetypal analysis, with the resulting archetypes used to identify metacells. However, these methods can produce different metacell partitions, which are also influenced by the hyperparameters they employ[34]. This lack of consensus leaves users uncertain about which metacell partition to use and to what extent the resulting metacell profiles preserve biological signals. Therefore, a formal definition and an evaluation standard for metacells are needed to guarantee principled metacell partitioning and ensure unbiasedness in downstream analyses.

To fill this gap, we propose a statistical definition of metacell and accordingly develop mcRigor, a novel statistical method to enhance the rigor of metacell partitioning in single-cell data analysis. Theoretically, a metacell is defined as a homogeneous group of single-cell profiles that could be viewed as resamples from the same original cell, with any variation within a metacell attributed solely to technical measurement errors, termed technical variation, rather than biological differences, termed biological variation. Built upon this definition, mcRigor can identify dubious metacells that are heterogeneous and violate this definition, while also optimizing the metacell partitioning strategy to ensure reliable metacell construction. Our results demonstrate that mcRigor successfully identifies and removes dubious metacells, revealing the COVID-related co-expression of adaptive immune response genes, which is enriched in COVID-19 patients compared to healthy individuals. We also show that mcRigor enhances gene regulatory analysis by revealing enhancer-gene associations that are obscured in single-cell multiome data or by the presence of dubious metacells, while excluding spurious associations biased by these metacells. Moreover, mcRigor balances the trade-off between data sparsity and signal distortion, identifying optimal metacell partitions to distinguish biological from non-biological zeros, detect differentially expressed (DE) genes, and reveal temporal trajectories of cellular immune responses.

## Results

### Overview of the mcRigor method

The mcRigor method is designed to improve the rigor of metacell partitioning and the reliability of downstream analyses by distinguishing between trustworthy and dubious metacells, optimizing the hyperparameters of any metacell partitioning method, and facilitating comparisons across different metacell partitions. mcRigor is built on the definition of a metacell as a homogeneous group of single cells that share the same biological state, characterized by the same true profile of features (e.g., genes or chromatin regions), such that the cells' observed profiles can be considered resampled from the same original cell. The essence of this definition, detailed in the "Methods" section, is that within a metacell, any variability among cells should be solely attributable to the technical variation. Aggregating such cells into a metacell helps reduce technical noise while preserving true biological signals, ensuring that the metacell's averaged profile unbiasedly represents the underlying biological state of these cells.

mcRigor has two main functionalities: detecting dubious metacells and optimizing metacell partitioning. For its dubious metacell detection functionality, mcRigor takes a single-cell dataset and an initial metacell partition as input, producing a refined set of trustworthy metacells as output. Specifically, mcRigor detects dubious metacells in the input partition using a statistical approach that assesses the internal homogeneity of each metacell (Fig. 1a). The approach is grounded in the premise that within a trustworthy metacell, which is internally homogeneous, feature correlations are driven exclusively by technical noise and should therefore be minimal. At the core of this approach, we define a feature-correlation-based statistic called the divergence score (mcDiv) for each metacell, which measures the deviation of the within-metacell feature correlation matrix from the identity matrix that indicates no feature correlation. Note that mcDiv includes a normalization factor derived from within-feature permutation, which simulates the baseline scenario where the same features become uncorrelated. A larger mcDiv value indicates greater heterogeneity within the metacell, suggesting it is more dubious.

To set mcDiv thresholds for identifying dubious metacells, mcRigor constructs a null divergence score (mcDiv$^{null}$) for each metacell through within-cell permutation, where feature values are shuffled independently for each cell, preserving cell library sizes (i.e., the sum of feature values per cell) while disrupting any biological correlations among features. As varying cell library sizes can induce spurious feature correlations, it is essential to preserve cell library sizes in defining mcDiv$^{null}$. Note that mcDiv$^{null}$ also includes a normalization factor derived from within-feature permutation applied to the features already permuted in the previous within-cell permutation step. Therefore, the calculation of mcDiv$^{null}$ involves two permutation steps: within-cell permutation followed by within-feature permutation, a novel procedure we refer to as the double permutation. Double

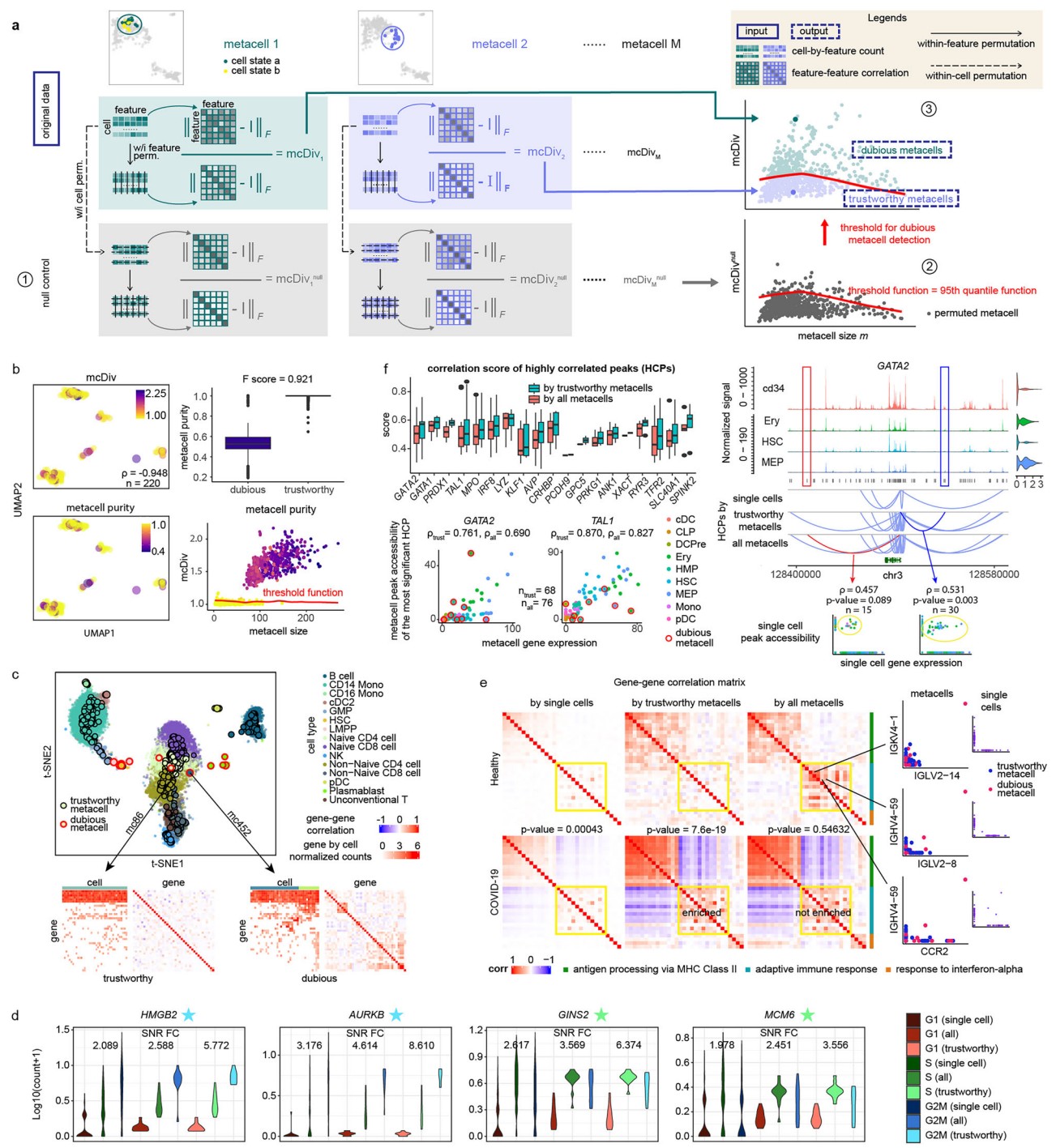

permutation is necessary because mcDiv and mcDiv$^{null}$ involve different features and thus require different normalization factors. Pooling the mcDiv$^{null}$ values from all metacells, mcRigor establishes thresholds to distinguish dubious metacells from trustworthy ones. These thresholds are conditional on the metacell size, i.e., the number of single cells within the metacell, recognizing that metacells with smaller sizes may exhibit greater variability in mcDiv values. Specifically, to determine if each metacell is dubious, mcRigor employs a sliding window approach to compute a local threshold for the metacell, defined as the 95th percentile of mcDiv$^{null}$ values among similarly sized metacells. Metacells whose mcDiv values exceed the metacell-size-specific thresholds are flagged as dubious, indicating likely heterogeneity within the cells they contain.

To ensure the reliability of downstream analyses, users may choose to remove the detected dubious metacells. However, in some cases, this removal may result in the loss of valuable information–particularly when dubious metacells contain single cells from rare cell types. In such scenarios, re-partitioning the dubious metacells into smaller trustworthy metacells may be a more desirable strategy. We discuss an extension of mcRigor to implement this strategy in the "Discussion" section.

For its metacell partition optimization functionality, mcRigor inputs a single-cell dataset along with metacell partitioning methods and their candidate hyperparameter settings, returning the optimized hyperparameter setting for each method and assessing the quality of each method's optimal metacell partition. That is, mcRigor optimizes

**Fig. 1 | mcRigor detects dubious metacells and rectifies downstream analysis for both scRNA-seq and multiome (RNA + ATAC) data. a** Schematic of the mcRigor method for dubious metacell detection. **b** mcRigor effectively assesses metacell heterogeneity and detects dubious metacells within the MetaCell method's partitioning on semi-synthetic data. Left: UMAP plots showing partitioned metacells, colored by mcDiv values compared to metacell purity (ground truth). A strong negative Spearman correlation ($\rho = -0.948$) was observed between mcDiv values and purity. Right: mcRigor distinguishes between dubious and trustworthy metacells with high accuracy. The box plots (top right) show the medians (center lines), the 25th and 75th percentiles (box bounds), and the whiskers (black lines) extending to 1.5 times the interquartile range from the box. **c** Dubious metacells identified by mcRigor exhibit internal heterogeneity and may occasionally appear as outliers, while trustworthy metacells remain internally homogeneous. Bottom: Heatmaps of gene-by-cell counts and gene-by-gene correlations for a trustworthy and a dubious metacell. **d** mcRigor enhances cell-cycle marker gene expression within cell lines. The violin plots display the $\log_{10}(\text{count} + 1)$ expression levels of four cell-cycle marker genes across single cells, all metacells ("all"), and trustworthy metacells ("trustworthy"). SNR FC represents the fold change in signal-to-noise ratio for the phase associated with each marker gene (indicated by a star) relative to the other two phases. **e** mcRigor reveals enriched co-expression of an adaptive immune response gene module (highlighted in yellow) in COVID-19 samples (bottom row) compared to healthy controls (top row), based on SuperCell partitions at $\gamma = 20$. Left heatmaps: Gene-gene correlation matrices for three key gene modules under COVID-19 and healthy control conditions, based on three data types: single cells, trustworthy metacells identified by mcRigor, and all metacells. Each $p$-value for the correlation comparisons in the adaptive immune response gene module between the two conditions for each data type was obtained using a one-sided Wilcoxon rank-sum test. Right scatter plots: Three gene pairs showing artifact correlations caused by dubious metacells, with no apparent correlations in single-cell data. **f** Applying mcRigor to the original metacell partition from the SEACells paper empowers gene regulatory inference (left) and produces reliable discoveries (right). Left: box plots (showing median, interquartile range, and whiskers extending to 1.5 times the interquartile range) and scatter plots showing that removing dubious metacells increases the correlations between genes and their highly correlated peaks (HCPs). Right: removing dubious metacells uncovers a validated HCP for *GATA2* (blue) and filters out a weakly supported one (red), corroborated by single-cell data. Each $p$-value assessing the significance of the correlation between nonzero *GATA2* expression and nonzero accessibility of the corresponding peak at the single-cell level was obtained using a two-sided Spearman's rho test.

metacell partitioning for a specific dataset by simultaneously evaluating various candidate method-hyperparameter configurations (Fig. 2a), with each configuration representing a metacell method combined with its hyperparameter setting. The most critical hyperparameter is the granularity level, $\gamma$, which represents the average number of single cells per metacell, since $\gamma$ is required by all existing metacell methods. For each metacell partition obtained from a method-hyperparameter configuration, mcRigor calculates an evaluation score that balances the dubious rate (the proportion of cells in dubious metacells) and the sparsity level in the aggregated data (measured by the proportion of zeros in the metacell expression profiles), recognizing that these two factors often trade off against each other (metacells containing more cells are more likely to be dubious but have less sparse profiles). The configuration that maximizes this evaluation score is considered optimal, striking a balance between preserving biological signals and minimizing technical noise. Notably, mcRigor is designed to be flexible and applicable across different metacell methods and single-cell data modalities.

### Validation of mcRigor using barcode multiplet data

Our statistical definition of metacell and the mcRigor method are both built upon the assumption that the observed variation among cells consists of two components: biological variation and technical variation. A natural question arises about the validity of this assumption, particularly regarding the presence of technical variation. We addressed this question by investigating technical variation using barcode multiplets identified in droplet-based single-cell assays.

A barcode multiplet refers to a set of cell-like observations in which each observation is assigned a unique cell barcode but actually originates from the same physical cell. This occurs when a single cell is tagged with multiple distinct barcodes during the droplet-based sequencing process, leading to multiple barcodes being incorrectly interpreted as separate cells[35]. Note that this term "barcode multiplet" should not be confused with the term "multiplet" as used in the doublet detection literature, where multiplets stand for multiple cells enclosed by a single droplet. A typical type of barcode multiplets, referred to as "barcode multiplets caused by heterogeneous beads" by ref. 36, form when multiple beads are encapsulated within a single droplet containing one cell. Consequently, the mRNA fragments captured by these beads are all sampled from the same pool of mRNA fragments from that cell[35]. Therefore, it is reasonable to assume that the variation observed within a barcode multiplet approximates the technical variation present among single cells in the same biological state.

In ref. 36, Lareau et al. designed a computational method, called bead-based ATAC processing (bap), to identify barcode multiples from a public droplet-based scATAC-seq dataset of 5000 peripheral blood mononuclear cells (PBMCs)[37]. In total, 16 barcode multiplets (caused by heterogeneous beads) were identified, each consisting of three to six cell-like observations. We found the observations within each barcode multiplet to be dispersed, suggesting the existence of substantial technical variation (multiplets 1–9 in Supplementary Fig. 1a). Furthermore, we tested mcRigor on this dataset by treating each barcode multiplet as a trustworthy metacell, as the cell-like observations in each barcode multiplet represent the same biological state. Remarkably, mcRigor successfully identified all 16 barcode multiplets as trustworthy metacells.

### Assessment of mcRigor's accuracy in detecting dubious metacells

Since barcode multiplets can only approximate trustworthy metacells but not dubious ones, we next assessed mcRigor's accuracy in detecting dubious metacells. We began with a simulation where the trustworthiness of metacells was known. Using the scDesign3 simulator[38], we generated a semi-synthetic dataset with cells of known biological states. This was done by modeling gene expression variation and separating it into biological and technical variations based on our model assumptions, using a reference dataset of bone marrow mononuclear cells measured by CITE-seq[39] (the `bmcite` dataset from the R package `SeuratData`[40]). Details of the semi-synthetic dataset generation process are provided in the "Methods" section. Our semi-synthetic dataset contains five major cell types—B cells, T cells, progenitor cells, natural killer (NK) cells, and monocytes or dendritic cells (Mono/DC)—and closely resembles the reference dataset (Supplementary Fig. 1b, c). By separating biological and technical variations, we were able to generate 50 ground-truth metacells, each corresponding to a distinct biological state, and then simulate single cells within each metacell to reflect technical variation, with the true granularity level set to $\gamma^* = 50$. Using the ground-truth metacells, we performed two tasks: (1) comparison of metacell partitioning methods by evaluating whether their constructed metacells were trustworthy or dubious, and (2) verification of mcRigor's ability to distinguish between trustworthy and dubious metacells. Both tasks were facilitated by the realistic nature of the semi-synthetic data and the availability of ground truth indicating metacell trustworthiness.

We applied the three popular metacell partitioning methods—MetaCell, SEACells, and SuperCell—each at varying granularity levels to this semi-synthetic dataset, obtaining a metacell partition for each

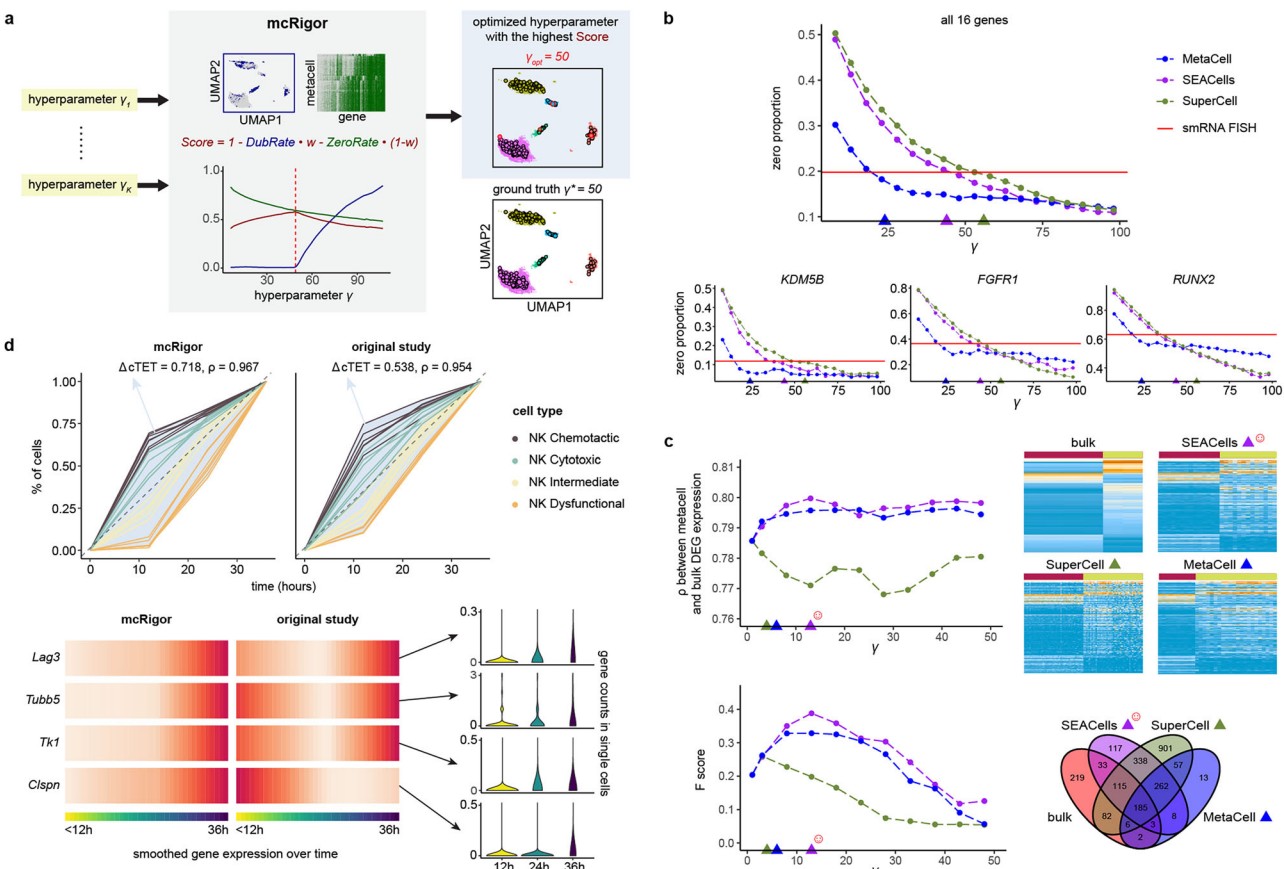

**Fig. 2 | mcRigor optimizes the metacell method and hyperparameter selection for various single-cell data analyses. a** Schematic of the mcRigor method for optimizing metacell partitioning, using Score as the optimization criterion to balance DubRate and ZeroRate, illustrated with the optimization of MetaCell partitions on semi-synthetic data as an example. **b** Line plots showing the zero proportions in metacell partitions generated by three methods (MetaCell, SEA-Cells, and SuperCell) across varying granularity levels ($\gamma$). The optimized metacell partitions (triangles) closely align with the zero proportion observed in smRNA FISH data (red line). **c** mcRigor optimizes the metacell method and hyperparameter selection for DGE analysis. Top: a line plot and heatmaps comparing the expression of the top 200 bulk DE genes across various metacell partitions to their expression in bulk data. The line plot depicts the concordance between the bulk and metacell profiles. Bottom: a line plot showing $F$-scores and a Venn diagram comparing the DE genes identified from various metacell partitions with those

detected from bulk data. The optimal metacell partition (SEACells with $\gamma = 13$, marked by the red smiling face) achieves the highest concordance (Pearson correlation $\rho = 0.800$) and $F$-score (0.400). **b**, **c** The colored triangles indicate the optimal $\gamma$ values selected by mcRigor for the three methods. **d** mcRigor's optimized metacell partition better reveals temporal immune cell trajectories compared to the original metacell partition from the Zman-seq study[30]. Top: Line plots comparing the metacells' continuous tumor exposure time (cTET) values calculated from mcRigor's optimized partition to those from the original partition. The value of $\Delta$cTET, proportional to the size of the light blue area, indicates the distinction of tumor transitional stages. Bottom: smoothed gene expression profiles of four marker genes in mcRigor's optimized partition compared to the original partition. Following optimization, the genes' expression patterns align more closely with biological expectations.

method and granularity level configuration. We also noticed the recent publication of a new metacell method, MetaQ[41], during the revision of this manuscript, and therefore included it in our comparison on this semi-synthetic dataset to provide a more comprehensive analysis. For evaluation, within a metacell partition, we defined the purity of a metacell as the highest fraction of cells originating from the same ground-truth metacell. Based on purity, the ground truth for metacell trustworthiness was defined as follows: metacells with a purity of 1 were considered truly trustworthy, while all others were considered truly dubious. Using this ground truth, we compared the four methods at each granularity level by evaluating the resulting metacell partitions based on two criteria: (1) the proportion of dubious metacells relative to the total number of metacells, and (2) the dubious rate, defined as the proportion of single cells assigned to dubious metacells out of all single cells. The results show that MetaCell, SEACells, and MetaQ performed better than SuperCell, yielding more reliable metacell partitions. MetaCell demonstrated a lower proportion of dubious metacells and a lower dubious rate compared to SEACells and MetaQ, but this came at the expense of assigning fewer single cells to

metacells. Specifically, at the granularity level $\gamma = 60$, which was above the true granularity level $\gamma^* = 50$, MetaCell, SEACells, SuperCell, and MetaQ produced 20.5%, 24.6%, 26.9%, and 25.6% dubious metacells, with dubious rates of 0.289, 0.308, 0.464, and 0.336, respectively. When the granularity level $\gamma$ was set to 30, below the true granularity level, MetaCell, SEACells, SuperCell, and MetaQ produced 0.2%, 0.2%, 25.3%, and 0.5% dubious metacells, with dubious rates of 0.003, 0.005, 0.380, and 0.005, respectively. At the true granularity level $\gamma = 50$, MetaCell, SEACells, SuperCell, and MetaQ produced 0.4%, 10.1%, 28.4%, and 7.8% dubious metacells, with dubious rates of 0.003, 0.161, 0.453, and 0.092, respectively.

With the ground-truth metacell purity values and trustworthiness, we then applied mcRigor to the metacell partitions produced by MetaCell, SEACells, SuperCell, and MetaQ, to test if mcRigor can accurately detect the dubious metacells from each partition. For metacell partitions generated by different methods, mcRigor computed per-metacell mcDiv scores that were strongly negatively correlated with the ground-truth metacell purity values (Fig. 1b and Supplementary Figs. 1d–f and 2a). By applying mcDiv thresholds

derived from mcDiv[null] values generated through double permutation, mcRigor effectively distinguished dubious metacells (purity < 1) from trustworthy metacells (purity = 1) for the MetaCell partitions, achieving an *F*-score (the harmonic mean of precision and recall for classifying between dubious and trustworthy metacells) of 0.921 (Fig. 1b). For the metacell partitions generated by SEACells, SuperCell, and MetaQ, mcRigor demonstrated similar effectiveness in detecting dubious metacells (Supplementary Fig. 1d–f), highlighting its applicability across various metacell partitioning methods.

Next, we tested mcRigor on the original bone marrow mononuclear cell CITE-seq dataset (the `bmcite` dataset from the R package `SeuratData`). As with the semi-synthetic data, we applied MetaCell, SEACells, and SuperCell, each at varying granularity levels, to this CITE-seq dataset. To estimate the purity of metacells in this real dataset, we considered fine-grained cell annotations, where T cells were further divided into subtypes including CD4 Memory, CD4 Naive, CD8 Effector, CD8 Memory, CD8 Naive, gdT, MAIT, and Treg; B cells were divided into Memory B, Naive B, and Plasmablast; progenitor cells were divided into GMP, HSC, and LMPP; Mono/DC were divided into CD14 Mono, CD16 Mono, cDC2, and pDC. From the 455 metacells generated by MetaCell with granularity level $\gamma = 30$, mcRigor detected 12 dubious metacells. For visualization, we projected the metacells onto the single-cell t-SNE embedding optimized by scDEED[42]. Interestingly, the dubious metacells either appeared as small clusters or lay at spurious positions (Fig. 1c). For example, the centroid of metacell mc452 fell within the cluster of LMPP and CD4 cells, while it mainly consisted of non-naive CD8 cells. This indicated that Metacell mc452 consisted of heterogeneous cells that were widely dispersed in the t-SNE embedding. The cell-by-gene expression matrix of mc452 supported this observation, revealing that the cells within it can be further divided into at least two clusters based on notable differences in gene expression (bottom right of Fig. 1c). In contrast, trustworthy metacells such as mc86 exhibited a high level of internal homogeneity and near-zero gene-gene correlations (bottom left of Fig. 1c). Other dubious metacells were mostly aggregates of Plasmablasts and HSCs, which are immature cells characterized by frequent self-renewal and multilineage differentiation. These results affirm that mcRigor effectively identifies dubious metacells that warrant further investigation.

## mcRigor's trustworthy metacells reveal cell-cycle phases within cell lines

As a middle ground between the single-cell and pseudobulk approaches, the metacell approach offers the advantage of revealing intra-cell-type heterogeneity, capturing biological variation among cells within the same cell type, including differences among cell subtypes or states. However, this heterogeneity can be obscured by the presence of dubious metacells, which may erroneously aggregate cells with different biological states, making these states indistinguishable. Therefore, to reveal intra-cell type heterogeneity accurately, it is essential to identify and exclude dubious metacells.

We evaluated mcRigor's ability to detect dubious metacells by testing its effectiveness in revealing cell cycle phases across five cell lines (A549, H2228, HCC827, HEK293T, and Jurkat) using scRNA-seq data. This evaluation is based on the rationale that each cell line represents a single cell type, with a major source of intra-cell-type heterogeneity arising from differences in cell-cycle phases (G1, S, and G2M). We applied MetaCell and SEACells with $\gamma = 20$ to each of the five cell lines separately to generate metacell partitions. To annotate single cells and metacells with cell-cycle phases, we used canonical gene markers to assign each single cell to a specific cell-cycle phase[43]. Each metacell was then assigned a cell-cycle phase based on the majority phase of its constituent single cells, and the phase purity of each metacell was calculated as the highest fraction of single cells belonging to the same phase. Finally, we applied mcRigor to identify dubious metacells within each metacell partition.

We observed that mcRigor successfully distinguished between metacells with high and low phase purity. Specifically, for both metacell methods and across all five cell lines, the trustworthy metacells identified by mcRigor exhibited significantly higher phase purity than the dubious metacells (mean *p*-value = 0.03740 for one-sided Wilcoxon rank-sum tests across 10 comparisons (2 methods × 5 cell lines), Supplementary Fig. 3). Furthermore, in the trustworthy metacells, cell-cycle marker genes exhibited less sparse expression within their corresponding phases and displayed a more distinct expression difference between the corresponding phase and the other two phases (Fig. 1d). For instance, the G2M phase marker gene *HMGB2* exhibited a fold change in the signal-to-noise (SNR) ratio (calculated as the mean expression divided by the standard deviation) of 5.773 in G2M relative to G1 and S when using trustworthy metacells. In comparison, the fold change in SNR ratio was only 2.089 and 2.588 when using single cells and all metacells, respectively (Fig. 1d). Similar patterns were observed for other cell-cycle marker genes (Fig. 1d and Supplementary Fig. 3b), underscoring mcRigor's ability to reveal intra-cell-type heterogeneity.

## mcRigor uncovers differential gene co-expression between healthy and COVID-19 patients

Gene-gene co-expression analysis is widely performed on scRNA-seq data to quantify the relatedness between genes, which is typically measured by the pairwise correlation of gene expression levels across cells of each cell type. However, technical noise, particularly sparsity in single-cell sequencing data, impedes the inference of gene correlations, biasing correlation estimates to an unpredictable degree[44,45]. Metacell partitioning offers a solution to better estimate gene correlations by reducing technical noise while retaining intra-cell-type heterogeneity[20]. Nonetheless, this approach achieves its intended effectiveness only when the metacells are trustworthy, which means a metacell only includes cells from the same biological state. As we prove in the "Methods" section, including dubious metacells in correlation estimation often leads to spurious co-expression findings. It is therefore crucial to apply mcRigor to eliminate dubious metacells from any downstream analysis, ensuring the reliability of discovering co-expressed gene pairs.

We applied mcRigor to an scRNA-seq dataset from human PBMCs of seven hospitalized COVID-19 patients and six healthy controls[46]. In this analysis, mcRigor successfully rectified co-expression estimates and uncovered gene modules differentially co-expressed between the COVID-19 and healthy cohorts. Given that co-expression patterns are often specific to cell type and condition[20,44], we applied SuperCell with $\gamma = 20$ separately to the 3028 B cells from COVID-19 samples and the 1994 B cells from control samples. From the two metacell partitions generated by SuperCell (one per condition), mcRigor detected 22 dubious metacells among the 152 metacells identified in the COVID-19 group and 26 dubious metacells out of the 99 metacells in the control group.

We found that excluding dubious metacells prior to correlation estimates removed biases and highlighted differentially co-expressed gene modules that may play a role in the COVID-19 disease mechanism. For example, correlation estimation using only trustworthy metacells revealed three co-expression gene modules enriched in the COVID-19 cohort (Fig. 1e), representing differential biological functions of B cells, including the antigen processing via MHC Class II gene module (*p*-value = 3.7e − 31 for one-sided Wilcoxon rank-sum test), the adaptive immune response gene module (*p*-value = 7.6e − 19), and the response to interferon-alpha gene module (*p*-value = 0.00328). These enrichment signals were notably strengthened compared to those in the raw single-cell data, where the *p*-values for these three gene modules are 2.0e − 13, 0.00043, and 0.08441, respectively. We also observed that these enrichment findings are consistent with those reported by the CS-CORE method[44].

In contrast, correlation estimation that included dubious metacells (using all metacells) yielded the counterintuitive result that the adaptive immune response gene module was not enriched in COVID-19 patients ($p$-value = 0.54632). We found that this misleading result was caused by an artifact: the strong co-expression of the adaptive immune response gene module under the healthy condition. Notably, this artifact was absent at single-cell resolution and was induced solely by the presence of dubious metacells (Fig. 1e). Applying mcRigor to the partitions generated by SEACells, MetaCell, and MetaCell2 similarly removed the artifact correlations introduced by dubious metacells and enhanced enrichment signals (Supplementary Fig. 4), demonstrating the applicability and robustness of mcRigor across various metacell methods.

## mcRigor empowers and rectifies gene regulatory inference on single-cell multiome data

Single-cell multiome data or integrated scATAC-seq and scRNA-seq data[47–49], which provide both gene expression and chromatin accessibility modalities in the same cells (either through direct measurement or computational integration), offer a lens to study the association between these two modalities. This approach reveals relationships between genes and regulatory elements (e.g., enhancers) with finer resolution, such as cell-type specificity, that is unattainable with bulk multi-omics data. However, associating genes with their regulatory elements can be challenging due to the high sparsity and noise in single-cell sequencing data, especially in the chromatin accessibility modality, so the reliability of inferred regulatory associations becomes questionable. Hence, aggregating homogeneous single cells into metacells has been implemented to help reduce data sparsity and improve gene regulatory inference[33,50].

Reference [33] demonstrated the effectiveness of metacell partitioning for empowering gene regulatory inference by applying their metacell partitioning method, SEACells, to a single-cell multiome dataset, which consists of 6881 hematopoietic stem and progenitor cells (HSPCs) from healthy bone marrow sorted for the pan-HSPC marker CD34. Compared to single-cell-based analysis, using metacells (the so-called "SEACells" generated by the SEACells method) significantly reduced data sparsity in both modalities, particularly for chromatin accessibility, and revealed gene-peak associations that were obscured at the single-cell resolution (where peaks represent open chromatin regions identified from the chromatin accessibility data).

On top of this analysis, we found that applying mcRigor to filter out dubious metacells improved gene regulatory inference in two key ways: (1) enhancing the identification of reliable gene-peak associations and (2) removing associations that were likely spurious. From the metacell partition generated by SEACells in the original study[33], mcRigor identified 7 dubious metacells out of the 85 metacells (Supplementary Fig. 5a). The effectiveness of mcRigor in improvement (1) is supported by the observation that gene-peak pairs consistently identified (adjusted $p$-value < 0.05 by the LinkPeaks function from the Signac R package, version 1.13.0, using both all metacells and trustworthy metacells) displayed significantly higher gene-peak association scores after dubious metacells were excluded (top left of Fig. 1f). For instance, the association score between the key erythroid lineage regulator *TAL1* and its most correlated peak increased from 0.8266 to 0.8703 when dubious metacells were removed. Similarly, the association score for another crucial erythroid factor, *GATA2*, with its highest correlated peak, increased from 0.6904 to 0.7606 (bottom left of Fig. 1f).

Furthermore, we demonstrated that mcRigor could recover gene-peak associations supported by the literature while filtering out unlikely associations (improvements (1) and (2)). By using the trustworthy metacells identified by mcRigor, we identified 5551 highly correlated gene-peak pairs, compared to the 5536 pairs identified using all metacells. Although the increase in the number of gene-peak pairs was

small, mcRigor refined the associations by removing weakly supported pairs and adding those with stronger data support. For instance, although using all metacells and using trustworthy metacells both identified 16 highly correlated peaks (HCPs) for the gene *GATA2*, the specific peaks identified differed between the two approaches. When dubious metacells were excluded, an additional peak was identified, while one previously identified peak was removed. The newly identified peak, *chr3-128532902-128533402*, which overlaps with *LOC117038772* (*chr3-128532862-128533362*), is an enhancer for *GATA2*, as supported by previous reports[51,52]. The peak's correlation with *GATA2* expression was also evident at the single-cell level (bottom right of Fig. 1f). In contrast, the peak not detected when using trustworthy metacells, *chr3-128409363-128409863*, showed a minimal correlation with *GATA2* expression at single-cell resolution, and its accessibility was extremely low across all cell types, providing insufficient evidence for association (Fig. 1f and Supplementary Fig. 5b). These findings confirm that mcRigor enhances regulatory analysis and, importantly, helps pinpoint likely false positives.

Intuitively, using a low granularity level may reduce the number of dubious metacells and yield reliable results even without mcRigor. However, this strategy often leaves sparsity unresolved, compromising statistical power in downstream analyses. To illustrate this, we compared results from a fine-grained metacell partition (SEACells with $\gamma = 5$, without mcRigor) to those from a coarse-grained partition (SEACells with $\gamma = 90$) followed by mcRigor filtering (Supplementary Fig. 6). We observed clear advantages with the SEACells ($\gamma = 90$) + mcRigor partition, which provided greater statistical power and identified more biologically supported enhancer-gene associations. For example, the enhancer *LOC117038771*, previously reported to regulate *GATA2*[51], was detected only with the trustworthy metacells from SEACells ($\gamma = 90$) + mcRigor, but not with SEACells ($\gamma = 5$) (Supplementary Fig. 6a). Similarly, for *TAL1*, multiple HCPs were identified using SEACells ($\gamma = 90$) + mcRigor, but none were detected using SEACells ($\gamma = 5$) or single-cell data, where sparsity limited detection power (Supplementary Fig. 6b). These findings demonstrate that simply lowering $\gamma$ is insufficient and that mcRigor is essential for improving statistical power while maintaining the reliability of metacell-based regulatory analyses.

## Assessment of mcRigor's ability to optimize metacell partitioning

A key hyperparameter in metacell partitioning is the granularity level $\gamma$, defined as the ratio of the number of single cells to the number of metacells (i.e., the average number of single cells per metacell). This hyperparameter controls the extent of size reduction from single cells to metacells, with a larger $\gamma$ indicating a greater reduction[34]. If $\gamma$ is too large, heterogeneous single cells may be aggregated into metacells, distorting downstream analyses. Conversely, if $\gamma$ is too small, the metacells closely resemble single cells, failing to address the sparsity issue. Therefore, optimizing metacell partitioning is fundamentally about balancing signal distortion and data sparsity.

To quantify this tradeoff and guide hyperparameter selection, mcRigor introduces two metrics: DubRate and ZeroRate, which measure the level of distortion introduced and the remaining level of sparsity, respectively (Fig. 2a). Intuitively, as $\gamma$ increases, DubRate rises while ZeroRate falls. mcRigor then calculates an evaluation score, termed Score, for each $\gamma$, defined as one minus a weighted sum of DubRate and ZeroRate (with the default weight set as 0.5), where a higher Score indicates a better metacell partition. The $\gamma$ value that maximizes this score is considered to balance the preservation of biological signals and the improvement of data sparsity.

We tested the effectiveness of Score as an optimization criterion on the semi-synthetic dataset used previously and described in detail in the "Methods" section, with the true granularity level as $\gamma^* = 50$. MetaCell was applied as the metacell partitioning method, with $\gamma$

values ranging from 2 to 100, generating one metacell partition per $\gamma$ value. mcRigor was then used to evaluate the DubRate, calculate the ZeroRate, and determine the final Score for each partition. We observed that DubRate exhibited an elbow point at the true granularity level, and the highest Score was achieved precisely at $\gamma = \gamma^*$ (Fig. 2a). Additionally, the optimized metacell partition closely matched the ground-truth metacells defined in the simulation, containing only four dubious metacells (Fig. 2a).

We similarly applied SEACells to identify metacell partitions and used mcRigor for granularity level optimization. In this case, DubRate displayed a less pronounced elbow around $\gamma = 50$, and mcRigor's optimized granularity level was 42, also close to the true $\gamma^*$ (Supplementary Fig. 7). However, for SuperCell and MetaCell2, the optimal $\gamma$'s selected by mcRigor ($\gamma = 4$ for both) are far away from $\gamma^*$. This is because most metacells built by these two methods were dubious, resulting in high DubRate values even at small granularity levels (Supplementary Fig. 7).

Since mcRigor's optimization criterion, Score (ranging from 0 to 1), is comparable across metacell partitions regardless of the partitioning method used, mcRigor can simultaneously determine the optimal hyperparameter $\gamma$ for each method and guide method selection for a specific single-cell dataset. For the semi-synthetic dataset, the maximal Score (i.e., the Score at the optimal $\gamma$ for each method) attained by MetaCell, SEACells, SuperCell, and MetaCell2 were 0.692, 0.642, 0.537, and 0.528, respectively (Supplementary Fig. 7). Thus, the optimal configuration selected by mcRigor is MetaCell with $\gamma = 50$, which best matched the ground truth. The higher maximal Score values for MetaCell and SEACells compared to SuperCell and MetaCell2 imply that MetaCell and SEACells produced better metacell partitions, with fewer dubious metacells and a higher reduction of sparsity. This finding is consistent with our observations on real single-cell data, where mcRigor often recommends MetaCell and SEACells over SuperCell and MetaCell2. Notably, mcRigor is task-agnostic and does not rely on prior knowledge, enabling its application to metacell-based data analysis and making it an effective and unbiased benchmarking tool for metacell partitioning methods.

## mcRigor optimizes metacell partitioning to distinguish biological and non-biological zeros

Single-cell data contain two types of zeros: biological zeros, which indicate absent or extremely low gene expression, and non-biological zeros, which result from gene expression being missed during the sequencing process[8]. Ideally, biological zeros should be retained as they provide valuable insights into the cells' biological states, while non-biological zeros should be filtered out to enhance data quality. Metacell partitioning offers a potential solution to this sparsity issue by averaging the expression profiles of multiple cells assumed to share the same biological state, thereby reducing the number of zeros. However, current metacell partitioning methods provide no guarantee to achieve two critical objectives: (1) preserving biological zeros and (2) removing non-biological zeros. This challenge arises due to the inherent trade-off between these two objectives—when the granularity level increases (i.e., more cells are merged into a metacell), the chance of removing non-biological zeros improves, but the risk of inadvertently eliminating biological zeros also rises.

Despite this challenge, mcRigor helps to balance the two competing objectives, which are about distinguishing biological zeros from non-biological zeros, by optimizing the granularity level for a metacell partitioning method. The optimization approach of mcRigor essentially minimizes the formation of dubious metacells (addressing objective (1)) while maximizing the removal of non-biological zeros (addressing objective (2)). Notably, minimizing the occurrence of dubious metacells is crucial for objective (1), as a dubious metacell contains cells from different states, where a gene may be expressed in

one state but not in another. Averaging such mixed states could inadvertently eliminate biological zeros.

We evaluated whether mcRigor can effectively distinguish biological zeros from non-biological ones using an scRNA-seq Drop-seq dataset paired with single-molecule RNA fluorescence in situ hybridization (smRNA FISH) data for 16 genes from a melanoma cell line[53]. The rationale for this approach is that smRNA FISH is widely considered the gold standard for single-cell gene expression measurement[53,54], making it reasonable to assume that all zeros in smRNA FISH data are biological zeros. We applied MetaCell, SEACells, and SuperCell to the scRNA-seq dataset, varying the granularity level ($\gamma = 2, ..., 100$), and calculated the proportion of zeros for the 16 genes in the resulting metacell-by-gene data matrix for each metacell partition. We then used mcRigor to calculate a Score for each partition and identify the optimal $\gamma$ that yielded the best Score for each method (Supplementary Fig. 8a, b). Notably, the $\gamma$ value selected by mcRigor for each metacell partitioning method resulted in a proportion of zeros that closely matched the proportion of zeros in the smRNA FISH data (Fig. 2b and Supplementary Fig. 8c). Compared to single-cell data, the expression distributions of each gene derived from the optimal metacell partitions (one per metacell method) closely approximated the corresponding distribution observed in the smRNA FISH data (Supplementary Fig. 8d). These results confirm that mcRigor can help distinguish biological zeros from non-biological zeros by optimizing metacell partitioning.

## mcRigor optimizes metacell partitioning for DGE analysis

DGE analysis on scRNA-seq data often suffers from data sparsity, leading to reduced statistical power. Aggregating single cells into metacells is one strategy to mitigate the sparsity, but the reliability of metacells is crucial. To evaluate whether mcRigor improves the reliability of metacell-based DGE analysis, we faced the challenge of the absence of ground-truth DE genes in scRNA-seq data. Therefore, we adopted an indirect validation approach: comparing the consistency of scRNA-seq DGE results with those from paired bulk RNA-seq data[10]. We applied this strategy to validate mcRigor's effectiveness, assessing whether the optimized metacell partition improved the consistency of DGE results with paired bulk RNA-seq data using a dataset obtained from both bulk and scRNA-seq experiments on human embryonic stem cells (ESC) and definitive endoderm cells (DEC)[55].

We applied MetaCell, SuperCell, and SEACells with varying granularity levels ($\gamma = 2, ..., 100$) to generate metacell partitions from the scRNA-seq data. mcRigor then calculated the *Score* value for each partition and optimized $\gamma$ for each method: $\gamma = 6$ for MetaCell, $\gamma = 4$ for SuperCell, and $\gamma = 13$ for SEACells. Among the three methods, SEACells with $\gamma = 13$ achieved the highest *Score*, making its metacell partition the optimal choice (Supplementary Fig. 9). As the first validation step, we used the DESeq2 method[56] to identify DE genes from the bulk RNA-seq data and assessed the concordance between the bulk and metacell profiles based on the expression of the top 200 DE genes. The optimal metacell partition showed the strongest concordance with the bulk data (Pearson correlation $\rho = 0.800$) (top panel of Fig. 2c). Concordance here is defined as the Pearson correlation between two expression vectors, each representing the top 200 DE genes in the bulk data (average of bulk samples) or the metacell partition (average of metacells).

Next, we evaluated mcRigor's ability to identify the metacell partition that could best recover the bulk DE genes from the paired scRNA-seq data. We applied DESeq2 to the scRNA-seq data given each metacell partition and recorded the DE genes identified at the false discovery rate (FDR) threshold of 0.05. Treating the bulk DE genes obtained from bulk data (also using a 0.05 FDR threshold) as the standard, we then calculated an *F*-score (the harmonic mean of precision and recall) for the metacell DE genes for each metacell partition. The metacell configuration selected by mcRigor (SEACells with $\gamma = 13$, Supplementary Fig. 9) produced the highest *F*-score (0.400), which is almost twice that of the *F*-score (0.204) for the single-cell DE genes

without using metacells (bottom panel of Fig. 2c). These findings indicate that the DE genes identified using the optimal metacell partition had the best agreement with the bulk DE genes, thereby indirectly affirming mcRigor's optimization of metacell partitioning, including the selection of the metacell method and its hyperparameter value.

## mcRigor's optimized metacell partition better reveals temporal immune cell trajectories

In oncogenesis, immune cells infiltrating tumors can be co-opted by the immunosuppressive tumor microenvironment (TME), transitioning from a cytotoxic to a dysfunctional state with reduced antitumor activity. To explore the transcriptomic dynamics of immune cells over time, a new technology, Zman-seq, was developed to collect data from 2431 intratumoral T and NK cells in mouse glioblastoma, along with time stamps documenting each cell's exposure duration within the tumor[30]. In the original study, the 2431 single cells were aggregated into 37 metacells using the MetaCell method with $\gamma = 66$, and temporal trajectories were tracked among these metacells, which were expected to represent distinct transcriptional states.

Specifically, the original study calculated a continuous tumor exposure time (cTET) value for each metacell. This was defined as one minus the normalized area under the curve (AUC) of the cumulative distribution function (CDF) for the frequencies of single cells within the metacell at three time points: 12 h, 24 h, and 36 h. In other words, the CDF represents a three-value discrete distribution (if all single cells in a metacell are from the 12 h time point, the AUC of the CDF is 1, and the cTET is 0). The cTET values were then used as continuous temporal labels for the metacells, where larger values indicated longer tumor exposure times. Accordingly, the following transitional stages of NK cells were expected to show increasing cTET values: NK chemotactic, NK cytotoxic, NK intermediate, and finally NK dysfunctional cells. However, in the original study, the cTET values of the metacells did not clearly distinguish these stages (top right of Fig. 2d), suggesting that some metacells might be dubious (containing a mix of NK cells of different biological states).

We applied mcRigor to optimize the metacell method-hyperparameter configuration and found that SEACells with $\gamma = 43$ provided the optimal metacell partition (Supplementary Fig. 10a, b). Specifically, compared to the original metacell partition (with *DubRate* = 0.222), the optimized partition had a significantly lower *DubRate* of 0.056. With this optimal metacell partition, the cTET values more effectively distinguished the four transitional stages, as indicated by a greater cTET difference between the earliest- and latest-stage metacells (0.718 vs 0.538 in the original study). Additionally, the cTET values showed a stronger correlation with the transitional stages of the metacells (Spearman's rank correlation = 0.967 vs 0.954 in the original study) (top panel of Fig. 2d). See the "Methods" section for further details on this data analysis.

In the original study, subsequent analysis involved ordering metacells based on their cTET values to identify genes most correlated with tumor exposure time. To evaluate the reliability of the metacell partition optimized by mcRigor, we identified 75 DE genes (FDR = 0.05) between the 12 h and 36 h time points using the original single-cell data and assessed whether these DE genes could be successfully recovered at the metacell level. The rationale is that while single-cell DE analysis often lacks power due to data sparsity, the identified single-cell DE genes should still be detectable from metacells, which are expected to enhance the power of DE analysis. Our results showed that these single-cell DE genes exhibited higher correlations with tumor exposure time (mean correlation 0.633 vs 0.570 in the original study) and lower *p*-values in DE analysis between 12h and 36h (mean adjusted *p*-value 0.035 vs 0.118 in the original study) when using the metacells from the optimal partition compared to the original partition (Supplementary Fig. 10c). Furthermore, the DE *p*-values of these genes at

the metacell level were more consistent with the single-cell *p*-values (Spearman's correlation $\rho = 0.517$ vs 0.452 in the original study). In total, the optimal metacell partition, compared to the original partition, helped identify more genes that are significantly correlated with tumor exposure time (260 vs 135 in the original study). These findings indicate that by reducing dubious metacells, mcRigor helps preserve biological signals in single-cell data, improving the reliability of downstream analyses, and gaining more power in uncovering important genes associated with time.

Additionally, mcRigor's optimized metacell partition corrected the questionable gene temporal patterns inferred from the metacells in the original study (bottom panel of Fig. 2d). For instance, *Lag3*, a gene associated with the suppression of antitumor functions[57,58] and known to act as a receptor for the ligand LSECtin[59], blunting tumor-specific immune responses, was incorrectly inferred to follow a pattern of initial downregulation followed by upregulation in the original study. In contrast, the optimized metacell partition identified a more biologically plausible pattern, showing *Lag3* as steadily upregulated over time, consistent with single-cell observations and previous studies that found *Lag3* increasingly expressed in NK cells as they progress towards a terminal intratumoral state[60]. Similarly, the optimized metacell partition corrected the temporal pattern for *Clspn*, aligning with both single-cell results and the known biology of the gene's human ortholog (gene *CLSPN*), which is typically upregulated in the immune microenvironment of most cancer types[61,62]. This was in contrast to the original study, where *Clspn* was incorrectly inferred to be downregulated over time. These findings demonstrate the necessity of using mcRigor to generate reliable metacell partitions, ensuring the accuracy of temporal gene expression analysis.

It is also worth noting that despite the finer granularity ($\gamma = 43$ in the optimized partition vs $\gamma = 66$ in the original partition), the optimized partition still effectively resolved data sparsity and enhanced biological signals. Specifically, the optimized partition successfully recovered all the important temporally dynamic genes identified in the original study (Supplementary Fig. 10d), including the upregulated module (*Pmepa1*, *Car2*, *Ctla2a*, *Xcl1*, *Itga1*, *Gzmc*, etc.) and the downregulated module (*S1pr5*, *Cx3cr1*, *Klrg1*, *Cma1*, *Ccl3*, *Gzmb*, etc.).

To explicitly distinguish the contribution of mcRigor from that of the underlying metacell methods themselves, we further conducted direct comparisons between the results obtained from each method with and without mcRigor−comparing MetaCell + mcRigor with MetaCell alone (as used in the original study), and SEACells + mcRigor (SEACells being the overall best-performing method selected by mcRigor) with SEACells alone (Supplementary Fig. 11). In both comparisons, incorporating mcRigor led to noticeable improvements. For instance, the MetaCell + mcRigor partition showed a greater cTET difference between the earliest- and latest-stage metacells (0.600) compared to MetaCell alone (0.538), as well as a stronger correlation between cTET values and the transitional stages of metacells (Spearman's rank correlation $\rho = 0.961$ vs 0.954, Supplementary Fig. 11a). Furthermore, adding mcRigor corrected questionable temporal expression patterns initially inferred by MetaCell alone, resulting in biologically consistent upregulation trends for genes including *Lag3*, *Tubb5*, *Tk1*, and *Clspn* (Supplementary Fig. 11b). Consistent with these findings, the single-cell DE genes exhibited stronger correlations with tumor exposure time and lower adjusted *p*-values at the metacell level when using MetaCell + mcRigor compared to MetaCell alone (Supplementary Fig. 11c). Similar improvements were observed when comparing SEACells + mcRigor to SEACells alone (Supplementary Fig. 11), confirming that the enhanced downstream analysis results− reflected by greater cTET distinctions and more biologically plausible gene temporal expression patterns−are indeed attributable to the use of mcRigor.

These results further underscore mcRigor's ability to identify the most suitable metacell method for a given dataset−not only by

optimizing the granularity level within each method, but also by selecting the most appropriate method. In this case, mcRigor selected SEACells over MetaCell (Supplementary Fig. 11), and accordingly, SEACells combined with mcRigor outperformed MetaCell alone, as our results showed.

## mcRigor's optimized metacell partition improves data integration and better captures T cell response dynamics

Beyond improving analysis at the individual-sample level, metacells represent structured and denoised analysis units that can potentially facilitate integration across large, cohort-level single-cell datasets. This strategy—integrating over metacells rather than single cells—is particularly valuable for large-scale datasets generated by consortia, which often exhibit substantial batch effects and sparsity, since at the single-cell level, biological differences can be difficult to distinguish from technical noise, rendering integration results unreliable. However, the effectiveness of metacell-based integration critically depends on the choice of metacell granularity: overly coarse partitions may obscure biologically meaningful variation even before integration, while overly fine ones may fail to adequately reduce technical noise. By optimizing the granularity level, mcRigor enables the construction of metacells that are best suited for biologically faithful integration and downstream analyses.

To evaluate the effectiveness of mcRigor in improving integration analysis, we reanalyzed a subset of the scRNA-seq dataset used in the SEACells study[33], comprising 96,466 PBMCs from 10 healthy donors and 10 patients with COVID-19. We focused on this data subset rather than the full dataset because, in large datasets, the impact of poor-quality metacells may be negligible. In contrast, when working with fewer cells, high-quality metacell partitioning becomes more critical, as a small number of dubious metacells may have a large distortion effect on biological signals if the total number of metacells is small. We compared SEACells metacells constructed using the default granularity level ($\gamma_{org} = 75$) with those optimized by mcRigor ($\gamma_{opt} = 49$), hereafter referred to as mcRigor metacells, and performed Harmony[63] integration using the resulting metacell expression profiles. In the SEACells study, the authors demonstrated that CD4 T cells from three different collection sites exhibit meaningful biological differences, and that such variation should be preserved, not eliminated, during integration[33]. Indeed, in the subset we analyzed, SEACells CD4T metacells constructed at the default granularity exhibited reduced similarities across different sites (mLISI, the mean Local Inverse Simpson's Index that measures batch similarities, reduced from 1.528 for single cells to 1.474 for metacells based on collection site) (Supplementary Fig. 12, bottom middle). However, the distinction between CD4 and CD8 T metacells became less pronounced after integration (mLISI increased from 1.063 for single cells to 1.152 for metacells based on cell type) (Supplementary Fig. 12, top middle). In contrast, the mcRigor metacells, compared with the unrefined SEACells metacells, better enhanced both site-specific differences (mLISI reduced from 1.528 for single cells to 1.344 for metacells based on collection site) and maintained the separation between CD4 and CD8 T cells (mLISI reduced from 1.063 for single cells to 1.028 for metacells based on cell type) (Supplementary Fig. 12, left). This suggests that data-driven granularity optimization, rather than a fixed heuristic granularity level, can be essential for robust biological signal recovery during integration.

We next investigated whether mcRigor metacells could more effectively reveal T cell response dynamics in COVID-19 using the integrated data. Following the analysis in the SEACells study[33], we further aggregated each set of metacells (SEACells metacells and mcRigor metacells) into second-level meta2cells, each consisting of ten metacells, by reapplying SEACells to their Harmony-corrected low-dimensional embeddings. From each meta2cell set, we then selected three representative meta2cells corresponding to early, middle, and late stages after COVID-19 onset (Supplementary Fig. 13a, b). Both

SEACells and mcRigor meta2cells captured temporal shifts in immune gene expression, including type I interferon-stimulated genes (IRF7, IRF9, ISG15, and IFITM1), inflammation-regulating genes (CCR10, FOXP3, and IL2RA), and hallmark $T_{17}$-related genes indicative of a transition toward type III inflammation (RORC and CCR6). Notably, mcRigor meta2cells exhibited more distinct temporal expression trajectories, especially for genes such as IFITM1, FOXP3, and CCR6 (Supplementary Fig. 13a). These results suggest that mcRigor's data-driven granularity optimization helps reveal the dynamic immune responses over the course of disease progression.

To further investigate the temporal dynamics of CD4 T cell responses, we extended our analysis to construct trajectories using all CD4 meta2cells, beyond the three representative ones in each set (SEACells and mcRigor meta2cells). Following a trajectory construction approach from a glioblastoma immune profiling study[30], we first clustered each set of meta2cells based on their Harmony[63] batch-corrected low-dimensional embeddings, varying the number of clusters $k \in [3, ..., K]$, where $K$ is the total number of CD4 meta2cells. For each $k$, we constructed a trajectory by ordering the meta2cell clusters based on their average time since disease onset, and then averaged the resulting $(K - 2)$ trajectories to generate a final temporal trajectory. Along this trajectory, we computed smoothed gene expression profiles, revealing progressive activation of immune gene modules over the course of COVID-19 infection (Supplementary Fig. 13c). Notably, trajectories based on mcRigor meta2cells showed clearer temporal progression and more coherent gene module expression patterns (Supplementary Fig. 13c, left), aligning more closely with both single-cell-level observations (Supplementary Fig. 13a, right) and established biological knowledge. For example, KLRG1—a gene known to increase in CD4 T cells during adaptive immune responses[64]—exhibited a consistent upward trend in the mcRigor-based trajectory but a reversed trend in the SEACells-based one. These results demonstrate that mcRigor's granularity optimization not only enhances data integration but also improves the reconstruction of temporal immune processes.

## Discussion

mcRigor is a novel statistical method designed to enhance the rigor of metacell partitioning in single-cell data analysis, ensuring reliable downstream analyses on metacells. By evaluating a given metacell partition, mcRigor identifies dubious metacells through quantifying per-metacell heterogeneity (i.e., the presence of mixed biological states) using a feature-correlation-based statistic. This statistic is assessed against a null distribution generated through a novel double permutation mechanism. Our findings demonstrate that the dubious metacells identified by mcRigor are indeed heterogeneous. Specifically, mcRigor can detect both dubious metacells that mix distinct major cell types and those composed of closely related subtypes within the same major type, capturing both broad and subtle forms of cellular heterogeneity (Supplementary section "Performance of mcRigor under varying cellular heterogeneity"; Supplementary Fig. 14). Applications of mcRigor to real datasets show that removing dubious metacells detected by mcRigor significantly enhances downstream analyses, such as gene co-expression studies and enhancer-gene regulatory inference. This improvement is achieved by revealing biological signals otherwise obscured by data sparsity and correcting conclusions drawn from distorted signals provided by dubious metacells. Leveraging its dubious metacell detection capability, mcRigor further optimizes metacell partitioning by selecting the method-hyperparameter configuration that best balances the trade-off between reducing data sparsity (fewer, larger metacells) and preserving biological signals (more, smaller metacells). This optimized configuration has demonstrated improved performance across various tasks, including distinguishing biological from non-biological zeros, identifying DE genes, and elucidating temporal immune cell trajectories.

We anticipate that mcRigor will be a valuable computational tool for single-cell researchers, enabling them to generate reliable metacells for addressing key scientific questions with metacell-level data. However, mcRigor has some limitations and open questions that warrant further exploration. First, mcRigor detects dubious metacells by constructing null values for the per-metacell mcDiv score through double permutation, which is performed only once on the cell-by-feature matrix of each metacell. From this double-permuted matrix, a null value (mcDiv$^{null}$) is computed for that metacell, and the mcDiv threshold is determined by pooling the null values of metacells of similar sizes. Future research could explore whether performing multiple rounds of double permutations, thereby generating multiple null values for each metacell, could lead to more robust detection of dubious metacells.

Second, while mcRigor can detect dubious metacells, it does not offer a strategy for reorganizing these metacells into trustworthy metacells; users are currently limited to removing dubious metacells in downstream analysis. A more comprehensive approach could involve recursively partitioning dubious metacells to achieve internal homogeneity, ensuring that all metacells are trustworthy without losing individual cells. Developing such a recursive partitioning algorithm remains an open question for future research. Furthermore, an interesting direction would be to design a new metacell partitioning method that constructs metacells directly from single cells based on mcRigor's evaluation criteria. This method should focus on generating trustworthy metacells from the outset, thereby minimizing the need for post-generation filtering.

As a first step toward improving metacell reconstruction, we developed a straightforward extension of mcRigor, termed mcRigor two-step, as described in the Supplementary file (Supplementary section "Performance of mcRigor under varying cellular heterogeneity"). This approach first identifies more dubious metacells using a relaxed mcDiv threshold—specifically, the 85th percentile of mcDiv null values conditional on metacell size, as opposed to the default 95th percentile used in standard mcRigor—and then re-applies metacell partitioning to the constituent cells of these dubious metacells, using a granularity level optimized by mcRigor. Rather than discarding the dubious metacells, mcRigor's two-step re-partitions their constituent cells, thereby reducing information loss. We demonstrated that this extension can further improve downstream analyses, including uncovering differential gene co-expression modules (Supplementary Figs. 15 and 16) and distinguishing biological from non-biological zeros (Supplementary Fig. 17), and more importantly, enables better detection of rare cell subpopulations (Supplementary Figs. 18 and 19), whose cells are often mixed with others in dubious metacells due to their small numbers. By reorganizing these cells into smaller, trustworthy metacells, mcRigor's two-step enhances mcRigor's ability to capture cellular heterogeneity and uncover biological states with varying abundance levels. Nonetheless, mcRigor two-step remains an initial attempt at refining metacell reconstruction—a more principled redesign of metacell partitioning, building on mcRigor, will require future, more deliberate method development.

Third, mcRigor could be enhanced to better handle multi-modality data (e.g., scRNA-seq and scATAC-seq). Currently, it operates on metacell partitions derived from a single modality, requiring users or the partitioning method to select which modality to prioritize. This single-modality approach may lead to suboptimal metacell partitions when modalities contain complementary biological signals. Therefore, a key future direction is to extend mcRigor to integrate multiple modalities in a data-driven manner. Additionally, our future work will explore metacell partitioning beyond single-cell gene expression and chromatin accessibility data. For example, the spatial niche concept in spatial transcriptomics is related to, but distinct from, the metacell concept, as spatial niches often encompass multiple cell types and represent complex microenvironments rather than homogeneous cell

groupings. Defining reliable criteria for spatial niches remains an open question. Expanding mcRigor's capabilities to support additional modalities and data types would increase its utility as a comprehensive tool for addressing data sparsity in diverse genomics data analyses.

As previously mentioned, both metacell partitioning and imputation aim to address data sparsity, and we consider them alternative strategies. Metacell partitioning reduces technical zeros by aggregating similar cells into metacells, thereby averaging out noise, whereas imputation attempts to recover technical zeros at the level of individual cells. In response to the question of whether applying imputation before metacell construction might better resolve sparsity, our view is that doing so would be redundant and potentially counterproductive. Imputation has been shown to introduce biases into the data[65], which may distort the underlying gene expression landscape and lead to the formation of spurious metacell groupings—ultimately increasing the number of dubious metacells rather than reducing them. Therefore, we recommend choosing either metacell partitioning or imputation based on the goals of downstream analysis, rather than applying both in combination.

An additional note is that doublets (and multiplets) in the single-cell droplet-based sequencing data should be removed before implementing metacell partitioning and mcRigor. Here, a doublet (or multiplet) refers to two (or more) cells being encapsulated within the same droplet during sequencing, resulting in them being tagged with the same barcode and mistakenly identified as a single cell[66]. These doublets (or multiplets) are highly unlikely to originate from the same biological state, and their inclusion in metacells can introduce within-metacell heterogeneity. Therefore, we recommend applying a doublet-detection method[67] before any metacell partitioning steps. That said, dubious metacells may arise from two major sources: (1) the inclusion of internally heterogeneous doublets or multiplets and (2) suboptimal metacell partitioning that groups single cells from different biological states. Although removing doublets and multiplets before metacell partitioning can help reduce the number of dubious metacells from the first source, those arising from the second source would still persist and can be detected by mcRigor. Furthermore, the fact that doublets or multiplets can lead to dubious metacells implies a potential additional role for mcRigor as a validation or benchmarking tool for doublet removal methods. An effective doublet removal method, when paired with a reliable metacell partitioning method, should result in few or no dubious metacells. Exploring mcRigor's utility for this purpose represents an interesting direction for future research.

Regarding the trustworthiness of metacells, an intuitive belief is that larger metacells are more likely to be dubious. While we observe that metacells exhibit substantial variability in size at the same granularity level (Supplementary Figs. 5a and 10b), no clear relationship exists between metacell size and trustworthiness (as determined by mcRigor). This finding underscores that dubious metacells cannot be reliably identified based solely on size. Moreover, metacell size distributions differ significantly across cell types and conditions (Supplementary Figs. 20 and 21). Notably, smaller metacells are more frequent in unstable biological states, such as in COVID-19 PBMCs compared to healthy controls (Supplementary Fig. 20a). For another example, in the `bmcite` dataset, progenitor cells are generally grouped into smaller metacells, whereas T cells are aggregated into larger metacells (Supplementary Fig. 21a). Hence, if we have a good understanding of the relative stability of biological conditions, metacell size distributions can serve as a quick sanity check for the quality of the metacell partitioning before applying mcRigor.

## Methods

### A statistical definition of metacells

As first introduced by ref. 17, a metacell is defined as a homogeneous collection of single-cell profiles that could have been resampled from the same original cell. This means that within a metacell, the single cells

should be in the same biological state, exhibiting the same expected expression levels of features (e.g., genes or chromatin regions) at equilibrium, referred to as biological signals. Any differences in the observed counts (of sequencing reads or unique molecular identifiers) among these single cells should therefore be attributed to measurement imperfections, known as technical variations. This definition ensures that aggregating these single cells into a metacell by averaging can reduce technical variations while preserving biological signals.

We formalize this metacell definition statistically by following the observation model for single-cell sequencing data described in ref. 68. In this model, the variations in observed counts can be decomposed into two components: the biological variation, which reflects differences in biological signals across cells, and the technical variation, introduced during the measurement process, including sequencing, that obscures these biological signals. Hence, the observation model is hierarchical, consisting of an expression model, which describes the distribution of biological signals across cells (i.e., the biological variation), and a measurement model, which describes the distribution of observed counts given these biological signals (i.e., the technical variation).

Mathematically, we consider a total of $n$ cells sequenced to measure the abundance of $p$ features, such as genes or chromatin regions. Regarding the biological signals, we let $u_{ij}$ denote the number of molecules presented in cell $i$, from feature $j$, $u_{i+} = \sum_{j=1}^{p} u_{ij}$ denote the total number of molecules in cell $i$, and $\lambda_{ij} = u_{ij}/u_{i+}$ denote the relative abundance of feature $j$ in cell $i$. It is commonly assumed that $\mathbf{\Lambda} = [\mathbf{\lambda}_1, \ldots, \mathbf{\lambda}_n]^\top = [\lambda_{ij}] \in \mathbb{R}^{n \times p}$ encapsulates all the biological signals that can be estimated from sequencing data, as absolute abundance information is not preserved in sequencing. Regarding the observed counts, we write $\mathbf{Y} = [y_{ij}] \in \mathbb{Z}_{\geq 0}^{n \times p}$, where $y_{ij}$ denotes the observed count in cell $i$ of feature $j$, and $y_{i+}$ denotes the total observed count (often referred to as the cell library size) in cell $i$. First, the expression model describes the distribution of $\mathbf{\lambda}_i$, which typically depends on cell $i$'s covariates, such as the cell type. Second, given $\mathbf{\lambda}_i$, the measurement model describes the distribution of $y_{ij}$, in a form such as

$$y_{ij}|\mathbf{\lambda}_i \overset{\text{ind}}{\sim} \text{Poisson}(c_i\lambda_{ij}), \text{ which implies } (y_{i1}, \ldots, y_{ip})|\mathbf{\lambda}_i, y_{i+} \overset{\text{ind}}{\sim} \\ \text{Mult}(y_{i+}, \lambda_{i1}, \ldots, \lambda_{ip}) \quad (1)$$

where $c_i = \mathbb{E}[y_{i+}|\mathbf{\lambda}_i]$. The measurement model reflects the technical variation among cells.

Under this hierarchical observation model and following the definition by ref. 17, we introduce a statistical definition: a metacell is a group of cells that share the same $\lambda$. We will show in the "Justification for the statistical definition of metacells" section that, under this definition, aggregating cells into a metacell through averaging can reduce the technical variance in Eq. (1) without introducing bias. In contrast, if cells with different $\lambda$ are aggregated, the estimation of $\lambda$ will most likely be biased, leading to distorted inference of the biological signals. Therefore, rigorous identification of metacells is crucial to guarantee the reliability of downstream analysis. Based on our statistical definition, we term metacells that satisfy this definition as trustworthy metacells, and those that do not are referred to as dubious metacells.

### The mcRigor algorithm

**Detection of dubious metacells.** The mcRigor algorithm begins by distinguishing between trustworthy and dubious metacells given a metacell partitioning. The premise is that, assuming cells share the same library size, any pair of features should display a negligible correlation among the cells within a trustworthy metacell. This lack of correlation follows our definition of a trustworthy metacell: given the same $\mathbf{\lambda} = (\lambda_1, \ldots, \lambda_p)^\top$, the variation among cells is purely technical, as represented by the measurement model Eq. (1). In the "Justification for

the statistical definition of metacells" section, we prove that under the measurement model, features are nearly pairwise uncorrelated, with only minimal correlations induced by the constraint $\sum_{j=1}^{p} \lambda_j = 1$, and the correlations decrease as $p$ increases and become negligible when $p$ is large. Guided by this premise, mcRigor detects dubious metacells by examining the $p \times p$ feature correlation matrix of each metacell, via a statistical test for each metacell with the null hypothesis:

$$H_0 : (y_{i1}, \ldots, y_{ip})|\mathbf{\lambda}, y_{i+} \overset{\text{ind}}{\sim} \text{Mult}(y_{i+}, \lambda_1, \ldots, \lambda_p), \forall \text{ cell } i \in \text{ metacell},$$
$$(2)$$

where $\mathbf{\lambda} = (\lambda_1, \ldots, \lambda_p)^\top$ denotes the corresponding metacell feature expression levels shared by all cells within the metacell, specifying the measurement model for $p$ features jointly as a multinomial distribution with the cell library size as the total and $\mathbf{\lambda}$ as the $p$-dimensional probabilities.

Specifically, given an $n \times p$ cell-by-feature data matrix (after Seurat V5 default normalization—the "LogNormalize" approach—and Seurat V5 default feature selection that selects the top 2000 highly variable features) and a metacell partition that defines $M$ metacells, mcRigor detects dubious metacells via the following four steps (Fig. 1a):

**Step 1 (metacell divergence scores):** mcRigor computes a divergence score for each metacell separately; the divergence score is essentially a test statistic that reflects deviation from the null hypothesis $H_0$. For the $k$th metacell, $k = 1, \ldots, M$, we define its metacell size as $m_k$, the number of single cells it includes. Then mcRigor uses the original $m_k \times p$ data matrix to calculate the metacell feature correlation matrix using the Pearson correlation, denoted by $\mathbf{R}_k \in [-1, 1]^{p \times p}$. The deviation of $\mathbf{R}_k$ from the identity matrix $\mathbf{I} \in \mathbb{R}^{p \times p}$ is then calculated as the Frobenius norm $\|\mathbf{R}_k - \mathbf{I}\|_F$. To obtain the baseline deviation value under no feature correlation, mcRigor permutes the $m_k \times p$ data matrix by within-feature (column-wise) permutation, i.e., independently shuffling the values of $m_k$ single cells for each feature. This results in a within-feature permuted data matrix from which mcRigor calculates the corresponding feature correlation matrix $\widetilde{\mathbf{R}}_k$. The metacell divergence score, called "mcDiv", is then defined as

$$\text{mcDiv}_k = \frac{\|\mathbf{R}_k - \mathbf{I}\|_F}{\|\widetilde{\mathbf{R}}_k - \mathbf{I}\|_F}, \quad (3)$$

where dividing $\|\mathbf{R}_k - \mathbf{I}\|_F$ by $\|\widetilde{\mathbf{R}}_k - \mathbf{I}\|_F$ can be viewed as normalization.

**Step 2 (null divergence scores):** mcRigor constructs a null divergence score for each metacell in a data-driven manner. These null scores serve as negative controls for thresholding the divergence scores of metacells, so as to determine the trustworthiness of metacells. Specifically, for the $k$th metacell, mcRigor applies the same procedure as in Step 1 to a within-cell permuted data matrix, obtained by independently shuffling the values of $p$ features for each of the $m_k$ single cells. Through within-cell (row-wise) permutation, the resulting permuted cells retain the original cells' library sizes. Let $\mathbf{\Pi}_k$ denote the correlation matrix of this within-cell permuted data matrix. Following the same procedure as in Step 1, within-feature permutation is applied to the within-cell permutated data matrix, resulting in $\widetilde{\mathbf{\Pi}}_k$, the correlation matrix of the double-permuted data matrix (first within-cell, then within-feature permutation). The null divergence score, called "mcDiv$^{\text{null}}$," derived from the $k$th metacell, is defined as

$$\text{mcDiv}_k^{\text{null}} = \frac{\|\mathbf{\Pi}_k - \mathbf{I}\|_F}{\|\widetilde{\mathbf{\Pi}}_k - \mathbf{I}\|_F}. \quad (4)$$

The double permutation (first within-cell, then within-feature permutation) is necessary for the null score calculation, as the null hypothesis (2) is conditional on the cell library sizes, and within-cell permutation preserves these library sizes. Notably, we cannot simply compare the numerators $\|\mathbf{R}_k - \mathbf{I}\|_F$ and $\|\mathbf{\Pi}_k - \mathbf{I}\|_F$ while ignoring the denominator in

the metacell divergence score definition. This is because $\mathbf{R}_k$ and $\mathbf{\Pi}_k$ are based on different sets of features (since within-cell permutation does not preserve features) and are therefore not directly comparable. Furthermore, the denominator $\| \widetilde{\mathbf{R}}_k - \mathbf{I} \|_F$, which is derived from within-feature permutation, is not a valid null for the numerator $\| \mathbf{R}_k - \mathbf{I} \|_F$, as the cell library sizes are not preserved in the calculation of $\widetilde{\mathbf{R}}_k$. We will present a result to demonstrate that within-feature permutation is inappropriate in Note 1 (below Step 4).

From the $M$ null divergence scores, $\mathrm{mcDiv}_1^{null}, \ldots, \mathrm{mcDiv}_M^{null}$, mcRigor learns the thresholds for distinguishing between dubious and trustworthy metacells based on their mcDiv scores in the following step.

**Step 3 (divergence score thresholds)**: an intuitive approach might suggest classifying the metacells whose mcDiv scores exceed the 95% quantile of $\mathrm{mcDiv}_1^{null}, \ldots, \mathrm{mcDiv}_M^{null}$ as the dubious metacells and those below the quantile as trustworthy metacells. However, the null divergence scores defined in Eq. (4) exhibit varying distributions depending on the metacell size (Supplementary Fig. 2b). Hence, mcRigor uses metacell-size-specific mcDiv thresholds instead of a single 95% quantile value to determine the trustworthiness of a metacell. Specifically, the threshold is metacell-specific and defined as a function of the metacell size:

$$\theta(m_k) = q_{0.95}\left(\left\{ \mathrm{mcDiv}_{k'}^{null} : m_{k'} \in [m_k - h, m_k + h], \ k' = 1, \ldots, M \right\}\right), \tag{5}$$

where $q_{0.95}(\cdot)$ computes the 95% quantile and $h$ stands for the metacell size bandwidth (default $h = 10$). That is, mcRigor computes the threshold for the $k$th metacell as the 95% quantile of the null divergence scores from metacells whose sizes fall within the bandwidth of $m_k$.

**Step 4 (dubious metacell detection)**: upon completion of the previous three steps, mcRigor categorizes all $M$ metacells as either dubious or trustworthy. Specifically, for the $k$th metacell, if $\mathrm{mcDiv}_k > \theta(m_k)$, it is classified as dubious, otherwise, it is considered trustworthy.

*Note 1*: if we use the within-feature permutation instead of the double permutation to find dubious metacell detection thresholds, i.e., the $k$th metacell is considered dubious if $\| \mathbf{R}_k - \mathbf{I} \|_F > q_{0.95}(\{\| \widetilde{\mathbf{R}}_{k'} - \mathbf{I} \|_F : m_{k'} \in [m_k - h, m_k + h], \ k' = 1, \ldots, M\})$, this approach sets the threshold too low, leading to the false flagging of trustworthy metacells as dubious. This issue arises because the magnitude of $\| \mathbf{R}_k - \mathbf{I} \|_F$ may simply reflect variations in cell library sizes within the $k$th metacell. However, such variations are not preserved in the within-feature permutation, resulting in a smaller magnitude of $\| \widetilde{\mathbf{R}}_k - \mathbf{I} \|_F$ compared to $\| \mathbf{R}_k - \mathbf{I} \|_F$. In a simulation study, we demonstrate that more than 35% of ground-truth trustworthy metacells—comprising single cells of the same cell state (according to our statistical definition)—are incorrectly classified as dubious, leading to poor overall classification performance (*F*-score < 0.4) in distinguishing between trustworthy and dubious metacells (Supplementary Fig. 22). This is in sharp contrast to the performance of mcRigor based on the double permutation: of all the ground-truth trustworthy metacells, mcRigor accurately identifies over 98% as trustworthy, with less than 2% incorrectly classified as dubious (Supplementary Fig. 22). Additionally, mcRigor's *F*-score for distinguishing between dubious and trustworthy metacells consistently exceeds 0.9, highlighting its high detection accuracy.

*Note 2*: it is important to note that there is extensive literature on high-dimensional covariance testing[69–71], which might appear relevant for detecting dubious metacells by testing whether the feature covariance matrix within a metacell significantly deviates from a diagonal matrix. However, those tests are not applicable for two main reasons. First, the metacell size is too small (commonly below 100) for the test statistics to approximately follow their asymptotic null distributions.

Second, the null hypotheses in these tests do not account for cell library sizes.

**Optimization of metacell partitioning: method and hyperparameter choices.** Employing the dubious metacell detection procedure outlined in the previous subsection, mcRigor further assists users in optimizing metacell partitioning by simultaneously identifying the best-performing metacell method and the optimal hyperparameter(s) for a particular single-cell dataset. In this work, we primarily focus on optimizing the granularity level hyperparameter, $\gamma$, which means the average number of single cells per metacell and is universally required by all existing metacell partitioning methods (some methods use the hyperparameter "target metacell number" instead, which is equal to $\lfloor n/\gamma \rfloor$, with $n$ as the number of single cells). While other hyperparameters—such as the number of neighbors ($k$) in kNN graph construction and the number of principal components used—can also be optimized by mcRigor, we omit their discussion here. We choose to optimize $\gamma$, as these other parameters have been reported to have a smaller impact on metacell partitioning and are not universally required by metacell methods[34].

As discussed in the "Results" section, the optimization of metacell partitioning is, at its core, about balancing between data sparsity and signal distortion, the latter being addressed by mcRigor's first functionality: the detection of dubious metacells. Intuitively, a smaller $\gamma$ value results in fewer dubious metacells and less signal distortion, as aggregating a smaller number of cells lowers the chance of merging cells of different states. However, the minimal value of $\gamma = 1$, which retains single cells without aggregating them into metacells and therefore avoids signal distortion, is not always desirable, as it leaves the sparsity issue unresolved. Hence, an ideal metacell partitioning should aggregate as many cells as possible, while ensuring that only homogeneous cells are combined. To strike this balance, mcRigor assesses the level of sparsity and signal distortion for each metacell partition, enabling a quantitative tradeoff.

Specifically, for a given metacell method and hyperparameter, mcRigor evaluates two competing factors that respectively reflect signal distortion and data sparsity (Fig. 2a). First, mcRigor reports the proportion of single cells that constitute dubious metacells, denoted as the dubious rate: $DubRate \in [0, 1]$. Second, to account for the remaining sparsity level, mcRigor considers another metric, $ZeroRate \in [0, 1]$, which is the proportion of zeros in the $M \times p$ metacell expression matrix after single cells are aggregated into $M$ metacells. Combining $DubRate$ and $ZeroRate$, mcRigor defines an evaluation score for a metacell partition:

$$Score = 1 - w \cdot DubRate - (1 - w) \cdot ZeroRate \in [0, 1], \tag{6}$$

where $w \in (0, 1)$ is the weight assigned to $DubRate$. Our recommendation is the default $w = 0.5$, a robust choice in our simulations, but users can adjust $w$ based on their desired emphasis on the two competing factors. Based on this definition (6), a higher evaluation score indicates a better metacell partition.

The evaluation score allows for universal comparison of metacell partitions across varying hyperparameter values of different metacell methods. For a particular dataset, mcRigor identifies the method-hyperparameter configuration that maximizes the evaluation score from a candidate set of metacell methods and hyperparameter values. By default, four metacell methods are considered: SEACells, MetaCell, MetaCell2, and SuperCell, with $\gamma$ ranging from 2 to 100.

## Justification for the statistical definition of metacells
A typical way of defining the metacell-by-feature expression matrix, $\mathbf{Z} = [z_{kj}] \in \mathbb{R}^{M \times p}$, is averaging the assigned single cell profiles, namely,

defining

$$z_{kj} = \frac{1}{m_k} \sum_{i \in mc_k} \psi(y_{ij}/y_{i+}) \qquad (7)$$

where $mc_k$ denotes the index set of single cells assigned to the $k$th metacell ($|mc_k| = m_k$), and $\psi(\cdot)$ is a function specified for a particular normalization method. For ease of understanding, we consider the simple case where $\psi(x) = x$.

If the $k$th metacell satisfies our statistical definition, the single cells assigned to the $k$th metacell share the same relative expression levels $\boldsymbol{\lambda}_k = (\lambda_{k1}, \ldots, \lambda_{kp})^\top$. For cell $i$ in the $k$th metacell, we have discussed in (2) that conditional on cell library size $y_{i+}$ and relative expression level $\boldsymbol{\lambda}_k$, the observed feature counts follow a multinomial distribution:

$$H_0 : (y_{i1}, \ldots, y_{ip})|\boldsymbol{\lambda}_k, y_{i+} \overset{ind}{\sim} \text{Mult}(y_{i+}, \lambda_{k1}, \ldots, \lambda_{kp}).$$

By the property of the multinomial distribution, we have

$$\begin{aligned}
\text{E}[z_{kj}|\boldsymbol{\lambda}_k, \mathbf{y}_{k+}] &= \frac{1}{m_k} \sum_{i \in mc_k} \frac{\text{E}[y_{ij}|\boldsymbol{\lambda}_k, \mathbf{y}_{k+}]}{y_{i+}} = \lambda_{kj}, \\
\text{Var}[z_{kj}|\boldsymbol{\lambda}_k, \mathbf{y}_{k+}] &= \frac{1}{m_k^2} \sum_{i \in mc_k} \text{Var}\left[\frac{y_{ij}}{y_{i+}}\Big|\boldsymbol{\lambda}_k, \mathbf{y}_{k+}\right] < \\
\frac{1}{m_k} \cdot \max_{i \in mc_k} \text{Var}&\left[\frac{y_{ij}}{y_{i+}}\Big|\boldsymbol{\lambda}_k, \mathbf{y}_{k+}\right] < \max_{i \in mc_k} \text{Var}\left[\frac{y_{ij}}{y_{i+}}\Big|\boldsymbol{\lambda}_k, \mathbf{y}_{k+}\right],
\end{aligned} \qquad (8)$$

where $\mathbf{y}_{k+} = (y_{i+})_{i \in mc_k}$. This indicates that the obtained metacell profile provides an unbiased estimate of the true expression level with reduced variance.

Moreover, the sample covariance within the $k$th metacell between any pair of features $j$ and $\ell$ is given by

$$\widehat{\Sigma}_{k,j\ell} = \frac{1}{m_k - 1} \sum_{i \in mc_k} \frac{y_{ij}}{y_{i+}} \frac{y_{i\ell}}{y_{i+}} - \frac{1}{m_k(m_k - 1)} \sum_{i \in mc_k} \frac{y_{ij}}{y_{i+}} \sum_{i \in mc_k} \frac{y_{i\ell}}{y_{i+}}, \qquad (9)$$

whose expectation is the conditional covariance between features $j$ and $\ell$ given $\lambda_k$ and $\mathbf{y}_{k+}$:

$$\begin{aligned}
\Sigma_{k,j\ell|\boldsymbol{\lambda}_k, \mathbf{y}_{k+}} &= \text{E}[\widehat{\Sigma}_{k,j\ell}|\boldsymbol{\lambda}_k, \mathbf{y}_{k+}] \\
&= \frac{1}{m_k - 1} \sum_{i \in mc_k} \text{E}\left[\frac{y_{ij}}{y_{i+}} \frac{y_{i\ell}}{y_{i+}}\Big|\boldsymbol{\lambda}_k, \mathbf{y}_{k+}\right] - \frac{1}{m_k(m_k - 1)} \text{E}\left[\sum_{i \in mc_k} \frac{y_{ij}}{y_{i+}} \sum_{i \in mc_k} \frac{y_{i\ell}}{y_{i+}}\Big|\boldsymbol{\lambda}_k, \mathbf{y}_{k+}\right] \\
&= \frac{1}{m_k - 1} \sum_{i \in mc_k} \text{E}\left[\frac{y_{ij}}{y_{i+}}\Big|\boldsymbol{\lambda}_k, \mathbf{y}_{k+}\right] \text{E}\left[\frac{y_{i\ell}}{y_{i+}}\Big|\boldsymbol{\lambda}_k, \mathbf{y}_{k+}\right] + \frac{1}{m_k - 1} \sum_{i \in mc_k} \text{Cov}\left(\frac{y_{ij}}{y_{i+}}, \frac{y_{i\ell}}{y_{i+}}\Big|\boldsymbol{\lambda}_k, \mathbf{y}_{k+}\right) \\
&\quad - \frac{1}{m_k(m_k - 1)} \sum_{i \in mc_k} \text{E}\left[\frac{y_{ij}}{y_{i+}}\Big|\boldsymbol{\lambda}_k, \mathbf{y}_{k+}\right] \sum_{i \in mc_k} \text{E}\left[\frac{y_{i\ell}}{y_{i+}}\Big|\boldsymbol{\lambda}_k, \mathbf{y}_{k+}\right] - \frac{1}{m_k(m_k - 1)} \sum_{i \in mc_k} \\
&\quad \text{Cov}\left(\frac{y_{ij}}{y_{i+}}, \frac{y_{i\ell}}{y_{i+}}\Big|\boldsymbol{\lambda}_k, \mathbf{y}_{k+}\right) \\
&= \frac{1}{m_k - 1} m_k \lambda_{kj}\lambda_{k\ell} - \frac{1}{m_k - 1} \sum_{i \in mc_k} \frac{\lambda_{kj}\lambda_{k\ell}}{y_{i+}} - \frac{1}{m_k(m_k - 1)} m_k \lambda_{kj} m_k \lambda_{k\ell} \\
&\quad + \frac{1}{m_k(m_k - 1)} \sum_{i \in mc_k} \frac{\lambda_{kj}\lambda_{k\ell}}{y_{i+}} \\
&= -\frac{\lambda_{kj}\lambda_{k\ell}}{m_k} \sum_{i \in mc_k} \frac{1}{y_{i+}}.
\end{aligned} \qquad (10)$$

Similarly, the sample variance of feature $j$ within the $k$th metacell is given by

$$\widehat{\Sigma}_{k,jj} = \frac{1}{m_k - 1} \sum_{i \in mc_k} \left(\frac{y_{ij}}{y_{i+}}\right)^2 - \frac{1}{m_k(m_k - 1)} \left(\sum_{i \in mc_k} \frac{y_{ij}}{y_{i+}}\right)^2, \qquad (11)$$

whose expectation is the conditional variance of feature $j$ given $\lambda_k$ and $\mathbf{y}_{k+}$.

$$\Sigma_{k,jj|\boldsymbol{\lambda}_k, \mathbf{y}_{k+}} = \text{E}[\widehat{\Sigma}_{k,jj}|\boldsymbol{\lambda}_k, \mathbf{y}_{k+}] = \frac{\lambda_{kj}(1 - \lambda_{kj})}{m_k} \sum_{i \in mc_k} \frac{1}{y_{i+}}. \qquad (12)$$

Thus, correspondingly, the conditional correlation within the $k$th metacell between features $j$ and $\ell$ is

$$\mathbf{R}_{k,j\ell|\boldsymbol{\lambda}_k, \mathbf{y}_{k+}} = \frac{\Sigma_{k,j\ell|\boldsymbol{\lambda}_k, \mathbf{y}_{k+}}}{\sqrt{\Sigma_{k,jj|\boldsymbol{\lambda}_k, \mathbf{y}_{k+}} \Sigma_{k,\ell\ell|\boldsymbol{\lambda}_k, \mathbf{y}_{k+}}}} = -\sqrt{\frac{\lambda_{kj}}{1 - \lambda_{kj}}}\sqrt{\frac{\lambda_{k\ell}}{1 - \lambda_{k\ell}}}, \qquad (13)$$

whose absolute value is under $\max_{j=1,\ldots,p} \frac{\lambda_{kj}}{1-\lambda_{kj}}$, a negligible value when $p$ is sufficiently large.

**Justification for why dubious metacells bias co-expression analysis.** If some metacells appear as a mixed collection of single cells in different biological states, the aggregated expressions of those metacells are most likely biased, and inflated feature correlations may be observed within the metacells. What makes things worse is that such a distortion could be carried to the inference of feature co-expression using metacells. Let us consider a simple case for cell-type-specific co-expression analysis where the single cells are of one cell type but belong to $N$ distinct cell states of balanced sizes. Let $cs_k$ denote the single cell index set of the $k$th cell state, and $\boldsymbol{\lambda}_k^* = (\lambda_{k1}^*, \ldots, \lambda_{kp}^*)^\top$ denote the corresponding cell-state feature expression levels.

Suppose we want to estimate the covariance between features $j$ and $\ell$, $\Omega_{j\ell}$, from the metacell-by-feature expression matrix $\mathbf{Z}$. If metacells are correctly identified, i.e., $M = N$ and $mc_k = cs_k$ for $k = 1, \ldots, M$,

$$\Omega_{j\ell} = \frac{1}{M-1} \sum_{k=1}^M \lambda_{kj}^* \lambda_{k\ell}^* - \frac{1}{M(M-1)} \sum_{k=1}^M \lambda_{kj}^* \sum_{k=1}^M \lambda_{k\ell}^*, \qquad (14)$$

and the estimate is

$$\widehat{\Omega}_{j\ell} = \frac{1}{M-1} \sum_{k=1}^M z_{kj} z_{k\ell} - \frac{1}{M(M-1)} \sum_{k=1}^M z_{kj} \sum_{k=1}^M z_{k\ell}. \qquad (15)$$

This estimate is asymptotically unbiased since

$$\begin{aligned}
\text{E}[\widehat{\Omega}_{j\ell}] &= \text{E}\left[\text{E}\left[\widehat{\Omega}_{j\ell}|\boldsymbol{\lambda}_1^*, \ldots, \boldsymbol{\lambda}_M^*, \mathbf{y}_{1+}, \ldots, \mathbf{y}_{M+}\right]\right] \\
&= \text{E}\left[\frac{1}{M-1} \sum_{k=1}^M \text{E}[z_{kj}|\boldsymbol{\lambda}_k^*, \mathbf{y}_{k+}] \text{E}[z_{k\ell}|\boldsymbol{\lambda}_k^*, \mathbf{y}_{k+}] + \frac{1}{M-1} \sum_{k=1}^M \text{Cov}(z_{kj}, z_{k\ell}|\boldsymbol{\lambda}_k^*, \mathbf{y}_{k+})\right] \\
&\quad - \text{E}\left[\frac{1}{M(M-1)} \sum_{k=1}^M \text{E}[z_{kj}|\boldsymbol{\lambda}_k^*, \mathbf{y}_{k+}] \sum_{k=1}^M \text{E}[z_{k\ell}|\boldsymbol{\lambda}_k^*, \mathbf{y}_{k+}]\right. \\
&\quad \left. + \frac{1}{M(M-1)} \sum_{k=1}^M \text{Cov}(z_{kj}, z_{k\ell}|\boldsymbol{\lambda}_k^*, \mathbf{y}_{k+})\right] \\
&\overset{(*)}{=} \text{E}\left[\frac{1}{M-1} \sum_{k=1}^M \text{E}[z_{kj}|\boldsymbol{\lambda}_k^*, \mathbf{y}_{k+}] \text{E}[z_{k\ell}|\boldsymbol{\lambda}_k^*, \mathbf{y}_{k+}]\right. \\
&\quad \left. - \frac{1}{M(M-1)} \sum_{k=1}^M \text{E}[z_{kj}|\boldsymbol{\lambda}_k^*, \mathbf{y}_{k+}] \sum_{k=1}^M \text{E}[z_{k\ell}|\boldsymbol{\lambda}_k^*, \mathbf{y}_{k+}]\right] + o\left(\frac{1}{Mp^2}\right) \\
&= \text{E}\left[\frac{1}{M-1} \sum_{k=1}^M \lambda_{kj}^* \lambda_{k\ell}^* - \frac{1}{M(M-1)} \sum_{k=1}^M \lambda_{kj}^* \sum_{k=1}^M \lambda_{k\ell}^*\right] + o\left(\frac{1}{Mp^2}\right) \\
&= \Omega_{j\ell} + o\left(\frac{1}{Mp^2}\right),
\end{aligned} \qquad (16)$$

where $\mathbf{y}_{k+} = (y_{i+})_{i \in mc_k}$ and the third equality (*) holds under the assumptions $\max_{j=1,\ldots,p} \lambda_{kj} = O(1/p)$, indicating that no single feature dominates, and $M = o(m_k \cdot \min_{i \in mc_k} y_{i+})$, indicating that the number of metacells is of a lower order compared to the total number of feature

counts within a metacell. We consider these two assumptions reasonable based on empirical observations. Then we have

$$
\begin{aligned}
&\left| E\left[ \frac{1}{M-1}\sum_{k=1}^{M} \mathrm{Cov}\left(z_{kj}, z_{k\ell} \big| \boldsymbol{\lambda}_k^*, \mathbf{y}_{k+}\right) - \frac{1}{M(M-1)}\sum_{k=1}^{M} \mathrm{Cov}\left(z_{kj}, z_{k\ell} \big| \boldsymbol{\lambda}_k^*, \mathbf{y}_{k+}\right) \right] \right| \\
&= \left| E\left[ \frac{1}{M}\sum_{k=1}^{M} \mathrm{Cov}\left(z_{kj}, z_{k\ell} \big| \boldsymbol{\lambda}_k^*, \mathbf{y}_{k+}\right) \right] \right| = \left| E\left[ \frac{1}{M}\sum_{k=1}^{M} \frac{1}{m_k^2}\sum_{i\in\mathrm{mc}_k} \mathrm{Cov}\left( \frac{y_{ij}}{y_{i+}}, \frac{y_{i\ell}}{y_{i+}} \big| \boldsymbol{\lambda}_k^*, \mathbf{y}_{k+}\right) \right] \right| \\
&= \left| E\left[ -\frac{1}{M}\sum_{k=1}^{M} \frac{1}{m_k^2}\sum_{i\in\mathrm{mc}_k} \frac{\lambda_{kj}\lambda_{k\ell}}{y_{i+}} \right] \right| \le E\left[ \frac{1}{M}\sum_{k=1}^{M} \frac{1}{m_k} \cdot \frac{\max_{j=1,\dots,p} \lambda_{kj}^2}{\min_{i\in\mathrm{mc}_k} y_{i+}} \right] = o\left( \frac{1}{Mp^2} \right).
\end{aligned}
\tag{17}
$$

However, this asymptotic unbiasedness may not hold if single cells of different states are grouped into a metacell, creating a dubious metacell. For example, let us consider a scenario where the first two metacells are both equal mixtures of two distinct cell states. Let $A_1$ represent the index set of single cells belonging to the first cell state and assigned to the first metacell, and $B_1$ represent the index set of single cells belonging to the second cell state and assigned to the first metacell. Namely, we assume $\mathrm{mc}_1 = A_1 \cup B_1$ and $\mathrm{mc}_2 = \mathrm{cs}_1 \cup \mathrm{cs}_2 \backslash \mathrm{mc}_1$, where $A_1 \subset \mathrm{cs}_1$, $B_1 \subset \mathrm{cs}_2$, and $|A_1| = |B_1| = m_1/2$, $|\mathrm{cs}_1| = |\mathrm{cs}_2| = (m_1 + m_2)/2$. Then we have

$$
\begin{aligned}
z_{1j} &= \frac{1}{m_1}\sum_{i\in A_1} \frac{y_{ij}}{y_{i+}} + \frac{1}{m_1}\sum_{i\in B_1} \frac{y_{ij}}{y_{i+}}, \\
z_{2j} &= \frac{1}{m_2}\sum_{i\in \mathrm{cs}_1 \backslash A_1} \frac{y_{ij}}{y_{i+}} + \frac{1}{m_2}\sum_{i\in \mathrm{cs}_2 \backslash B_1} \frac{y_{ij}}{y_{i+}},
\end{aligned}
\tag{18}
$$

whose conditional expectations are $E\left[ z_{1j} \big| \boldsymbol{\lambda}_1^*, \boldsymbol{\lambda}_2^*, \mathbf{y}_{1+} \right] = E\left[ z_{2j} \big| \boldsymbol{\lambda}_1^*, \boldsymbol{\lambda}_2^*, \mathbf{y}_{2+} \right] = \left( \lambda_{1j}^* + \lambda_{2j}^* \right)/2$, and a similar derivation as in ref. 16 gives

$$
\begin{aligned}
E\left[ \widehat{\Omega}_{j\ell} \right] &\overset{(*)}{=} E\left[ \frac{1}{M-1}\left( 2\cdot\frac{\lambda_{1j}^* + \lambda_{2j}^*}{2}\frac{\lambda_{1\ell}^* + \lambda_{2\ell}^*}{2} + \sum_{k=3}^{M}\lambda_{kj}^*\lambda_{k\ell}^* \right) - \frac{1}{M(M-1)}\sum_{k=1}^{M}\lambda_{kj}^*\sum_{k=1}^{M}\lambda_{k\ell}^* \right] \\
&\quad + o\left( \frac{1}{Mp^2} \right) \\
&= \Omega_{j\ell} + \frac{1}{2(M-1)} E\left[ \lambda_{1j}^*\lambda_{2\ell}^* + \lambda_{2j}^*\lambda_{1\ell}^* - \lambda_{1j}^*\lambda_{1\ell}^* - \lambda_{2j}^*\lambda_{2\ell}^* \right] + o\left( \frac{1}{Mp^2} \right).
\end{aligned}
\tag{19}
$$

This indicates that the covariance estimate is biased, and the bias $\frac{1}{2(M-1)} E\left[ \lambda_{1j}^*\lambda_{2\ell}^* + \lambda_{2j}^*\lambda_{1\ell}^* - \lambda_{1j}^*\lambda_{1\ell}^* - \lambda_{2j}^*\lambda_{2\ell}^* \right]$ is non-negligible if the number of dubious metacells is in the same order of $M$ (so the bias does not go down to zero as $M$ becomes large). An extreme example is that the covariance estimate would have expectation 0 when all single cells from various cell states are indiscriminately merged into equal-sized metacells, regardless of the true covariance value.

### Semi-synthetic data generation for simulation

We generated a semi-synthetic dataset with true granularity level $\gamma^* = 50$ in a reference-based manner, making use of the scDesign3 simulator[38]. The reference data is a scRNA-seq dataset from $n = 13,408$ bone marrow mononuclear cells of one donor processed using CITE-seq, which can be accessed from the dataset `bmcite` in the R package `SeuratData`, and the top $p = 2000$ highly variable genes are considered as features. We use the variable $x_i \in \{1, 2, 3, 4, 5\}$ to denote the cell type of cell $i$, which is one-dimensional and consists of five categories, including B cells, T cells, progenitor cells, NK cells, and monocytes or dendritic cells (Mono/DC).

**Model fitting.** Specifically, we followed scDesign3[38] to model the distribution of $y_{ij}$, the count of feature $j$ in cell $i$, as a Negative Binomial

(NB) distribution with the mean parameter, $\mu_{ij}$, and the dispersion (size) parameter, $\sigma_{ij}$, conditional on the cell type variable $x_i$. For each feature $j$,

$$
\begin{cases}
y_{ij}|x_i \overset{\mathrm{ind}}{\sim} \mathrm{NB}(\mu_{ij}, \sigma_{ij}) \\
\log(\mu_{ij}) = \alpha_{j0} + f_j(x_i) + \log(y_{i+}) \\
\log(\sigma_{ij}) = \beta_{j0} + g_j(x_i)
\end{cases}
\tag{20}
$$

Since the NB distribution models both the biological and technical variations, we decompose it into the expression model and the measurement model for feature $j$:

$$
y_{ij}|x_i \overset{\mathrm{ind}}{\sim} \mathrm{NB}(\mu_{ij}, \sigma_{ij}) \iff
\begin{cases}
\theta_{ij}|x_i \overset{\mathrm{ind}}{\sim} \mathrm{Gamma}(\sigma_{ij}, 1/\sigma_{ij}), \\
y_{ij}|\theta_{ij}, x_i \overset{\mathrm{ind}}{\sim} \mathrm{Poisson}(\mu_{ij}\theta_{ij})
\end{cases},
\tag{21}
$$

where $\theta_{ij}$ denotes the biological variation around the true expression level $\mu_{ij}$ given $x_{ij}$. Note that $\mu_{ij}\theta_{ij} = c_i\lambda_{ij}$ in (1).

The distribution of $y_{ij}|x_i$ in (20) is fitted by the function `gamlss()` in the R package `gamlss` (version 5.4-22), which provides the estimated parameters $\hat{u}_{ij}$ and $\hat{\sigma}_{ij}$, which are needed for the measurement model and expression model, respectively. For convenience of notation in the following description, we denote the CDF of feature $j$'s expression model $\mathrm{Gamma}(\sigma_{ij}, 1/\sigma_{ij})$ as $F_j(\cdot|x_i)$ and, accordingly, the CDF of the fitted distribution $\mathrm{Gamma}(\hat{\sigma}_{ij}, 1/\hat{\sigma}_{ij})$ as $\hat{F}_j(\cdot|x_i)$.

With a slight modification to scDesign3, we fitted a joint expression model of the $p$ features conditional on the cell type variable, denoted by the CDF $F(\cdot|x_i) : \mathbb{R}^p \to [0, 1]$, using a Gaussian copula:

$$
F(\boldsymbol{\theta}_i|x_i) = \Phi_p\left( \Phi^{-1}\left( F_1(\theta_{i1}|x_i) \right), \cdots, \Phi^{-1}\left( F_p(\theta_{ip}|x_i) \right); \mathbf{R}(x_i) \right),
$$

where $\boldsymbol{\theta}_i = (\theta_{i1}, \dots, \theta_{ip})^\top$, $\Phi^{-1}$ denotes the inverse of the CDF of the standard Gaussian distribution, and $\Phi_p(\cdot; \mathbf{R}(x_i)) : \mathbb{R}^p \to [0, 1]$ denotes the CDF of a $p$-dimensional Gaussian distribution with a zero mean vector and a covariance matrix equal to the correlation matrix $\mathbf{R}(x_i)$. To estimate the copula, we adopted the plug-in approach, where $\hat{\mathbf{R}}(x_i)$ is calculated as the sample correlation matrix of $\{(\Phi^{-1}(\hat{F}_1(y_{i'1}/\hat{\mu}_{i'1}|x_{i'})), \dots, \Phi^{-1}(\hat{F}_p(y_{i'p}/\hat{\mu}_{i'p}|x_{i'})))^\top : x_{i'} = x_i\}$, which includes all cells of the same type as cell $i$. The estimated joint CDF is thus

$$
\hat{F}(\boldsymbol{\theta}_i|x_i) = \Phi_p\left( \Phi^{-1}\left( \hat{F}_1(\theta_{i1}|x_i) \right), \cdots, \Phi^{-1}\left( \hat{F}_p(\theta_{ip}|x_i) \right); \hat{\mathbf{R}}(x_i) \right).
\tag{22}
$$

**Sampling from the fitted model.** Next, we generated the semi-synthetic data by sampling from the fitted expression and measurement models. We started by generating $M = \lfloor n/\gamma^* \rfloor$ synthetic metacells, each representing a different cell state, by sampling without replacement from the $n$ real cells. We kept the $M$ sampled real cells' parameter estimates for the Poisson mean parameter (of the measurement model for feature $j = 1, \dots, p$) in (21) as $\{\tilde{\mu}_{kj}\}_{k=1}^{M} \subset \{\hat{\mu}_{ij}\}_{i=1}^{n}$ and their cell types as $\{\tilde{x}_k\}_{k=1}^{M} \subset \{x_i\}_{i=1}^{n}$ for the $M$ synthetic metacells, where $\tilde{x}_k \in \{1, 2, 3, 4, 5\}$. Then we independently sampled the biological variation of the $k$th synthetic metacell, $\tilde{\boldsymbol{\theta}}_k$, from the joint expression model $\hat{F}(\cdot|\tilde{x}_k)$, $k = 1, \dots, M$ (22). In detail, to sample the $k$th synthetic metacell, given its cell-type covariate $\tilde{x}_k$, we independently sampled a $p$-dimensional vector from the estimated $p$-dimensional Gaussian distribution:

$$
(v_{k1}, \dots, v_{kp})^\top \overset{\mathrm{ind}}{\sim} \Phi_p\left( \cdot; \hat{\mathbf{R}}(\tilde{x}_k) \right), \quad k = 1, \dots, M.
$$

Then, based on the fitted expression model $\hat{F}_j(\cdot|\tilde{x}_k)$ of feature $j = 1, \dots, p$, we calculated the expression level of feature $j$ in the $k$th

synthetic metacell:

$$\tilde{\theta}_{kj} = \hat{F}_j^{-1}\left(\Phi(v_{kj})\big|\tilde{x}_k\right), \quad j = 1, \ldots, p.$$

The sizes of synthetic metacells, denoted by $m_1, \ldots, m_M$, are randomly generated integers from the range 20–80 with the restriction that their mean equals to $\gamma^* = 50$. We then independently sampled synthetic single-cell feature counts from the fitted measurement model:

$$y_{i'j} \sim \text{Poisson}\left(\cdot; \omega_{i'j}\tilde{\mu}_{kj}\tilde{\theta}_{kj}\right), \quad i' \in \text{mc}_k,$$

where $\omega_{i'j}$ is the amplification factor for the synthetic cell $i'$ within the $k$th synthetic metacell, allowing for varying cell library sizes within the metacell, and $\text{mc}_k$ denotes the index set of single cells assigned to the $k$th metacell ($|\text{mc}_k| = m_k$). To ensure that the cell library sizes in the semi-synthetic dataset closely resemble those in the reference dataset, we independently sampled $\omega_{i'j}$ from the empirical distribution $\{\hat{\mu}_{ij}/\tilde{\mu}_{kj} : x_i = \tilde{x}_k; i = 1, \ldots, n\}$ for the $k$th metacell, where $\hat{\mu}_{ij}$ is the parameter estimate for the $i$th real cell, of the same type as the metacell, in the reference dataset, and $\tilde{\mu}_{kj}$ is the parameter of the $k$th metacell.

As a result of the above steps, the generated semi-synthetic dataset highly mimics the reference dataset (Supplementary Fig. 1b, c).

### Sensitivity analysis of the number of features used

mcRigor requires as input a metacell partition to evaluate and the single cell sequencing data represented as an $n \times p$ cell-by-feature data matrix, where $p$ is the number of highly variable features selected (following `Seurat` V5 default feature selection) and can be viewed as a hyperparameter of mcRigor. Intuitively, a larger $p$ results in a sparser data matrix, making the computation of metacell feature correlation matrices in Steps 1 and 2 of mcRigor more challenging. Conversely, a smaller $p$ retains less information, potentially omitting important features and leading to unreliable results. The default setting in mcRigor is $p = 2000$, consistent with `Seurat`'s default feature selection. In this section, we conduct a sensitivity analysis on $p$ and demonstrate that this default value is reasonable.

### Sensitivity analysis for scRNA-seq data. 

We first investigated how the choice of $p$ affects mcRigor's performance on scRNA-seq data. Specifically, we applied mcRigor using $p = 100, 200, 300, 500, 1000, 1500, 2000, 2500, 3000, 3500, 4000, 4500,$ and 5000 highly variable genes, selected by the R package `Seurat` (v5.1.0), to the bone marrow mononuclear cell CITE-seq dataset (`bmcite` dataset[39] from the R package `SeuratData`). We used the metacell partition generated by SEACells, which by default selects 2000 features via the Python package `Scanpy`. For each value of $p$, we evaluated the number of dubious metacells detected by mcRigor, the number of single cells composing these dubious metacells, and the Score computed by mcRigor for $\gamma = 2, 3, \ldots, 100$ (Supplementary Fig. 23a). As expected, the number of dubious metacells and their constituent single cells increased with $\gamma$ across all values of $p$. Notably, for $p \geq 1000$, these quantities exhibit stable patterns across $\gamma$, whereas smaller values of $p$ yield more erratic behavior (Supplementary Fig. 23a, top left and middle). Similarly, the Score values for larger values of $p$ ($p \geq 1000$) were consistent, with their maxima occurring at the same $\gamma$ (Supplementary Fig. 23a, top right). These results suggest that mcRigor produces stable outputs when used with sufficiently large values of $p$ ($p \geq 1000$). Furthermore, we computed the Jaccard indices between the sets of dubious metacells identified at different values of $p$ and observed strong agreement for $p \geq 1500$ (Supplementary Fig. 23a, bottom left), indicating that the default setting of $p = 2000$ yields results comparable to those obtained with larger $p$. These findings support the choice of $p = 2000$ as a reasonable default.

To further assess the sensitivity of mcRigor and our double permutation approach to the choice of $p$, we examined the distributions of mcDiv and mcDiv$^{\text{null}}$ at each $p$ for the SEACells metacell partition with $\gamma = 50$. We observed that as $p$ increases, the mcDiv distribution becomes more concentrated, and mcDiv$^{\text{null}}$ follows a similar trend (Supplementary Fig. 23a, bottom middle and right). This parallel behavior indicates that the influence of $p$ on mcDiv is effectively captured by the corresponding mcDiv$^{\text{null}}$ distribution, supporting the validity of our double permutation approach as a reliable method for constructing the null distribution of mcDiv.

We performed similar analyses on the `bmcite` dataset using metacell partitions generated by SuperCell and MetaCell, and obtained consistent results (Supplementary Fig. 24). Based on these findings, we conclude that mcRigor is robust to the number of selected features $p$, provided that $p$ is not too small ($p \geq 1500$), and that our default choice of $p = 2000$ is appropriate.

### Sensitivity analysis for scATAC-seq data. 

scATAC-seq data typically exhibits higher dimensionality and greater sparsity than scRNA-seq data, due to its binary nature and lower detection efficiency for regulatory elements. As a result, the number of selected features, $p$, is expected to have a more pronounced impact on the performance of mcRigor for scATAC-seq data than for scRNA-seq data. In this subsection, we assessed the impact.

Following the analysis described in the previous subsection, we applied mcRigor with $p = 100, 200, 300, 500, 1000, 1500, 2000, 2500, 3000, 3500, 4000, 4500,$ and 5000 to the scATAC-seq modality of a single-cell multiome dataset comprising 6881 HSPCs[33]. Using the metacell partition generated by SEACells—the same partition used in the "Results" section "mcRigor empowers and rectifies gene regulatory inference on single-cell multiome data"—we evaluated, for each value of $p$, the same three mcRigor-derived metrics: the number of dubious metacells, the number of single cells comprising these dubious metacells, and the Score, across an extended range of $\gamma$ values ($\gamma = 2, 3, \ldots, 200$). Similar to the results for scRNA-seq data, these metrics remained stable for $p \geq 1000$ (Supplementary Fig. 23b, top). To assess the adequacy of the default $p = 2000$, we computed the Jaccard indices between sets of dubious metacells identified at different values of $p$ and observed strong concordance among results for large $p$ ($p \geq 1000$) (Supplementary Fig. 23b, bottom left). These findings confirm that mcRigor's performance is robust as long as $p$ is sufficiently large ($p \geq 1000$), and they support the use of $p = 2000$ as a reasonable default.

As in the previous subsection, the validity of the double permutation approach is further supported by the observation that the mcDiv$^{\text{null}}$ distribution exhibits the same trend as the mcDiv distribution as $p$ increases (Supplementary Fig. 23b, bottom middle and right), indicating that the effect of $p$ on divergence score computation is effectively accounted for by the null distribution.

### Execution time of mcRigor

We report the execution time of mcRigor on an Intel Xeon E5-2660 v3 system with 2.6 GHz CPU and 90.64 GB RAM (Table 1). mcRigor's computational cost for detecting dubious metacells is low, consistently under 2 min for all datasets. While optimizing metacell configuration takes longer—typically under 2 h for a large dataset of around 10,000 cells—the computation cost remains manageable and can be significantly reduced by employing parallel computing across different $\gamma$ values. Additionally, the cost can be further lowered by using a smaller candidate pool of $\gamma$ values, such as $\gamma = 5, 10, \ldots, 100$ instead of $\gamma = 2, 3, \ldots, 100$. This is a reasonable approach in practice since close $\gamma$ values often yield similar partitionings[34].

**Table 1 | The execution time of mcRigor (in min) for detecting dubious metacells and optimizing metacell method-hyperparameter configuration**

| Dataset | # cell | γ | Detecting dubious metacells | | | | | | | | γ | Optimizing metacell partitioning | | | | | | | |
|---|---|---|---|---|---|---|---|---|---|---|---|---|---|---|---|---|---|---|---|
| | | | SEACells | | MetaCell | | MetaCell2 | | SuperCell | | | SEACells | | MetaCell | | MetaCell2 | | SuperCell | |
| | | | Original | mcRigor | Original | mcRigor | Original | mcRigor | Original | mcRigor | | Original | mcRigor | Original | mcRigor | Original | mcRigor | Original | mcRigor |
| COVID-19 PBMC | 3028 | 50 | 5.646 | 0.185 | 3.840 | 0.232 | 0.367 | 0.216 | 0.162 | 0.186 | 2-100 | 920.064 | 8.788 | 225.412 | 7.382 | 17.281 | 7.734 | 17.283 | 8.962 |
| scMulitome | 6800 | 100 | 9.519 | 1.319 | — | — | — | — | 3.705 | 1.355 | 2-200 | >1000 | 91.183 | — | — | — | — | 743.041 | 104.013 |
| Semi-synthetic | 13400 | 50 | 3.001 | 1.620 | 2.066 | 1.543 | 2.830 | 1.782 | 1.712 | 1.695 | 2-100 | >1000 | 136.022 | 220.464 | 125.906 | 327.598 | 116.879 | 167.522 | 148.076 |
| Dropseq + smRNA FISH | 8498 | 50 | 58.608 | 0.721 | 4.057 | 0.673 | 2.394 | 0.659 | 1.956 | 0.801 | 2-100 | >1000 | 75.586 | 479.197 | 68.833 | 200.246 | 73.795 | 193.688 | 75.281 |
| scRNA-seq + bulk ESC | 350 | 30 | 0.568 | 0.192 | 0.821 | 0.173 | — | — | 0.038 | 0.153 | 2-50 | 26.562 | 4.070 | 39.654 | 3.247 | — | — | 1.935 | 3.646 |
| Zman-seq | 2431 | 50 | 4.749 | 0.185 | 1.341 | 0.167 | 0.578 | 0.185 | 0.219 | 0.177 | 2-100 | 774.616 | 8.284 | 154.848 | 7.926 | 30.493 | 7.383 | 22.242 | 7.925 |

Column names: "original": the time for implementing the original metacell methods; "mcRigor": the time for applying mcRigor to the generated metacell partition(s). "—" indicates the method is inapplicable to the data (MetaCell and MetaCell2 are inapplicable to the scATAC-seq modality in "scMultiome" dataset; MetaCell2 is inapplicable to the "scRNA-seq + bulk ESC" dataset due to the small number of cells).

## Implementation of metacell partitioning methods

In our analysis, we implemented four different metacell partitioning methods—SEACells[33], MetaCell[17], MetaCell2[31], and SuperCell[32]—all of which mcRigor is compatible with. The SEACells method was executed using the Python package `SEACells` (version 0.3.3), following the detailed vignette available at https://github.com/dpeerlab/SEACells/blob/main/notebooks/SEACell_computation.ipynb. MetaCell was implemented with the R package `metacell` (version 0.3.7) following the vignette at https://tanaylab.github.io/metacell/articles/a-basic_pbmc8k.html, and MetaCell2 was implemented using the Python package `metacells` (version 0.9.5). The SuperCell method was carried out with the R package `SuperCell` (version 1.0).

Note that originally, only the SEACells method could be applied to identify metacell partitions for the scATAC-seq modality. To adapt the SuperCell method for scATAC-seq, we modified it by replacing its original normalization and dimensionality reduction steps with term frequency-inverse document frequency (TF-IDF) and singular value decomposition (SVD), respectively. However, since these two steps are built within the MetaCell and MetaCell2 methods, we were not able to modify them to accommodate scATAC-seq. Consequently, we used only SEACells and SuperCell for the scMultiome (RNA + ATAC) data (Table 1).

## Projection of metacells onto the single-cell space for visualization

We visualized single cells with the identified metacells overlaid (Fig. 1c and Supplementary Figs. 4, 5a, and 7–10b). Specifically, single cells were first plotted based on their two-dimensional embeddings (UMAP or t-SNE) using the `geom_point()` function from the R package `ggplot2` (v3.6.6). Each metacell was then positioned at the centroid of the single cells it encompasses in the two-dimensional embedding space and colored according to the predominant cell type within it. The dot size for each metacell reflects its actual metacell size, which is the number of single cells it includes.

## Data analysis

**Cell line data for cell cycle analysis.** We considered five scRNA-seq datasets, each from one cell line. Three of these datasets, from cell lines A549, H2228, and HCC827, were generated using scRNA-seq with the 10× Chromium protocol[72]. The other two datasets, from cell lines HEK293T and Jurkat, were sequenced using the Illumina HiSeq2500 Rapid Run V2 platform[3]. Raw sequencing data for A549, H2228, and HCC827 are available at the NCBI Gene Expression Omnibus (GEO) with accession code GSE126906, while the raw data for HEK293T and Jurkat can be accessed at http://support.10xgenomics.com/single-cell/datasets under the titles "293T Cells" and "Jurkat Cells," respectively. These five datasets from cell lines A549, H2228, HCC827, HEK293T, and Jurkat contain 1237, 744, 567, 2885, and 3258 single cells, respectively.

**Cell-cycle phase annotation.** we assigned each single cell to a specific cell-cycle phase using canonical gene markers provided by ref. 43. This cell-cycle phase annotation was performed with the `CellCycleScoring` function from the `Seurat` R package (version 5.1.0). The corresponding R code is: `CellCycleScoring(Seurat.obj, s.features = s.genes, g2m.features = g2m.genes)`, where `s.genes` and `g2m.genes` represent the canonical gene markers for the S and G2M phases, respectively. Each metacell was annotated with the cell-cycle phase that contained the highest fraction of single cells within this metacell.

**COVID-19 PBMC data for gene co-expression analysis.** The data were collected by applying scRNA-seq through the Seq-Well platform[73] to profile PBMCs from seven patients hospitalized for COVID-19 and 6 healthy controls[46]. The processed count data and corresponding

metadata can be accessed from the COVID-19 Cell Atlas (https://www.covid19cellatlas.org/index.patient.html) under the label PBMCs. Raw sequencing data are available at NCBI GEO with accession code GSE150728. The processed data contains 44,721 single cells, among which 28,094 are from COVID-19 patients and 16,627 are from healthy controls. We used a subset of this process data, all the cells annotated as B cells, for metacell partitioning and gene co-expression analysis. Note that both metacell partitioning and gene co-expression calculation were performed on the COVID-19 group (3028 cells) and the healthy group (1994 cells) separately.

**Gene module co-expression enrichment testing**: for each gene module with $g$ genes, we viewed the $g \times g$ gene correlation values as observations. Between the two conditions (COVID-19 vs healthy), we applied the Wilcoxon rank-sum test to determine if the average correlation in the COVID-19 group was significantly higher than in the healthy group. A gene module was considered enriched in co-expression for COVID-19 if the Wilcoxon rank-sum test yielded a small one-sided $p$-value. We note that while we used the $p$-value to indicate enrichment (with smaller $p$-values representing stronger enrichment), we acknowledge that the use of the Wilcoxon rank-sum test in this context is ad hoc, as the correlation values are not independent nor identically distributed.

**Single-cell multiome data for gene regulatory inference.** This single-cell multiome (scMultiome) dataset, generated by ref. 33, is archived at NCBI GEO with accession code GSE200046. Data were collected by performing single-cell multiome ATAC + gene expression on cryopreserved bone marrow stem/progenitor CD34+ cells from a healthy donor, utilizing the 10X Genomics Chromium system and the Chromium Next GEM Single Cell Multiome Reagent Kit. The preprocessed dataset consists of 6881 HSPCs from healthy bone marrow, sorted for the pan-HSPC marker CD34.

**Peak-gene associations**: gene-peak pairs were identified using the `LinkPeaks` function from the `Signac` R package (version 1.13.0). For each gene, this function computes a correlation score and $p$-value between gene expression and the accessibility of each peak within a specified distance (set at 100,000 base pairs) from the gene's transcription start site (TSS).

**Paired scRNA-seq and smRNA FISH cell line data for distinguishing between biological and non-biological zeros.** Data were generated by ref. 53 using scRNA-seq via the Drop-seq platform on 8498 cells from a melanoma cell line. Paired smRNA FISH data were obtained from the same cell line, measuring 26 drug resistance markers and housekeeping genes across 7000–88,000 cells. These paired datasets are available at NCBI GEO with accession code GSE99330. Sixteen genes overlap between the two datasets, including seven drug-resistance markers (*C1S*, *FGFR1*, *FOSL1*, *JUN*, *RUNX2*, *TXNRD1*, and *VCL*) and nine housekeeping genes (*BABAM1*, *GAPDH*, *LMNA*, *CCNA2*, *KDM5A*, *KDM5B*, *MITF*, *SOX10*, and *VGF*). Since our focus is on the number of zero counts, no normalization was performed to account for the technical differences between the platforms.

**Zero proportion calculation**: for the 16 overlapping genes as a whole, the zero proportion was calculated as the proportion of zeros within the observation-by-gene matrix (spatial spot-by-gene matrix for the smRNA FISH data or metacell-by-gene matrix for the metacell data derived from scRNA-seq data). For each individual gene, the zero proportion was calculated as the proportion of zeros within the vector of that gene's counts across all observations (spatial spots for the smRNA FISH data or metacells for the metacell data derived from scRNA-seq data).

**Paired scRNA-seq and bulk data of human progenitor cells for DGE analysis.** This dataset was obtained by performing scRNA-seq on snapshots of lineage-specific progenitor cells differentiated from H1

human embryonic stem cells (H1 hESC)[55]. The original dataset, archived at NCBI GEO with accession code GSE75748, contains 1018 single cells and 19 bulk RNA-seq samples from snapshot progenitors. We used a subset of this dataset comprising 350 scRNA-seq cells—212 from H1 hESC and 138 from DEC—and six corresponding bulk RNA-seq samples (four from H1 hESC and two from DEC). Due to the small total cell number, we considered smaller granularity levels ($\gamma = 2, \ldots, 50$) for metacell partitioning optimization.

**Identification of DE genes**: the `FindMarkers` function from the `Seurat` R package (version 5.1.0) was used to identify DE genes between the two cell types—H1 hESC and DEC. For fair comparison, we employed the DESeq2 method to identify DE genes in both the bulk and scRNA-seq data. The R code used is: `FindMarkers(Seurat.obj, ident.1 = 'H1 hESC', ident.2 = 'DEC', test.use = 'DESeq2', min.cells.group = 2)`.

**Zman-seq data for temporal trajectory analysis.** This data, generated by ref. 30, profiles immune cells in mouse glioblastoma using the Zman-seq technology, which tracks transcriptomic dynamics over time by introducing fluorescent time stamps into immune cells. Processed data and metadata, including time stamps, are available at NCBI GEO with accession code GSE232040. We used a subset of the processed data consisting of 2431 intratumoral T and NK cells.

**Metacell annotation**: we annotated each metacell by first examining the highest fraction of single cells within this metacell that originated from the same cell type. If this fraction is above 80%, the metacell is annotated with the corresponding major cell type. Otherwise, manual annotation was performed using marker genes suggested by ref. 30, including *S1pr5*+ for NK Chemotactic; *Prf1*, *Gzma*+, and *Gzmb*+ for NK Cytotoxic; *Itga1*+, *Xcl1*+, *Eomes*−, and *Pmepa1*+ for NK Dysfunctional.

**Identification of time-dependent genes**: following the pipeline from ref. 30, we identified time-dependent genes and generated heatmaps for smoothed gene expression over time. The R functions for this analysis, along with step-by-step guidance, are available at https://github.com/kenxie7/ZmanR. Specifically, after the cTET value is obtained for each metacell, we identified top genes significantly correlated with cTET using Spearman's rank correlation. Over the obtained $p$-values (from the Spearman's rank correlation test), we then applied Benjamini–Hochberg (BH) correction to adjust for multiple testing and identify significant time-dependent genes (FDR = 0.05). Gene expression was smoothed across cTET using loess fitting and normalized to range from 0 to 1 across all metacells. With these smoothed expressions, the `Heatmap` function from the R package `ComplexHeatmap` (version 2.18.0) was used to visualize and cluster genes with similar temporal patterns.

### Reporting summary
Further information on research design is available in the Nature Portfolio Reporting Summary linked to this article.

## Data availability
The datasets used in this paper are all publicly available. The COVID-19 PBMC dataset is available at the NCBI GEO under accession code GSE150728. The scMultiome dataset is available at the NCBI GEO under accession code GSE200046. The Dropseq + smRNA FISH dataset is available at the NCBI GEO under accession code GSE99330. The scRNA-seq + bulk ESC dataset is available at the NCBI GEO under accession code GSE75748. The Zman-seq dataset is available at the NCBI GEO under accession code GSE232040. For the cell line datasets, raw data for A549, H2228, and HCC827 are available at the NCBI GEO under accession code GSE126906, and the raw data for HEK293T and Jurkat can be accessed at the 10x Genomics website [http://support.10xgenomics.com/single-cell/datasets] under the titles "293T Cells" and "Jurkat Cells," respectively. Source Data files sufficient to generate

the figures and Supplementary Figs. are provided with this paper through Zenodo [https://doi.org/10.5281/zenodo.16309527].

## Code availability

The `mcRigor` R package is available at the GitHub repository [https://github.com/JSB-UCLA/mcRigor] (Zenodo [https://doi.org/10.5281/zenodo.16436345])[74]. Implementations of SEACells, MetaCell, MetaCell2, and Supercell, as well as the generated semi-synthetic dataset, are also included in the GitHub repository. The source code for reproducing the results are available at https://github.com/chrystal23/mcRigor_reproduce.

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

## Acknowledgements

We thank Mannix Burns for the valuable idea of using barcode multiplet data for our method validation. This work was supported by the following grants: National Science Foundation DBI-1846216 and DMS2113754, NIH/NIGMS R35GM140888, and Silicon Valley Community Foundation 2022-249355 (Chan-Zuckerberg Initiative Single-Cell Biology Data Insights Grant) (to J.J.L.). Additional support was provided by the National Human Genome Research Institute (NHGRI) through an Opportunity Fund subaward from the Technology Development Coordinating Center (TDCC) U24HG011735. This work was also supported by the Institute for Quantitative and Computational Biosciences (QCBio) at the University of California, Los Angeles, through the QCBio Collaboratory Fellowship (to Pan Liu).

## Author contributions

P.L. and J.J.L. wrote the manuscript. P.L. prepared the figures and tables. P.L. and J.J.L. revised the manuscript, and all authors approved the final version of the manuscript for publication.

## Competing interests

The authors declare no competing interests.
