## [Transparent Peer Review file · Nature Communications]

mcRigor: a statistical method to enhance the rigor of metacell partitioning in single-cell data analysis

Corresponding Author: Professor Jingyi Jessica Li

Version 0:

Reviewer comments:

Reviewer #1

(Remarks to the Author)

mcRigor is a novel tool for identifying homogeneous versus heterogeneous metacell partitions from any metacell algorithms. The method is well grounded in statistical motivation, approach and experimental tests. This is a high-quality manuscript that addresses a timely need in the field to increase the robustness of analysis using metacells. The manuscript is well-written placing prior work in context, highlighting the need for mcRigor and in description of the methods and results. Therefore, this submission is an excellent candidate for publication in Nature Communications.

I have two comments for the authors to consider:

1. I think it has been clear in the field and as highlighted in the manuscript as well, the so-called “dubious” metacells are not always truly dubious but a resolution issue where heterogeneous states occupying low-density/transitory regions of the phenotypic manifold are aggregated together. Given the large differences in how states are distributed in any single-cell dataset, it is very hard to derive one particular resolution parameter that fits the complexity of the data without throwing out cells as once highlighted by the authors. Therefore, one could consider using a step-wise approach to resolve this: In step 1, mcRigor can be applied with a resolution optimized by the Score in Fig 2. In step 2, rather than throwing out dubious metacells, a lower threshold can be used (say 85%) and the metacell algorithm can be rerun only on the subset of cells that are now marked as dubious with an updated resolution parameter. This might help resolve biologically meaningful cell-states and truly dubious metacells.

2. The divergence score relies on computation of Pearson correlations on extremely sparse data. The null divergence with double shuffling clearly helps solve the issues of computing correlations on sparse data but it will be good to add robustness analyses based on number of genes selected. I am particularly concerned with this with scATAC data where the structure is identified better by genomic bins rather than gene scores and as a result sparsity of features is substantially amplified.

(Remarks on code availability)

I briefly worked through the repository and it is well written, well documented and instructions for usage were clear.

Reviewer #2

(Remarks to the Author)

Partitioning single cells into metacells is a meaningful step in single cell data analysis. By defining metacells, we can pay greater attention to cellular heterogeneity. However, there is always a conflict between the homogeneity required by metacells and the actual heterogeneity of cells. Therefore, finding a balance between the two presents a significant challenge. To solve this issue, this study proposed a novel statistical method to enhance metacell selection, which could identify dubious metacells that are heterogeneous. It is an interesting point, which would be highly beneficial for large-scale single-cell data analysis. The results show its effectiveness in real scRNA-seq datasets. Here are some of my comments:

1) mcRigor firstly calculates the metacell feature correlation matrix R using Pearson correlation. However, when the number of cells in this metacells is relatively small, does the calculation of Pearson correlation still hold statistical significance?

2) The divergence score threshold parameter likely has a significant impact on the results. In selecting the divergence score threshold in Equation 6, the author mentions the bandwidth m_k . Why is this bandwidth used?

- 3) In section 2.4, the author said "...the trustworthy metacells identified by mcRigor exhibited significantly higher phase purity than the dubious metacells....", however, the p-value for Metacell in H2228 is 0.13904, and the authors need to provide an explanation for this non-significant result in order to offer guidance to users regarding the selection of the metacell method.
- 4) In fact, we observe that although there is a difference in cell cycle stage purity between trustworthy metacells and dubious metacells, what is the ratio of these two groups of metacells? If the proportions are similar, the violin plot reveals that many trustworthy metacells and dubious metacells have overlapping cell cycle stage purities. In practical applications, how significant is the identification of intra-cell-type heterogeneity in this context?
- 5) It is evident that cellular heterogeneity is hierarchical, including different major cell types, subtypes, and even cell states. The authors could discuss the ability of mcRigor across different levels of heterogeneity to demonstrate its generalizability.
- 6) This article discusses that many of the comparisons and analyses focus on the issue of biological zero, and many imputation methods also solve this problem. Therefore, if imputation is performed prior to constructing metacells, could this significantly reduce the number of dubious metacells?

(Remarks on code availability)

Yes, the code provide a README file with enough instructions for installing and running mcRigor

Reviewer #3

(Remarks to the Author)

The study introduces mcRigor, a statistical framework designed to evaluate whether a metacell comprises a sufficiently homogeneous population of cells. The method operates under the assumption that a properly constructed metacell consists of cells in the same biological state, with the variability within the metacell attributable only to technical noise and not biological differences. Central to mcRigor is the hypothesis that, in an ideal metacell, the correlation matrix of the metacell feature approximates an identity matrix due to the absence of biological variance. Based on this premise, the authors define a metacell divergence score (mcDiv) to quantify the extent of deviation from this ideal state. Furthermore, mcRigor optimizes metacell partitioning by balancing two metrics: DubRate (a measure of dubious cell inclusion) and ZeroRate (a reflection of the sparsity of data).

Through extensive analyses, the authors demonstrate the efficacy of mcRigor in identifying dubious cells and the optimal number of metacells in various datasets. However, the framework is primarily a post-processing step applied after the construction of metacells by existing methods, which somewhat limits its originality. Rather than simply excluding dubious cells, an approach that reconstructs metacells directly could potentially enhance the robustness and utility of the method.

Questions:

1. In the metacell approach, clusters are formed based on γ , and each cluster is averaged to create a metacell. During this process, "dubious" metacells—those containing heterogeneous cells—may emerge, which mcRigor subsequently removes. A potential concern is that small clusters with rare cells ($\gamma^k \ll \gamma$) are prone to merging into larger clusters, resulting in heterogeneous metacells that are flagged and removed as dubious. Consequently, rare cells might be lost because mcRigor labels them as dubious. To address this, the authors should present experiments examining how mcRigor handles rare cell types. In particular, how well does mcRigor identify rare cell states? A systematic evaluation is necessary to ensure that rare cell types are not inadvertently removed due to their inclusion in dubious metacells.
2. How does mcRigor perform when applied across multiple integrated cohorts? SEACells has demonstrated its ability to preserve biological signals despite substantial technical noise—for instance, by analyzing CD4+ T cells from multiple cohorts in Figure 6 of the SEACells paper. The question remains whether mcRigor can further enhance these outcomes.
3. In Fig. 1e, "trustworthy metacells" effectively distinguished healthy samples from those with COVID-19. However, a question arises whether "all metacells" could achieve similar performance by adjusting certain parameters, such as γ . For example, constructing metacells under stricter conditions (e.g., using a lower γ) might yield high-quality metacells even without mcRigor, although this could increase the number of zero counts. Moreover, single-cell analysis alone yielded performance comparable to mcRigor, indicating that zero counts do not substantially hinder the analysis.
4. In Section 2.6, mcRigor was applied to single-cell multiome data. The authors should compare results from a "rigid" setting—producing many metacells—to a "flexible" setting—producing fewer metacells, then using mcRigor to filter out dubious ones. In my view, using a rigid setting (e.g., lowering γ) can reduce dubious metacells, albeit at the cost of increased sparsity. However, with 2,000 highly variable genes, this sparsity would likely not be severe. Most importantly, the authors must demonstrate that mcRigor's utility cannot be replicated simply by tuning existing parameters such as γ . If it can, the novelty of mcRigor is undermined.
5. In Section 2.10, to demonstrate that mcRigor's optimized metacell partition better reveals temporal immune cell trajectories, the authors compared MetaCell (from the original study) with SEACells followed by mcRigor. However, it remains unclear whether the improvement is due to mcRigor or SEACells itself. To clarify this, the authors should provide direct comparisons of each method with and without mcRigor—for example, MetaCell vs. MetaCell + mcRigor, and SEACells vs. SEACells + mcRigor.

(Remarks on code availability)

I ran tutorials including "Functionality 1: detect dubious metacells for a given metacell partition" and "Functionality 2: optimize metacell partitioning". The codes in the tutorials worked as explained.

Version 1:

Reviewer comments:

Reviewer #1

(Remarks to the Author)

The authors have satisfactorily addressed my concerns and I support the publication of this manuscript in Nature Communications.

(Remarks on code availability)

Reviewer #2

(Remarks to the Author)

The authors have thoroughly addressed the majority of the comments. This paper is a valuable contribution, and I recommend the paper for publication.

(Remarks on code availability)

Reviewer #3

(Remarks to the Author)

The mcRigor two-step approach appears to offer a promising solution for the precise detection of dubious metacells. However, the descriptions of Supplementary Figures 18 and 19 require further clarification. In particular, the distinction between the results produced by mcRigor and those obtained with the mcRigor two-step method is not sufficiently clear. Providing detailed information on the number of trustworthy and dubious metacells identified per cell type, as well as a clear explanation of the significance of the brown area in the bar plots, would greatly enhance the interpretability of these results.

(Remarks on code availability)

Response to Reviewers' Comments on “mcRigor: a statistical method to enhance the rigor of metacell partitioning in single-cell data analysis”

We sincerely thank the three reviewers for their thoughtful evaluation of our manuscript and their constructive and encouraging feedback. We have carefully addressed all comments and revised the manuscript accordingly.

Our detailed, point-by-point responses to the reviewers' comments are provided on the following pages. Reviewers' comments appear in **mahogany**, our responses are in **black**, and changes made to the revised manuscript and supplementary materials are highlighted in **blue** and quoted in this response letter.

Answers to Reviewer 1

mcRigor is a novel tool for identifying homogeneous versus heterogeneous metacell partitions from any metacell algorithms. The method is well grounded in statistical motivation, approach and experimental tests. This is a high-quality manuscript that addresses a timely need in the field to increase the robustness of analysis using metacells. The manuscript is well-written placing prior work in context, highlighting the need for mcRigor and in description of the methods and results. Therefore, this submission is an excellent candidate for publication in Nature Communications.

We thank the reviewer for appreciating the value of our work and providing the constructive comments below.

Comment R1.1 I think it has been clear in the field and as highlighted in the manuscript as well, the so-called “dubious” metacells are not always truly dubious but a resolution issue where heterogenous states occupying low-density/transitory regions of the phenotypic manifold are aggregated together. Given the large differences in how states are distributed in any single-cell dataset, it is very hard to derive one particular resolution parameter that fits the complexity of the data without throwing out cells as once highlighted by the authors. Therefore, one could consider using a step-wise approach to resolve this: In step 1, mcRigor can be applied with a resolution optimized by the Score in Fig 2. In step 2, rather than throwing out dubious metacells, a lower threshold can be used (say 85%) and the metacell algorithm can be rerun only on the subset of cells that are now marked as dubious with an updated resolution parameter. This might help resolve biologically meaningful cell-states and truly dubious metacells.

Answer to R1.1 We thank the reviewer for suggesting this insightful two-step approach for handling detected dubious metacells. This comment also guided our response to Reviewer 3’s Comment 1 concerning the detection of rare cell types. We have implemented the proposed approach and evaluated it on two datasets: (1) the COVID-19 PBMC dataset and (2) the paired scRNA-seq and smRNA FISH cell line dataset. This approach demonstrated strong performance in both cases. We have described it—termed “mcRigor two-step”—in the Discussion section (quoted below), with a detailed methodological description and analysis results provided in the Supplementary file (quoted below).

Quote from the Discussion section: “As a first step toward improving metacell reconstruction, we developed a straightforward extension of mcRigor, termed *mcRigor two-step*, as described in the Supplementary file (Supplementary section “Performance of mcRigor under varying cellular heterogeneity”). This approach first identifies more dubious metacells using a relaxed mcDiv threshold—specifically, the 85th percentile of mcDiv null values conditional on metacell size, as opposed to the default 95th percentile used in standard mcRigor—and then re-applies metacell partitioning to the constituent cells of these dubious metacells, using a granularity level opti-

mized by mcRigor. Rather than discarding the dubious metacells, mcRigor two-step re-partitions their constituent cells, thereby reducing information loss. We demonstrated that this extension can further improve downstream analyses, including uncovering differential gene co-expression modules (Supplementary Fig 15 and Supplementary Fig 16) and distinguishing biological from non-biological zeros (Supplementary Fig 17), and more importantly, enables better detection of rare cell subpopulations (Supplementary Fig 18 and Supplementary Fig 19), whose cells are often mixed with others in dubious metacells due to their small numbers. By reorganizing these cells into smaller, trustworthy metacells, mcRigor two-step enhances mcRigor’s ability to capture cellular heterogeneity and uncover biological states with varying abundance levels. Nonetheless, mcRigor two-step remains an initial attempt at refining metacell reconstruction—a more principled redesign of metacell partitioning, building on mcRigor, will require future, more deliberate method development.”

Quote from the Supplementary file: “Following the mcRigor pipeline described in our main text, one can generate a metacell partition from a given single-cell dataset using any preferred partitioning method at a chosen granularity—either user-defined or selected via mcRigor—and then apply mcRigor to identify dubious metacells within the partition. To ensure the reliability of downstream analyses, a straightforward option is to remove these dubious metacells. However, this risks discarding cells from rare biological states, potentially leading to the loss of critical information relevant to biological processes, such as disease occurrence. To address this issue, we propose a two-step approach, referred to as *mcRigor two-step*, which dissects the identified dubious metacells and reorganizes their constituent cells into more trustworthy metacells.

Step 1: A method–hyperparameter configuration (i.e., a metacell partitioning method with a granularity level γ_1) is either specified by the user or selected by mcRigor. This configuration is then applied to partition single cells into metacells. If mcRigor detects dubious metacells within the partition, it is re-applied to the same partition using a lower divergence score threshold. Specifically, the default threshold $q_{0.95}$ (the 95th percentile of mcDiv null values conditional on metacell size) in equation (5) is replaced with a more relaxed threshold, $q_{0.85}$ (the 85th percentile), to label more metacells as dubious for further dissection and reorganization.

Step 2: The selected metacell partitioning method is re-applied to the subset of single cells that belong to the metacells now marked as dubious. This yields a refined metacell partition under a new granularity level $\gamma_2 < \gamma_1$, which can be selected by mcRigor from the candidate set of granularity levels $2, \dots, \gamma_1 - 1$. ”

Also in the **Supplementary file**, we have included the **analysis results of mcRigor two-step** on two datasets. A brief summary of the results is provided below.

(1) The COVID-19 PBMC dataset (gene co-expression analysis).

Quote from the main text regarding the results of mcRigor: “We applied mcRigor to an

scRNA-seq dataset from human PBMCs of seven hospitalized COVID-19 patients and six healthy controls [13]. In this analysis, mcRigor successfully rectified co-expression estimates and uncovered gene modules differentially co-expressed between the COVID-19 and healthy cohorts. Given that co-expression patterns are often specific to cell type and condition [7, 10], we applied SuperCell with $\gamma = 20$ separately to the 3,028 B cells from COVID-19 samples and the 1,994 B cells from control samples. From the two metacell partitions generated by SuperCell (one per condition), mcRigor detected 22 dubious metacells among the 152 metacells identified in the COVID-19 group and 26 dubious metacells out of the 99 metacells in the control group.

We found that excluding dubious metacells prior to correlation estimates removed biases and highlighted differentially co-expressed gene modules that may play a role in the COVID-19 disease mechanism. For example, correlation estimation using only trustworthy metacells revealed three co-expression gene modules enriched in the COVID-19 cohort (Fig 1e), representing differential biological functions of B cells, including the antigen processing via MHC Class II gene module (p-value = $3.7e-31$ for one-sided Wilcoxon rank-sum test), the adaptive immune response gene module (p-value = $7.6e-19$), and the response to interferon-alpha gene module (p-value = 0.00328). These enrichment signals were notably strengthened compared to those in the raw single-cell data, where the p-values for these three gene modules are $2.0e - 13$, 0.00043, and 0.08441, respectively. We also observed that these enrichment findings are consistent with those reported by the CS-CORE method [10].

In contrast, correlation estimation that included dubious metacells (using all metacells) yielded the counterintuitive result that the adaptive immune response gene module was not enriched in COVID-19 patients (p-value = 0.54632). We found that this misleading result was caused by an artifact: the strong co-expression of the adaptive immune response gene module under the healthy condition. Notably, this artifact was absent at single-cell resolution and was induced solely by the presence of dubious metacells (Fig 1e). Applying mcRigor to the partitions generated by SEACells, MetaCell, and MetaCell2 similarly removed the artifact correlations introduced by dubious metacells and enhanced enrichment signals (Supplementary Fig 4), demonstrating the applicability and robustness of mcRigor across various metacell methods.”

Quote from the Supplementary file regarding the new results of mcRigor two-step (with Supplementary Fig 15 shown on Page 6): “We applied mcRigor two-step to the metacell partitions generated by SEACells, SuperCell, and MetaCell under $\gamma_1 = 30$ for the COVID-19 PBMC dataset (Supplementary Fig 15 and Supplementary Fig 16). Compared to the original mcRigor results, the two-step approach substantially reduced the number of dubious metacells by re-partitioning their constituent cells at a finer resolution. This reduction improved downstream analyses: the co-expression enrichment of adaptive immune response genes in COVID-19 patients—previously undetectable when using all metacells from the original partitions—became

evident when using all metacells from the refined partitions generated by the two-step approach.

Specifically, starting from the SEACells partition with a first-step granularity level of $\gamma_1 = 30$, mcRigor applied a finer second-step granularity level of $\gamma_2 = 10$ to re-partition only the cells belonging to initially identified dubious metacells (11 in the healthy cohort and 13 in the COVID-19 cohort). This two-step refinement resulted in a new partition containing only 3 dubious metacells in the healthy cohort and 5 in the COVID-19 cohort (Supplementary Fig 15a, left)—a marked reduction compared to the original SEACells partition at $\gamma_1 = 30$ (Supplementary Fig 15b, left).

Using the updated SEACells partition (without filtering out the few remaining dubious metacells), we observed a statistically significant enrichment of gene co-expression within the adaptive immune response module in the COVID-19 cohort compared to the healthy cohort (p-value = 0.00158, one-sided Wilcoxon rank-sum test). This result is consistent with our earlier finding using the original SEACells partition after removing dubious metacells (11 in the healthy cohort and 13 in the COVID-19 cohort) (Supplementary Fig 15b, middle), but was not observed when using the original SEACells partition without filtering out dubious metacells (Supplementary Fig 15b, right vs. Supplementary Fig 15a, right). Moreover, after filtering out the remaining dubious metacells (3 in the healthy cohort and 5 in the COVID-19 cohort) from the updated SEACells partition, the enrichment became even more significant (p-value = 5.4e-09, one-sided Wilcoxon rank-sum test) (Supplementary Fig 15a, middle). Similar improvements in gene co-expression analysis were observed for the SuperCell and MetaCell partitions, as shown in Supplementary Fig 16a–b.

These results demonstrate that the mcRigor two-step approach improves the detection of fine-grained cell states, which are essential for capturing cell state transitions and identifying rare cell types.”

(2) The paired scRNA-seq and smRNA FISH cell line dataset (used to distinguish biological from non-biological zeros).

Quote from the main text regarding the results of mcRigor: “We evaluated whether mcRigor can effectively distinguish biological zeros from non-biological ones using an scRNA-seq Drop-seq dataset paired with single-molecule RNA fluorescence in situ hybridization (smRNA FISH) data for 16 genes from a melanoma cell line [12]. The rationale for this approach is that smRNA FISH is widely considered the gold standard for single-cell gene expression measurement [3, 12], making it reasonable to assume that all zeros in smRNA FISH data are biological zeros. We applied MetaCell, SEACells, and SuperCell to the scRNA-seq dataset, varying the granularity level ($\gamma = 2, \dots, 100$), and calculated the proportion of zeros for the 16 genes in the resulting metacell-by-gene data matrix for each metacell partition. We then used mcRigor to calculate a *Score* for each partition and identify the optimal γ that yielded the best *Score* for each method (Supplementary Fig 8a–b). Notably, the γ value selected by mcRigor for each metacell partitioning method resulted

Supplementary Fig 15: **mcRigor two-step reduces the number of dubious metacells and improves gene co-expression analysis in the COVID-19 PBMC dataset using SEACells ($\gamma = 30$) metacell partitions.** **a**, Left: Single-cell UMAP plots showing metacell partitions and dubious metacells detected by mcRigor two-step under healthy (top row) and COVID-19 (bottom row) conditions. Right: Gene-gene correlation matrices for three key gene modules under healthy (top row) and COVID-19 (bottom row) conditions, based on three data types: single cells, trustworthy metacells (mcRigor two-step with filtering), and all metacells (mcRigor two-step without filtering). For each data type, the p-value comparing the correlation matrices of the adaptive immune response gene module between the two conditions was computed using a one-sided Wilcoxon rank-sum test. **b**, Same as **(a)**, but showing the results from mcRigor alone (without the mcRigor two-step extension).

in a proportion of zeros that closely matched the proportion of zeros in the smRNA FISH data (Fig 2b, Supplementary Fig 8c). Compared to single-cell data, the expression distributions of each gene derived from the optimal metacell partitions (one per metacell method) closely approximated

the corresponding distribution observed in the smRNA FISH data (Supplementary Fig 8d). These result confirm that mcRigor can help distinguish biological zeros from non-biological zeros by optimizing metacell partitioning.”

Quote from the Supplementary file regarding the new results of mcRigor two-step (with Supplementary Fig 17 shown on Page 8): “We applied mcRigor two-step to the mcRigor-optimized metacell partitions generated by MetaCell ($\gamma_1 = 24$), SEACells ($\gamma_1 = 44$), and SuperCell ($\gamma_1 = 56$) for the paired scRNA-seq and smRNA FISH dataset (Supplementary Fig 17). Notably, mcRigor two-step reduced the number of dubious metacells—for example, from five to three based on the MetaCell method (Supplementary Fig 17a–b)—and improved the distinction between biological and non-biological zeros. This is evidenced by better alignment between the zero proportions in the metacell-by-gene matrix and the gold-standard reference measured by smRNA FISH (Supplementary Fig 17c). For instance, under MetaCell partitioning, the proportion of zeros increased from 0.1622 (original mcRigor result) to 0.1846 after applying mcRigor two-step, approaching the smRNA FISH reference value of 0.1979. These results suggest that mcRigor two-step more effectively resolves cell states and better distinguishes biological zeros from technical artifacts.”

Comment R1.2 The divergence score relies on computation of pearson correlations on extremely sparse data. The null divergence with double shuffling clearly helps solve the issues of computing correlations on sparse data but it will be good to add robustness analyses based on number of genes selected. I am particularly concerned with this with scATAC data where the structure is identified better by genomic bins rather than gene scores and as a result sparsity of features is substantially amplified.

Answer to R1.2 We fully agree with the reviewer that the number of selected features (e.g., genes or genomic bins), denoted by p , is a critical factor that may influence mcRigor’s performance. Intuitively, increasing p leads to a sparser data matrix, which can make the computation of a feature correlation matrix for each metacell more challenging. Conversely, a smaller p may omit important features, reducing information content and potentially compromising the reliability of the results. To address this concern, we conducted comprehensive sensitivity analyses on both scRNA-seq and scATAC-seq datasets to assess the robustness of mcRigor to the choice of p . These analyses support the reasonableness of our default choice of $p = 2000$. The corresponding results have been added to the Methods section of the revised main text. A summary of our findings is provided below.

(1) Sensitivity analysis of the number of selected features (p) for scRNA-seq data:

Quote from the Methods section regarding the sensitivity analysis (with Sup-

Supplementary Fig 17: **mcRigor two-step reduces the number of dubious metacells and better distinguishes biological zeros from non-biological zeros in the paired scRNA-seq and smRNA FISH dataset.** **a**, Single-cell UMAP plots showing the refined metacell partitions generated by mcRigor two-step, applied to each of the three metacell partitioning methods (MetaCell, SEACells, and SuperCell), with dubious metacells highlighted by red circles. **b**, Same as (a), but showing the original metacell partitions from the three methods, with dubious metacells detected by mcRigor (without the mcRigor two-step extension) highlighted. **c**, Line plots showing the zero proportions in metacell partitions generated by the three methods across varying granularity levels (γ for mcRigor, right; and γ_2 for mcRigor two-step, left). The refined metacell partitions from mcRigor two-step (diamonds, left) align more closely with the zero proportion observed in smRNA FISH data (red line) compared to the mcRigor-optimized partitions (triangles, right).

plementary Fig 23 shown on Page 11): “We first investigated how the choice of p affects mcRigor’s performance on scRNA-seq data. Specifically, we applied mcRigor using $p = 100, 200, 300, 500, 1000, 1500, 2000, 2500, 3000, 3500, 4000, 4500,$ and 5000 highly variable genes, selected by the R package `Seurat` (v5.1.0), to the bone marrow mononuclear cell CITE-seq dataset (`bmcite` dataset [9] from the R package `SeuratData`). We used the metacell partition generated by `SEACells`, which by default selects 2000 features via the Python package `Scanpy`. For each value of p , we evaluated the number of dubious metacells detected by mcRigor, the number of single cells composing these dubious metacells, and the *Score* computed by mcRigor for $\gamma = 2, 3, \dots, 100$ (Supplementary Fig 23a). As expected, the number of dubious metacells and their constituent single cells increased with γ across all values of p . Notably, for $p \geq 1000$, these quantities exhibit stable patterns across γ , whereas smaller p values yield more erratic behavior (Supplementary Fig 23a, top left and middle). Similarly, the *Score* values for larger values of p ($p \geq 1000$) were consistent, with their maxima occurring at the same γ (Supplementary Fig 23a, top right). These results suggest that mcRigor produces stable outputs when used with sufficiently large values of p ($p \geq 1000$). Furthermore, we computed the Jaccard indices between the sets of dubious metacells identified at different values of p and observed strong agreement for $p \geq 1500$ (Supplementary Fig 23a, bottom left), indicating that the default setting of $p = 2000$ yields results comparable to those obtained with larger p . These findings support the choice of $p = 2000$ as a reasonable default.

To further assess the sensitivity of mcRigor and our double permutation approach to the choice of p , we examined the distributions of `mcDiv` and `mcDivnull` at each p for the `SEACells` metacell partition with $\gamma = 50$. We observed that as p increases, the `mcDiv` distribution becomes more concentrated, and `mcDivnull` follows a similar trend (Supplementary Fig 23a, bottom middle and right). This parallel behavior indicates that the influence of p on `mcDiv` is effectively captured by the corresponding `mcDivnull` distribution, supporting the validity of our double permutation approach as a reliable method for constructing the null distribution of `mcDiv`.

We performed similar analyses on the `bmcite` dataset using metacell partitions generated by `SuperCell` and `MetaCell`, and obtained consistent results (Supplementary Fig 24). Based on these findings, we conclude that mcRigor is robust to the number of selected features p , provided that p is not too small ($p \geq 1500$), and that our default choice of $p = 2000$ is appropriate. ”

(2) Sensitivity analysis of the number of selected features (p) for scATAC-seq data:

Quote from the Methods section regarding the sensitivity analysis (with Supplementary Fig 23 shown on Page 11): “scATAC-seq data typically exhibits higher dimensionality and greater sparsity than scRNA-seq data, due to its binary nature and lower detection efficiency for regulatory elements. As a result, the number of selected features, p , is expected to have a more pronounced impact on the performance of mcRigor for scATAC-seq data than for scRNA-seq data.

In this subsection, we assessed the impact.

Following the analysis described in the previous subsection, we applied mcRigor with $p = 100, 200, 300, 500, 1000, 1500, 2000, 2500, 3000, 3500, 4000, 4500,$ and 5000 to the scATAC-seq modality of a single-cell multiome dataset comprising 6,881 HSPCs [8]. Using the metacell partition generated by SEACells—the same partition used in the Results section “mcRigor empowers and rectifies gene regulatory inference on single-cell multiome data”—we evaluated, for each value of p , the same three mcRigor-derived metrics: the number of dubious metacells, the number of single cells comprising these dubious metacells, and the *Score*, across an extended range of γ values ($\gamma = 2, 3, \dots, 200$). Similar to the results for scRNA-seq data, these metrics remained stable for $p \geq 1000$ (Supplementary Fig 23b, top). To assess the adequacy of the default $p = 2000$, we computed the Jaccard indices between sets of dubious metacells identified at different values of p and observed strong concordance among results for large p ($p \geq 1000$) (Supplementary Fig 23b, bottom left). These findings confirm that mcRigor’s performance is robust as long as p is sufficiently large ($p \geq 1000$), and they support the use of $p = 2000$ as a reasonable default.

As in the previous subsection, the validity of the double permutation approach is further supported by the observation that the $\text{mcDiv}^{\text{null}}$ distribution exhibits the same trend as the mcDiv distribution as p increases (Supplementary Fig 23b, bottom middle and right), indicating that the effect of p on divergence score computation is effectively accounted for by the null distribution.”

Supplementary Fig 23: **Sensitivity analysis of the number of features (p) used on the bmcite scRNA dataset and the single-cell multiome dataset based on the SEACells method.** **a**, Sensitivity analysis on the bmcite scRNA dataset by examining the number of single cells that constitute dubious metacells (top left), the number of dubious metacells (top middle), the *Score* value (top right), the distribution of mcDiv (bottom middle), the distribution of mcDiv^{null} (bottom right) across different p values, and by measuring the similarity of dubious metacells found at different p values using Jaccard indices (bottom left). **b**, Sensitivity analysis on the scATAC modality of the single-cell multiome dataset using the same approaches.

Answers to Reviewer 2

Partitioning single cells into metacells is a meaningful steps in single cell data analysis. By defining metacells, we can pay greater attention to cellular heterogeneity. However, there is always a conflict between the homogeneity required by metacells and the actual heterogeneity of cells. Therefore, finding a balance between the two presents a significant challenge. To solve this issue, this study proposed a novel statistical method to enhance metacell selection, which could identify dubious metacells that are heterogeneous. It is an interesting point, which would be highly beneficial for large-scale single-cell data analysis. The results show it effectiveness in real scRNA-seq datasets. Here are some of my comments:

We thank the reviewer for recognizing the value of our work for single-cell data analysis.

Comment R2.1 mcRigor firstly calculates the metacell feature correlation matrix R using Pearson correlation, However, when the number of cells in this metacells is relatively small, does the calculation of Pearson correlation still hold statistical significance?

Answer to R2.1 We understand the reviewer’s question regarding the assessment of the statistical significance of Pearson correlation when the number of cells in a metacell is relatively small. Indeed, Pearson correlation estimates can exhibit higher variance with limited sample sizes, potentially affecting the stability of the resulting correlation matrix computed for each metacell. This is precisely why mcRigor adopts a permutation-based framework to assess significance, rather than relying on theoretically derived asymptotic distributions that assume large sample sizes. The permutation approach enables empirical calibration of significance thresholds for our defined mcDiv score, ensuring robust detection of dubious metacells even when the number of constituent cells is small.

Our approach was validated using the semi-synthetic dataset described in the main text, in which ground-truth cell state labels were available. As shown in Figure R1, mcRigor successfully classified small metacells (those containing fewer than 10 cells) with high accuracy across metacell partitions constructed by all three methods: SEACells, MetaCell, and SuperCell. For example, among the pure metacells generated by SEACells (i.e., metacells in which all single cells belonged to the same ground-truth cell state), 97.8% were correctly identified as trustworthy. Similarly, among the impure SEACells metacells, 71.4% were correctly flagged as dubious by mcRigor. Although the detection accuracy for small metacells is somewhat lower than that for all metacells (with corresponding accuracies of 99.9% and 85.8%, respectively), it remains substantially high and practically useful.

Figure R1: mcRigor identifies trustworthy and dubious metacells with high accuracy among small metacells (a) and metacells of all sizes (b).

Comment R2.2 The divergence score threshold parameter likely has a significant impact on the results. In selecting the divergence score threshold in Equation (5), the author mentions the bandwidth m_k . Why is this bandwidth used?

Answer to R2.2 We thank the reviewer for this insightful question, which highlights a key design choice in mcRigor. Ideally, the divergence score threshold would be computed individually for each metacell using null divergence scores obtained from many rounds of double permutation on that specific metacell. However, due to computational constraints, we aim to perform the double permutation only once per metacell. This necessitates pooling null scores from other metacells of the same size to estimate the threshold. Unfortunately, there are typically too few metacells of exactly the same size. For example, in the SEACells, MetaCell, and SuperCell partitions with $\gamma = 30$ on the `bmcite` dataset, the number of metacells of any given size usually ranges from 10 to 20 (Figure R2), which is insufficient for reliably determining a robust threshold.

To address this, we aggregate null divergence scores from metacells of similar sizes—a smoothing strategy commonly used to reduce variance and improve the stability of threshold estimation. Specifically, we define a bandwidth parameter, m_k , which determines the range of metacell sizes considered similar enough to share the same null distribution. We set $m_k = 10$, ensuring that at least 100 mcDiv null values are pooled for threshold estimation at each metacell size. While this approach introduces a trade-off between accuracy (by pooling across similar sizes) and computational efficiency (by requiring only one permutation per metacell), we believe the efficiency gained justifies this practical compromise.

Figure R2: **Distributions of metacell sizes for SEACells, MetaCell, and SuperCell partitions under $\gamma = 30$ for the `bmcite` dataset.** The number of metacells of any given size usually ranges from 10 to 20.

Comment R2.3 In section 2.4, the author said “...the trustworthy metacells identified by `mcRigor` exhibited significantly higher phase purity than the dubious metacells...”, however, the p-value for MetaCell in H2228 is 0.13904, and the authors need to provide an explanation for this non-significant result in order to offer guidance to users regarding the selection of the metacell method.

Answer to R2.3 We acknowledge the reviewer’s observation regarding the p-value of 0.13904 for the MetaCell partition in the H2228 cell line, which is “insignificant” and therefore may not appear to support our conclusion that trustworthy metacells exhibit significantly higher phase purity than dubious ones. However, this modest p-value is primarily due to the small-sample-size issue: the comparison involves only 8 dubious metacells versus 32 trustworthy ones—a small and imbalanced sample in which the Wilcoxon rank-sum test may lack sufficient power to yield statistical significance.

To further assess whether the 8 dubious metacells identified by `mcRigor` are indeed questionable, we closely examined each (`mc1` through `mc8`) in Figure R3. Manual inspection confirmed that all 8 metacells are biologically dubious. Specifically, 7 of them contained mixtures of single cells from different cell cycle phases (Figure R3a). The remaining metacell, `mc8`, was annotated as entirely G2M-phase based on Seurat’s `CellCycleScoring` function. However, as noted in the Seurat documentation, the continuous G2M and S scores provide more nuanced information than the discrete phase labels. When we visualized these scores for the constituent cells in `mc8`, we found that four cells (highlighted in red) had very low absolute values for both G2M and S scores (Figure R3b), suggesting they are likely non-cycling outliers rather than true G2M-phase cells [11]. This undermines the biological homogeneity of `mc8` and supports `mcRigor`’s classification of it as dubious.

Therefore, although the statistical comparison between dubious and trustworthy metacells yields a non-significant p-value—primarily due to the limited and imbalanced sample sizes—we believe that `mcRigor`’s identification of dubious metacells in the MetaCell partition remains bio-

logically justified and reasonable.

Figure R3: **Inspection of the biological impurity of dubious metacells detected by mcRigor within the MetaCell partition of the H2228 cell line.** **a**, Pie charts showing the cell cycle phase composition of each dubious metacell identified by mcRigor. Seven out of the eight metacells are biologically impure, with the exception of metacell mc8. **b**, Scatter plot showing the G2M and S Scores of all cells within metacell mc8. Red dots indicate cells with low absolute scores for both phases, suggesting they are likely non-cycling cells and raising concerns about the biological purity of mc8.

Comment R2.4 In fact, we observe that although there is a difference in cell cycle stage purity between trustworthy metacells and dubious metacells, what is the ratio of these two groups of metacells? If the proportions are similar, the violin plot reveals that many trustworthy metacells and dubious metacells have overlapping cell cycle stage purities. In practical applications, how significant is the identification of intra-cell-type heterogeneity in this context?

Answer to R2.4 We thank the reviewer for this thoughtful comment. While we are not entirely certain what is meant by “proportions are similar” in the second sentence—specifically, whether it refers to the ratio of trustworthy to dubious metacells mentioned earlier—we have done our best to respond based on our interpretation.

In our study, the ratio of dubious to trustworthy metacells indeed varies across cell lines and metacell partitioning methods. For instance, representative cases include ratios of 8 to 32, 6 to 22, and 20 to 143. To clarify this in the manuscript, we have now annotated the number of metacells in each group directly within the violin plots in Supplementary Fig 3 (Page 16), allowing for easier comparison.

As for the observed overlap in cell cycle stage purity between trustworthy and dubious metacells, we believe this may stem from two sources. First, the cell cycle phase annotations are not ground truth but computational predictions generated by Seurat’s `CellCycleScoring` function, and thus may be imperfect. Second, the classification of metacells as “dubious” or “trustworthy” is based

Supplementary Fig 3: **mcRigor’s trustworthy metacells reveal cell-cycle phases within cell lines.** **a**, Violin plots comparing the cell cycle-phase purity distributions of dubious metacells and trustworthy metacells. Trustworthy metacells consistently exhibit higher purity than dubious metacells.

on a thresholding procedure applied to the continuous mcDiv score. In our analysis, we used the 95th percentile of the null distribution as the default threshold. However, this threshold can be adjusted—raised for a more stringent definition of dubiousness or lowered for a more relaxed one. Therefore, some level of overlap between the two groups is expected and reflects both biological and methodological uncertainty.

Regarding the practical significance of identifying intra-cell-type heterogeneity, we believe it is substantial. A core strength of single-cell technologies is their ability to reveal subpopulations within annotated cell types—subpopulations that often correspond to functionally distinct and biologically meaningful rare cell states. For example, in the `bmcite` dataset, plasmablasts and hematopoietic stem cells (HSCs) are classified as subtypes within broader parent categories—B cells and progenitor cells, respectively (Supplementary Figure 14a–b, left). Although they collectively comprise less than 1.1% of the total cell population, these rare subtypes are critical to immune function and hematopoiesis. Accurately identifying such intra-cell-type heterogeneity is essential for downstream analyses and biological insight.

Additionally, as shown in our rare cell type analysis (suggested by Reviewer 3 and presented in Supplementary Fig 18, which is pasted on Page 23), the mcRigor two-step procedure—developed in response to Reviewer 1’s suggestion—was able to refine dubious metacells and recover trustworthy metacells corresponding to these rare subtypes. In contrast, standard metacell methods either failed to detect them or mixed them with unrelated cells. These results highlight the ability of mcRigor and its two-step extension to effectively identify and resolve intra-cell-type heterogeneity, confirming its practical utility in single-cell data analysis tasks.

Comment R2.5 It is evident that cellular heterogeneity is hierarchical, including different major cell types, subtypes, and even cell states. The authors could discuss the ability of mcRigor across different levels of heterogeneity to demonstrate its generalizability.

Answer to R2.5 We appreciate the reviewer’s insightful suggestion to discuss the generalizability of mcRigor across different levels of cellular heterogeneity. We have investigated the performance of mcRigor under varying cellular heterogeneity and included the analysis results in the Supplementary file, quoted below (with Supplementary Fig 14 shown on Page 18):

“Cellular heterogeneity is hierarchical, spanning broad cell types, finer subtypes, and distinct cell states. To evaluate mcRigor’s ability to operate across this hierarchy, we applied it to metacell partitions of the `bmcite` dataset generated by SEACells at varying granularity levels, enabling the detection of dubious metacells reflecting heterogeneity at different biological resolutions.

At a coarse granularity level ($\gamma = 90$), mcRigor successfully identified dubious metacells that mixed distinct major cell types—for example, metacell mc90-47, which included both T cells and progenitor cells (Supplementary Fig 14a). At a finer granularity level ($\gamma = 20$), it also flagged dubious metacells composed of closely related subtypes within the same major cell type—for instance, metacell mc20-9, which contained both CD8 memory T cells and CD4 naive T cells (Supplementary Fig 14b). In both cases, mcRigor exhibited high accuracy in detecting biologically impure metacells (Supplementary Fig 14c–d), demonstrating its robustness across levels of heterogeneity.

These findings support mcRigor’s generalizability and effectiveness in capturing both broad and subtle forms of cellular heterogeneity.”

We have also incorporated a discussion of this point into the revised Discussion section of the main text, quoted below:

“mcRigor is a novel statistical method designed to enhance the rigor of metacell partitioning in single-cell data analysis, ensuring reliable downstream analyses on metacells. By evaluating a given metacell partition, mcRigor identifies dubious metacells through quantifying per-metacell heterogeneity (i.e., the presence of mixed biological states) using a feature-correlation-based statistic. This statistic is assessed against a null distribution generated through a novel double permutation mechanism. Our findings demonstrate that the dubious metacells identified by mcRigor are indeed heterogeneous. Specifically, mcRigor can detect both dubious metacells that mix distinct major cell types and those composed of closely related subtypes within the same major type, capturing both broad and subtle forms of cellular heterogeneity (Supplementary section “Performance of mcRigor under varying cellular heterogeneity”; Supplementary Fig 14). Applications of mcRigor to real datasets show that removing dubious metacells detected by mcRigor significantly enhances downstream analyses, such as gene co-expression studies and enhancer-gene regulatory inference.”

Supplementary Fig 14: **mcRigor identifies dubious metacells across varying levels of cellular heterogeneity.** **a**, Single-cell UMAP plot showing the metacell partition generated by SEACells with $\gamma = 90$ and the corresponding dubious metacells identified by mcRigor (left), along with the single-cell composition of one representative dubious metacell, mc90-47 (right). **b**, Same as **a**, but for the SEACells partition with $\gamma = 20$, highlighting another example dubious metacell, mc20-9. **c**, Bar plot showing the number of biologically heterogeneous metacells correctly identified by mcRigor as dubious versus those not flagged, in the SEACells $\gamma = 90$ partition. **d**, Same as **c**, but for the SEACells partition with $\gamma = 20$.

Comment R2.6 This article discusses that many of the comparisons and analyses focus on the issue of biological zero, and many imputation methods also solve this problem. Therefore, if imputation is performed prior to constructing metacells, could this significantly reduce the number of dubious metacells?

Answer to R2.6 We thank the reviewer for raising this interesting point. We agree that imputation provides an alternative strategy for addressing non-biological zeros while aiming to preserve biologically meaningful signal. However, this is also the goal of metacell partitioning. If imputation can fully resolve the issue of technical sparsity, then applying metacell partitioning afterward may become redundant. In this sense, we view metacell partitioning and imputation as alternative, rather than sequential, approaches to denoising sparse single-cell data.

In addition, prior studies have shown that imputation methods can introduce bias into the data [4], potentially distorting a cell’s true in-situ gene expression profile. This distortion may lead to artificial cell states and, paradoxically, could increase the number of dubious metacells rather than reduce them. We have added a discussion of this point to the revised Discussion section of the main text, quoted below:

“As mentioned in the Introduction, both metacell partitioning and imputation aim to address data sparsity, and we consider them alternative—not sequential—strategies. Metacell partitioning reduces technical zeros by aggregating similar cells into metacells, thereby averaging out noise, whereas imputation attempts to recover technical zeros at the level of individual cells. In response to the question of whether applying imputation before metacell construction might better resolve sparsity, our view is that doing so would be redundant and potentially counterproductive. Imputation has been shown to introduce biases into the data [4], which may distort the underlying gene expression landscape and lead to the formation of spurious metacell groupings—ultimately increasing the number of dubious metacells rather than reducing them. Therefore, we recommend choosing either metacell partitioning or imputation based on the goals of downstream analysis, rather than applying both in combination.”

Answers to Reviewer 3

The study introduces mcRigor, a statistical framework designed to evaluate whether a metacell comprises a sufficiently homogeneous population of cells. The method operates under the assumption that a properly constructed metacell consists of cells in the same biological state, with the variability within the metacell attributable only to technical noise and not biological differences. Central to mcRigor is the hypothesis that, in an ideal metacell, the correlation matrix of the metacell feature approximates an identity matrix due to the absence of biological variance. Based on this premise, the authors define a metacell divergence score (mcDiv) to quantify the extent of deviation from this ideal state. Furthermore, mcRigor optimizes metacell partitioning by balancing two metrics: DubRate (a measure of dubious cell inclusion) and ZeroRate (a reflection of the sparsity of data). Through extensive analyses, the authors demonstrate the efficacy of mcRigor in identifying dubious cells and the optimal number of metacells in various datasets. However, the framework is primarily a post-processing step applied after the construction of metacells by existing methods, which somewhat limits its originality. Rather than simply excluding dubious cells, an approach that reconstructs metacells directly could potentially enhance the robustness and utility of the method.

We sincerely thank the reviewer for the detailed summary of our manuscript and the insightful comments. We agree that mcRigor functions as an add-on framework, applied after metacell partitioning by existing methods. While its performance naturally depends on the quality of the input metacell partition, this add-on design confers an important advantage: users retain the flexibility to select the metacell method most appropriate for their dataset, while mcRigor provides an independent statistical layer to evaluate and enhance the trustworthiness of the resulting metacells—complementing, rather than replacing, existing approaches.

We also appreciate the reviewer’s point that when a large number of dubious metacells are detected—indicating poor-quality input partitions—simply removing these metacells may lead to substantial information loss. In such cases, reconstructing improved metacells is preferable. In response to a related suggestion from Reviewer 1, we developed an extension of mcRigor, termed *mcRigor two-step*, to address this concern. Rather than discarding dubious metacells, mcRigor two-step re-applies metacell partitioning to their constituent cells at a finer granularity. As demonstrated in our revised Supplementary file and in our **Answer to R1.1** (Page 2), this extension improves downstream analyses and enhances the resolution of rare cell types. While mcRigor two-step is a heuristic and relatively simple approach, its strong empirical performance highlights mcRigor’s utility not only as a practical diagnostic tool but also as a conceptual foundation for developing more principled metacell partitioning strategies in future work.

Comment R3.1 In the metacell approach, clusters are formed based on γ , and each cluster is averaged to create a metacell. During this process, “dubious” metacells—those containing heterogeneous cells—may emerge, which mcRigor subsequently removes. A potential concern is that small clusters with rare cells ($\gamma^k \ll \gamma$) are prone to merging into larger clusters, resulting in heterogeneous metacells that are flagged and removed as dubious. Consequently, rare cells might be lost because mcRigor labels them as dubious. To address this, the authors should present experiments examining how mcRigor handles rare cell types. In particular, how well does mcRigor identify rare cell states? A systematic evaluation is necessary to ensure that rare cell types are not inadvertently removed due to their inclusion in dubious metacells.

Answer to R3.1 We thank the reviewer for this insightful comment. We agree that directly removing dubious metacells may result in the unintended loss of rare cell states. To address this concern, and in response to Reviewer 1’s Comment 1, we propose a two-step extension of mcRigor, termed *mcRigor two-step*. In this approach, the single cells comprising dubious metacells are re-partitioned using a metacell method with a smaller granularity level (denoted by γ_2), followed by a reapplication of mcRigor to identify dubious metacells within this refined partition. This strategy enables more granular segmentation in low-density or transitional regions of the phenotypic space, thereby improving the resolution of rare or subtle cell states. Full details of mcRigor two-step, along with its application to two datasets, are provided in our revised Supplementary file and in our **Answer to R1.1**. We have also added a new subsection to the revised Supplementary file titled “Capability of mcRigor two-step to resolve rare cell types,” quoted below.

Quote from the Supplementary file (with Supplementary Fig 18 shown on Page 23 and Supplementary Fig 19 on Page 24): “To systematically assess the ability of mcRigor two-step to capture rare cell types, we analyzed an scRNA-seq dataset of bone marrow mononuclear cells profiled by CITE-seq (the `bmcite` dataset [9]) used in our main text, which contains two rare cell types: plasmablasts and HSCs, constituting 0.8% and 1.1% of the total cell population, respectively (Supplementary Fig 18a). Specifically, we first applied SEACells, SuperCell, and MetaCell with $\gamma = 50$ to generate baseline metacell partitions, and then applied mcRigor and mcRigor two-step to evaluate and refine each partition. To quantify whether a rare cell type was well captured, we examined the number and composition of trustworthy metacells in which at least 50% of the constituent single cells belonged to the rare cell type of interest (Supplementary Fig 18b-d). A rare cell type was considered well captured if it was represented by a sufficient number of such trustworthy metacells.

Notably, in the initial MetaCell partition, no trustworthy metacells representing plasmablasts or HSC cells were identified (Supplementary Fig 18d). The original SuperCell partition contained only two very small trustworthy plasmablast metacells and none for HSCs (Supplementary Fig 18c), failing to reliably capture either rare cell type. After applying mcRigor two-step, selecting

$\gamma_2 = 22$ for SuperCell and $\gamma_2 = 20$ for MetaCell, several trustworthy metacells representing each rare cell type emerged (Supplementary Fig 18c-d), demonstrating mcRigor two-step’s effectiveness in resolving rare cell types. For the SEACells partition, which already contained trustworthy metacells for both plasmablasts and HSCs, mcRigor two-step further increased the number of such metacells (Supplementary Fig 18b), confirming its utility in enhancing rare cell resolution.

To evaluate the capability of mcRigor two-step to resolve even rarer cell types, we downsampled HSCs from 1.1% to 0.5% of the total cell population while retaining all other cells (Supplementary Fig 19a) and repeated the above analysis on this modified dataset. Compared to the original `bmcite` dataset, mcRigor two-step selected smaller γ_2 values for all three metacell methods on the downsampled data (Supplementary Fig 19b–d), which is intuitive since finer granularity may be required to resolve less abundant cell types. As with the original dataset, trustworthy metacells representing plasmablasts and HSCs were recovered across all three methods (Supplementary Fig 19b–d). However, we observed a reduction in the number of trustworthy HSC metacells identified by mcRigor two-step, which is expected due to the smaller pool of available HSCs. These results suggest that mcRigor two-step remains effective at resolving rare cell types even at lower abundance, though its resolving power becomes increasingly challenged as the rare cell type frequency decreases. Nevertheless, mcRigor two-step offers flexibility for further extension via iterative re-partitioning of remaining dubious metacells, ultimately down to single-cell resolution if necessary. This iterative refinement provides a promising strategy for resolving even the rarest cell states without prematurely discarding them.”

Comment R3.2 How does mcRigor perform when applied across multiple integrated cohorts? SEACells has demonstrated its ability to preserve biological signals despite substantial technical noise—for instance, by analyzing CD4+ T cells from multiple cohorts in Figure 6 of the SEACells paper. The question remains whether mcRigor can further enhance these outcomes.

Answer to R3.2 We thank the reviewer for this insightful suggestion to further validate mcRigor’s performance. In response, we tested mcRigor on a subset of an scRNA-seq dataset used in the SEACells study [8] and demonstrated that mcRigor improves integration analysis on this subset.

Quote from the Results section in the main text (with Supplementary Fig 12 shown on Page 27 and Supplementary Fig 13 on Page 28): “To evaluate the effectiveness of mcRigor in improving integration analysis, we reanalyzed a subset of the scRNA-seq dataset used in the SEACells study [8], comprising 96,466 peripheral blood mononuclear cells (PBMCs) from 10 healthy donors and 10 patients with critical Coronavirus Disease 2019 (COVID-19). We focused on this data subset rather than the full dataset because, in large datasets, the impact of poor-quality metacells may be negligible. In contrast, when working with fewer cells, high-quality

Supplementary Fig 18: **Capability of *mcRigor two-step* to resolve rare cell types demonstrated using the bmcite dataset.** **a**, UMAP plot of the bmcite dataset with the two rare cell types—plasmablasts and HSCs—highlighted. **b–d**, Demonstration of *mcRigor two-step*’s ability to recover these rare cell types when applied to metacell partitions generated by SEACells (**b**), SuperCell (**c**), and MetaCell (**d**). Top: Single-cell UMAP plots showing the metacell partitions produced by the original method (left) and by *mcRigor two-step* (right), with dubious metacells identified by either *mcRigor* or *mcRigor two-step* highlighted in red circles. Bottom: Bar plots showing the cell-type composition of trustworthy metacells in which at least 50% of the constituent cells belong to the corresponding rare cell type. The number of bars reflects the number of such trustworthy metacells (*mcRigor*: left, *mcRigor two-step*: right).

Supplementary Fig 19: **Capability of mcRigor two-step to resolve rare cell types demonstrated using the *bmcite* dataset with HSC cells downsampled.** **a**, UMAP plot of the *bmcite* dataset with the two rare cell types—plasmablasts and HSCs—highlighted. **b–d**, Demonstration of mcRigor two-step’s ability to recover these rare cell types when applied to metacell partitions generated by SEACells (**b**), SuperCell (**c**), and MetaCell (**d**). Top: Single-cell UMAP plots showing the metacell partitions produced by the original method (left) and by mcRigor two-step (right), with dubious metacells identified by either mcRigor or mcRigor two-step highlighted in red circles. Bottom: Bar plots showing the cell-type composition of trustworthy metacells in which at least 50% of the constituent cells belong to the corresponding rare cell type. The number of bars reflects the number of such trustworthy metacells (mcRigor: left, mcRigor two-step: right).

metacell partitioning becomes more critical, as a small number of dubious metacells may have a large distortion effect on biological signals if the total number of metacells is small. We compared SEACells metacells constructed using the default granularity level ($\gamma_{\text{org}} = 75$) with those optimized by mcRigor ($\gamma_{\text{opt}} = 49$), hereafter referred to as mcRigor metacells, and performed Harmony [6] integration using the resulting metacell expression profiles. In the SEACells study, the authors demonstrated that CD4 T cells from three different collection sites exhibit meaningful biological differences, and that such variation should be preserved, not eliminated, during integration [8]. Indeed, in the subset we analyzed, SEACells CD4T metacells constructed at the default granularity exhibited reduced similarities across different sites (mLISI, the mean Local Inverse Simpson’s Index that measures batch similarities, reduced from 1.528 for single cells to 1.474 for metacells based on *collection site*) (Supplementary Fig 12, bottom middle). However, the distinction between CD4 and CD8 T metacells became less pronounced after integration (mLISI increased from 1.063 for single cells to 1.152 for metacells based on *cell type*) (Supplementary Fig 12, top middle). In contrast, the mcRigor metacells, compared with the unrefined SEACells metacells, better enhanced both site-specific differences (mLISI reduced from 1.528 for single cells to 1.344 for metacells based on *collection site*) and maintained the separation between CD4 and CD8 T cells (mLISI reduced from 1.063 for single cells to 1.028 for metacells based on *cell type*) (Supplementary Fig 12, left). This suggests that data-driven granularity optimization, rather than a fixed heuristic granularity level, can be essential for robust biological signal recovery during integration.

We next investigated whether mcRigor metacells could more effectively reveal T cell response dynamics in COVID-19 using the integrated data. Following the analysis in the SEACells study [8], we further aggregated each set of metacells (SEACells metacells and mcRigor metacells) into second-level meta2cells, each consisting of 10 metacells, by reapplying SEACells to their Harmony-corrected low-dimensional embeddings. From each meta2cell set, we then selected three representative meta2cells corresponding to early, middle, and late stages after COVID-19 onset (Supplementary Fig 13a–b). Both SEACells and mcRigor meta2cells captured temporal shifts in immune gene expression, including type I interferon-stimulated genes (*IRF7*, *IRF9*, *ISG15*, *IFITM1*), inflammation-regulating genes (*CCR10*, *FOXP3*, *IL2RA*), and hallmark T₁₇-related genes indicative of a transition toward type III inflammation (*RORC*, *CCR6*). Notably, mcRigor meta2cells exhibited more distinct temporal expression trajectories, especially for genes such as *IFITM1*, *FOXP3*, and *CCR6* (Supplementary Fig 13a). These results suggest that mcRigor’s data-driven granularity optimization helps reveal the dynamic immune responses over the course of disease progression.

To further investigate the temporal dynamics of CD4 T cell responses, we extended our analysis to construct trajectories using all CD4 meta2cells, beyond the three representative ones in each set (SEACells and mcRigor meta2cells). Following a trajectory construction approach from

a glioblastoma immune profiling study [5], we first clustered each set of meta2cells based on their Harmony [6] batch-corrected low-dimensional embeddings, varying the number of clusters $k \in [3, \dots, K]$, where K is the total number of CD4 meta2cells. For each k , we constructed a trajectory by ordering the meta2cell clusters based on their average time since disease onset, and then averaged the resulting $(K - 2)$ trajectories to generate a final temporal trajectory. Along this trajectory, we computed smoothed gene expression profiles, revealing progressive activation of immune gene modules over the course of COVID-19 infection (Supplementary Fig 13c). Notably, trajectories based on mcRigor meta2cells showed clearer temporal progression and more coherent gene module expression patterns (Supplementary Fig 13c, left), aligning more closely with both single-cell-level observations (Supplementary Fig 13a, right) and established biological knowledge. For example, *KLRG1*—a gene known to increase in CD4 T cells during adaptive immune responses [1]—exhibited a consistent upward trend in the mcRigor-based trajectory but a reversed trend in the SEACells-based one. These results demonstrate that mcRigor’s granularity optimization not only enhances data integration but also improves the reconstruction of temporal immune processes.”

Comment R3.3 In Fig. 1e, “trustworthy metacells” effectively distinguished healthy samples from those with COVID-19. However, a question arises whether “all metacells” could achieve similar performance by adjusting certain parameters, such as γ . For example, constructing metacells under stricter conditions (e.g., using a lower γ) might yield high-quality metacells even without mcRigor, although this could increase the number of zero counts. Moreover, single-cell analysis alone yielded performance comparable to mcRigor, indicating that zero counts do not substantially hinder the analysis.

Answer to R3.3 We thank the reviewer for raising this insightful question regarding whether stricter metacell construction—using a lower granularity level γ without subsequent filtering of dubious metacells—could replicate the performance of mcRigor. To investigate this, we compared two settings: SEACells with a small $\gamma = 5$ (rigid setting, no mcRigor; Figure R5a, right) and SEACells with a larger $\gamma = 30$ followed by mcRigor filtering (flexible setting, with mcRigor; Figure R5a, middle). Although the small- γ setting may yield metacells of higher homogeneity, it failed to fully address the data sparsity issue, which weakens the biological signal in downstream analyses. Specifically, while SEACells($\gamma = 5$) without mcRigor helped identify enrichment of co-expression for the adaptive immune response module in COVID-19 compared to healthy controls (p-value = 0.02138 for one-sided Wilcoxon rank-sum test), the significance was notably weaker than that obtained using SEACells($\gamma = 30$) + mcRigor (p-value = 1.4e-13). The enrichment signal was also less visually apparent in the gene-gene correlation matrix (Figure R5a). A similar trend was

Supplementary Fig 12: **mcRigor’s optimized metacell partition improves integration fidelity.** **a**, UMAP plots of integrated data after batch correction using Harmony [6], showing mcRigor + SEACells metacells (left), SEACells default metacells (middle), and single cells (right), colored by cell type. Integration based on mcRigor + SEACells metacells better preserves the separation between CD4 and CD8 T cells, as reflected by a lower mLISI value based on *cell type* (1.028) compared to single cells (1.063) and SEACells default metacells (1.152). mLISI (mean Local Inverse Simpson’s Index) is computed from the Harmony-corrected low-dimensional embedding; a higher mLISI indicates greater mixing across batches (e.g., cell types or collection sites), and a lower value reflects better separation. **b**, Same as (a), but with points colored by sample collection site instead of cell type. Integration based on mcRigor + SEACells metacells better preserves between-site biological differences, as indicated by the decreased mLISI value based on *collection site* (1.344), compared to single cells (1.528) and SEACells default metacells (1.474).

observed when comparing SuperCell($\gamma = 5$) without mcRigor and SuperCell($\gamma = 30$) + mcRigor (Figure R5b).

More importantly, these results underscore that using small metacells or unaggregated single cells (Figure R5, left) can compromise statistical power due to technical noise and data sparsity. Such noise and sparsity can inflate the p-values of biologically meaningful signals, reducing confidence in gene module rankings and potentially leading to misleading experimental follow-up. While metacell partitioning with a small γ may lower the number of dubious metacells, it does not match the overall performance of mcRigor, which more effectively balances data sparsity and signal distortion.

Supplementary Fig 13: **mcRigor’s optimized metacell partition better captures T cell response dynamics.** **a**, Heatmaps showing gene expression patterns across disease stages, derived from three representative CD4 T meta2cells constructed from mcRigor + SEACells metacells (left), SEACells default metacells (middle), and pseudobulk profiles based on aggregated single cells (right), all in the Harmony-corrected embedding space. Meta2cells from mcRigor + SEACells representing early, middle, and late stages are labeled m1, m2, and m3, respectively; those from SEACells default are labeled s1, s2, and s3. **b**, UMAP plots of CD4 T meta2cells generated from mcRigor + SEACells (left) and SEACells default (right), colored by days since disease onset. In each plot, the three representative meta2cells corresponding to early, middle, and late stages are highlighted. **c**, Smoothed expression trajectories of representative immune marker genes along CD4 T meta2cells derived from mcRigor + SEACells (left) and SEACells default (right), demonstrating clearer temporal dynamics in the mcRigor-based results.

Comment R3.4 In Section 2.6, mcRigor was applied to single-cell multiome data. The authors should compare results from a “rigid” setting—producing many metacells—to a “flexible” setting—producing fewer metacells, then using mcRigor to filter out dubious ones. In my view, using a rigid setting (e.g., lowering γ) can reduce dubious metacells, albeit at the cost of increased sparsity. However, with 2,000 highly variable genes, this sparsity would likely not be severe. Most importantly, the authors must demonstrate that mcRigor’s utility cannot be replicated simply by tuning existing parameters such as γ . If it can, the novelty of mcRigor is undermined.

Figure R5: **Comparison of gene co-expression from small- γ metacell construction without mcRigor versus large- γ construction with mcRigor.** **a**, Gene-gene correlation matrices for three key gene modules under COVID-19 and healthy control conditions, based on three data types: single cells (left), trustworthy metacells by SEACells($\gamma = 30$) + mcRigor (middle), and all metacells by SEACells($\gamma = 5$) (right). Each p-value comparing gene-gene correlations within the adaptive immune response gene module between the two conditions, for each data type, was computed using a one-sided Wilcoxon rank-sum test. **b**, Same as **(a)**, but showing the results based on SuperCell.

Answer to R3.4 We thank the reviewer for the helpful suggestion. In response, we compared the results from a fine-grained metacell partition (rigid setting: SEACells with $\gamma = 5$) to those from a coarse-grained partition (flexible setting: SEACells with $\gamma = 90$) followed by mcRigor filtering (Supplementary Fig 6).

Quote from the Results section in the main text (with Supplementary Fig 6 shown on Page 30): “Intuitively, using a low granularity level may reduce the number of dubious metacells and yield reliable results even without mcRigor. However, this strategy often leaves sparsity unresolved, compromising statistical power in downstream analyses. To illustrate this, we compared results from a fine-grained metacell partition (SEACells with $\gamma = 5$, without mcRigor) to those from a coarse-grained partition (SEACells with $\gamma = 90$) followed by mcRigor filtering (Supplementary Fig 6). We observed clear advantages with the SEACells ($\gamma = 90$) + mcRigor partition, which provided greater statistical power and identified more biologically supported enhancer–gene associations. For example, the enhancer *LOC117038771*, previously reported to regulate *GATA2* [2], was detected only with the trustworthy metacells from SEACells ($\gamma = 90$) + mcRigor, but not with SEACells ($\gamma = 5$) (Supplementary Fig 6a). Similarly, for *TAL1*, multiple HCPs were identified using SEACells ($\gamma = 90$) + mcRigor, but none were detected using SEACells ($\gamma = 5$) or single-cell data, where sparsity limited detection power (Supplementary Fig 6b). These findings demonstrate that simply lowering γ is insufficient and that mcRigor is essential for improving statistical power while maintaining the reliability of metacell-based regulatory analyses.”

Supplementary Fig 6: **mcRigor improves gene regulatory inference and provides greater statistical power than metacell partitioning at low granularity.** **a**, Highly correlated peaks for the gene *GATA2* identified using trustworthy metacells from a coarse-grained partition (SEACells with $\gamma = 90$ + mcRigor), all metacells from a fine-grained partition (SEACells with $\gamma = 5$), or single cells. **b**, Same as **a**, but for the gene *TAL1*.

Regarding data sparsity, we examined the proportion of zeros in both the original single-cell profiles and the aggregated metacell profiles across different feature selection thresholds (Table R1).

Notably, in scATAC-seq data analysis, it is uncommon to retain only the top 2,000 highly variable (HV) features. Instead, practitioners typically retain the top 25% most frequently observed features—or even up to 95%—as implemented in the `FindVariableFeatures` function of `Signac`, due to the high dimensionality and low signal density of scATAC-seq data. In our scMultiome dataset, the top 25% and top 95% most frequently observed peaks correspond to 61,549 and 246,113 features, respectively. Even with these large feature sets, SEACells metacells at $\gamma = 5$ exhibit high sparsity, with 68.51% and 87.65% zero entries, respectively. This persistent sparsity likely contributes to the reduced statistical power observed under the small- γ setting in the preceding analysis (Supplementary Fig 6).

More importantly, we wish to emphasize that, to the best of our knowledge, there is currently no established method in the literature for tuning hyperparameters in metacell partitioning. Yet, as discussed in the Introduction, different hyperparameter choices—even within the same metacell method—can lead to substantially different results. Therefore, hyperparameter selection is a critical step for ensuring the reliability of metacell-based analyses, and addressing this challenge is one of the core functionalities offered by mcRigor.

Table R1: Proportions of zeros in the single-cell and metacell profiles (obtained using SEACells) under different numbers of retained features. Each number in parentheses indicates the number of features retained.

	single cells	metacells with $\gamma = 5$	metacells with $\gamma = 90$
all features (246,113)	97.09%	88.22%	47.19%
top 95% most frequently observed features (233,871)	96.95%	87.65%	45.05%
top 25% most frequently observed features (61,549)	91.43%	68.51%	12.68%
top 2,000 highly variable features	69.82%	25.54%	0.30%

Comment R3.5 In Section 2.10, to demonstrate that mcRigor’s optimized metacell partition better reveals temporal immune cell trajectories, the authors compared MetaCell (from the original study) with SEACells followed by mcRigor. However, it remains unclear whether the improvement is due to mcRigor or SEACells itself. To clarify this, the authors should provide direct comparisons of each method with and without mcRigor—for example, MetaCell vs. MetaCell + mcRigor, and SEACells vs. SEACells + mcRigor.

Answer to R3.5 We agree with the reviewer that directly comparing each metacell method with and without mcRigor provides a fairer assessment of mcRigor’s contribution. Following this suggestion, we compared MetaCell + mcRigor with MetaCell alone (as used in the original

study), and SEACells + mcRigor (SEACells being the overall best-performing method selected by mcRigor) with SEACells alone (Supplementary Fig 11). These comparisons demonstrate that mcRigor itself contributes meaningful improvements to trajectory analysis.

Quote from the Results section in the main text (with Supplementary Fig 11 shown on Page 33): “To explicitly distinguish the contribution of mcRigor from that of the underlying metacell methods themselves, we further conducted direct comparisons between the results obtained from each method with and without mcRigor—comparing MetaCell + mcRigor with MetaCell alone (as used in the original study), and SEACells + mcRigor (SEACells being the overall best-performing method selected by mcRigor) with SEACells alone (Supplementary Fig 11). In both comparisons, incorporating mcRigor led to noticeable improvements. For instance, the MetaCell + mcRigor partition showed a greater cTET difference between the earliest- and latest-stage metacells (0.600) compared to MetaCell alone (0.538), as well as a stronger correlation between cTET values and the transitional stages of metacells (Spearman’s rank correlation $\rho = 0.961$ vs. 0.954, Supplementary Fig 11a). Furthermore, adding mcRigor corrected questionable temporal expression patterns initially inferred by MetaCell alone, resulting in biologically consistent upregulation trends for genes including *Lag3*, *Tubb5*, *Tk1*, and *Clspn* (Supplementary Fig 11b). Consistent with these findings, the single-cell DE genes exhibited stronger correlations with tumor exposure time and lower adjusted p-values at the metacell level when using MetaCell + mcRigor compared to MetaCell alone (Supplementary Fig 11c). Similar improvements were observed when comparing SEACells + mcRigor to SEACells alone (Supplementary Fig 11), confirming that the enhanced downstream analysis results—reflected by greater cTET distinctions and more biologically plausible gene temporal expression patterns—are indeed attributable to the use of mcRigor.

These results further underscore mcRigor’s ability to identify the most suitable metacell method for a given dataset—not only by optimizing the granularity level within each method, but also by selecting the most appropriate method. In this case, mcRigor selected SEACells over MetaCell (Supplementary Fig 11), and accordingly, SEACells combined with mcRigor outperformed MetaCell alone, as our results showed.”

Supplementary Fig 11: Comparison of analysis results from four metacell partitions—SEACells + mcRigor (mcRigor-selected), SEACells, MetaCell + mcRigor, and MetaCell (original study)—on the Zman-seq dataset. **a**, Line plots of continuous tumor exposure time (cTET) values computed from the four metacell partitions. The size of $\Delta cTET$, indicated by the light blue shaded area, reflects the resolution of tumor transitional stages. **b**, Smoothed expression profiles of four representative marker genes derived from each metacell partition. **c**, DE genes identified at the single-cell level exhibited stronger correlations with tumor exposure time and lower p-values when using metacells from SEACells + mcRigor or MetaCell + mcRigor, compared to SEACells or MetaCell alone.

References

- [1] Ager, C.R., Zhang, M., Chaimowitz, M., Bansal, S., Tagore, S., Obradovic, A., Jugler, C., Rogava, M., Melms, J.C., McCann, P., et al.: Klr_g1 marks tumor-infiltrating cd4 t cell subsets associated with tumor progression and immunotherapy response. *Journal for Immunotherapy of Cancer* **11**(9), e006782 (2023)
- [2] Fishilevich, S., Nudel, R., Rappaport, N., Hadar, R., Plaschkes, I., Iny Stein, T., Rosen, N., Kohn, A., Twik, M., Safran, M., et al.: Genehancer: genome-wide integration of enhancers and target genes in genecards. *Database* **2017**, bax028 (2017)
- [3] Grün, D., Kester, L., Van Oudenaarden, A.: Validation of noise models for single-cell transcriptomics. *Nature Methods* **11**(6), 637–640 (2014)
- [4] Jiang, R., Li, W.V., Li, J.J.: mbimpute: an accurate and robust imputation method for microbiome data. *Genome Biology* **22**(1), 192 (2021)
- [5] Kirschenbaum, D., Xie, K., Ingelfinger, F., Katzenelenbogen, Y., Abadie, K., Look, T., Sheban, F., San Phan, T., Li, B., Zwicky, P., et al.: Time-resolved single-cell transcriptomics defines immune trajectories in glioblastoma. *Cell* **187**(1), 149–165 (2024)
- [6] Korsunsky, I., Millard, N., Fan, J., Slowikowski, K., Zhang, F., Wei, K., Baglaenko, Y., Brenner, M., Loh, P.r., Raychaudhuri, S.: Fast, sensitive and accurate integration of single-cell data with harmony. *Nature Methods* **16**(12), 1289–1296 (2019)
- [7] Morabito, S., Reese, F., Rahimzadeh, N., Miyoshi, E., Swarup, V.: hdwgcna identifies co-expression networks in high-dimensional transcriptomics data. *Cell Reports Methods* **3**(6), 100498 (2023)
- [8] Persad, S., Choo, Z.N., Dien, C., Sohail, N., Masilionis, I., Chaligné, R., Nawy, T., Brown, C.C., Sharma, R., Pe’er, I., et al.: Seacells infers transcriptional and epigenomic cellular states from single-cell genomics data. *Nature Biotechnology* **41**(12), 1746–1757 (2023)
- [9] Stuart, T., Butler, A., Hoffman, P., Hafemeister, C., Papalexi, E., Mauck, W.M., Hao, Y., Stoeckius, M., Smibert, P., Satija, R.: Comprehensive integration of single-cell data. *Cell* **177**(7), 1888–1902 (2019)
- [10] Su, C., Xu, Z., Shan, X., Cai, B., Zhao, H., Zhang, J.: Cell-type-specific co-expression inference from single cell rna-sequencing data. *Nature Communications* **14**(1), 4846 (2023)

- [11] Tirosh, I., Izar, B., Prakadan, S.M., Wadsworth, M.H., Treacy, D., Trombetta, J.J., Rotem, A., Rodman, C., Lian, C., Murphy, G., et al.: Dissecting the multicellular ecosystem of metastatic melanoma by single-cell rna-seq. *Science* **352**(6282), 189–196 (2016)
- [12] Torre, E., Dueck, H., Shaffer, S., Gospic, J., Gupte, R., Bonasio, R., Kim, J., Murray, J., Raj, A.: Rare cell detection by single-cell rna sequencing as guided by single-molecule rna fish. *Cell Systems* **6**(2), 171–179 (2018)
- [13] Wilk, A.J., Rustagi, A., Zhao, N.Q., Roque, J., Martínez-Colón, G.J., McKechnie, J.L., Ivison, G.T., Ranganath, T., Vergara, R., Hollis, T., et al.: A single-cell atlas of the peripheral immune response in patients with severe covid-19. *Nature Medicine* **26**(7), 1070–1076 (2020)

Response to Reviewers' Comments on “mcRigor: a statistical method to enhance the rigor of metacell partitioning in single-cell data analysis”

We sincerely thank the three reviewers for their thoughtful evaluation of our manuscript and their constructive and encouraging feedback. We have carefully addressed all comments and revised the manuscript accordingly.

Our detailed, point-by-point responses to the reviewers' comments are provided on the following pages. Reviewers' comments appear in **mahogany**, our responses are in **black**, and changes made to the revised manuscript and supplementary materials are highlighted in **blue** and quoted in this response letter.

Answers to Reviewer 3

The mcRigor two-step approach appears to offer a promising solution for the precise detection of dubious metacells. However, the descriptions of Supplementary Figures 18 and 19 require further clarification. In particular, the distinction between the results produced by mcRigor and those obtained with the mcRigor two-step method is not sufficiently clear. Providing detailed information on the number of trustworthy and dubious metacells identified per cell type, as well as a clear explanation of the significance of the brown area in the bar plots, would greatly enhance the interpretability of these results.

We thank the reviewer for recognizing the value of our proposed mcRigor two-step approach and for the helpful suggestions to improve our manuscript. To enhance clarity, we have added detailed information on the numbers of trustworthy and dubious metacells identified by mcRigor and mcRigor two-step, respectively, in Supplementary Figures 18 and 19 (pasted below), thereby making the distinction between the results of the two approaches clearer. We have also included additional explanations in the supplementary file regarding this distinction and the significance of the brown areas in the bar plots.

Quote from the Supplementary file (with Supplementary Fig 18 shown on Page 4 and 19 on Page 5): “Notably, in the initial MetaCell partition, no trustworthy metacells representing plasmablasts or HSC cells were identified—the two plasmablast metacells and the two HSC metacells were all marked as dubious (Supplementary Fig 18d). The original SuperCell partition contained only two very small trustworthy plasmablast metacells (the other three plasmablast metacells were marked as dubious) and none for HSCs (all nine HSC metacells were marked as dubious; Supplementary Fig 18c), failing to reliably capture either rare cell type.

After applying mcRigor two-step, selecting $\gamma_2 = 22$ for SuperCell and $\gamma_2 = 20$ for MetaCell, several trustworthy metacells representing each rare cell type emerged (Supplementary Fig 18b-c), demonstrating mcRigor two-step’s effectiveness in resolving rare cell types. For the SEACells partition, which already contained trustworthy metacells for both plasmablasts and HSCs, mcRigor two-step further increased the numbers of trustworthy metacells while reducing or maintaining the numbers of dubious metacells (e.g., from 3 trustworthy and 3 dubious plasmablast metacells to 12 trustworthy and 1 dubious plasmablast metacell; Supplementary Fig 18d), confirming its utility in enhancing rare cell resolution. We also observed that some of the trustworthy HSC metacells included not only HSCs but also single cells annotated as Lympho-Myeloid Primed Progenitors (LMPPs; brown areas in the bar plots in Supplementary Fig 18b–d), which could reflect both the similarity in gene expression profiles between HSCs and LMPPs and potential inaccuracies in cell type annotation.

To evaluate the capability of mcRigor two-step to resolve even rarer cell types, we downsampled HSCs from 1.1% to 0.5% of the total cell population while retaining all other cells (Supplementary

Fig 19a) and repeated the above analysis on this modified dataset. Compared to the original `bmcite` dataset, mcRigor two-step selected smaller γ_2 values for all three metacell methods on the downsampled data (Supplementary Fig 19b–d), which is intuitive since finer granularity may be required to resolve less abundant cell types. As with the original dataset, mcRigor two-step recovered trustworthy metacells representing plasmablasts and HSCs across all three methods, increasing their numbers while ensuring that the number of dubious metacells did not rise (Supplementary Fig 19b–d).”

Supplementary Fig 18: **Capability of *mcRigor two-step* to resolve rare cell types demonstrated using the bmcite dataset.** **a**, UMAP plot of the bmcite dataset with the two rare cell types—plasmablasts and HSCs—highlighted. **b–d**, Demonstration of *mcRigor two-step*’s ability to recover these rare cell types when applied to metacell partitions generated by SEACells (b), SuperCell (c), and MetaCell (d). Top: Single-cell UMAP plots showing the metacell partitions produced by the original method (left) and by *mcRigor two-step* (right), with dubious metacells identified by either *mcRigor* or *mcRigor two-step* highlighted in red circles. The numbers of trustworthy and dubious metacells representing plasmablasts and HSCs are reported at the top right of the UMAP plots (e.g., for MetaCell, *mcRigor two-step* detected 4 trustworthy and 1 dubious plasmablast metacells). Bottom: Bar plots showing the cell-type composition of trustworthy metacells in which at least 50% of the constituent cells belong to the corresponding rare cell type. The number of bars reflects the number of such trustworthy metacells (*mcRigor*: left, *mcRigor two-step*: right).

Supplementary Fig 19: **Capability of mcRigor two-step to resolve rare cell types demonstrated using the *bmcite* dataset with HSC cells downsampled.** **a**, UMAP plot of the *bmcite* dataset with the two rare cell types—plasmablasts and HSCs—highlighted. **b–d**, Demonstration of mcRigor two-step’s ability to recover these rare cell types when applied to metacell partitions generated by SEACells (**b**), SuperCell (**c**), and MetaCell (**d**). Top: Single-cell UMAP plots showing the metacell partitions produced by the original method (left) and by mcRigor two-step (right), with dubious metacells identified by either mcRigor or mcRigor two-step highlighted in red circles. The numbers of trustworthy and dubious metacells representing plasmablasts and HSCs are reported at the top right of the UMAP plots (e.g., for MetaCell, mcRigor two-step detected 3 trustworthy and 2 dubious plasmablast metacells). Bottom: Bar plots showing the cell-type composition of trustworthy metacells in which at least 50% of the constituent cells belong to the corresponding rare cell type. The number of bars reflects the number of such trustworthy metacells (mcRigor: left, mcRigor two-step: right).